# Measuring Intent Comprehension in LLMs: A Variance Decomposition Framework

## Abstract

People judge interactions with large language models (LLMs) as successful when outputs match what they want, not what they type. Yet LLMs are trained to predict the next token solely from text input, not underlying intent. Because written language is an imperfect proxy for intent, and correlations between phrasing and desired outcomes can break down in training data, models that rely too heavily on surface cues may respond inconsistently to semantically equivalent prompts. This makes it essential to evaluate whether LLMs can reliably infer user intent—especially in high-stakes settings where robustness and generalization are critical. We introduce a formal framework for assessing intent comprehension in LLMs: whether a model demonstrates robust understanding of user intent by producing consistent outputs across semantically equivalent prompts while differentiating between prompts with distinct intents. Our evaluation approach is based on a variance decomposition of model responses into three components: variability due to user intent, user articulation, and model uncertainty. Models that understand what users want, and are not overly sensitive to textual cues, should attribute most output variance to intent differences, rather than articulation style. Applying this framework across diverse domains, we find that, within the five LLaMA and Gemma models we evaluate, larger models typically assign a greater share of variance to intent, indicating stronger comprehension of intent, although gains are uneven and often modest with increasing model size. These results motivate moving beyond accuracy-only benchmarks toward semantic diagnostics that directly assess whether models understand what users intend.

## 1 Introduction

Human communication involves a fundamental socio-cognitive process: we form an intent we want to articulate, then we choose words to express it, hoping the recipient will sympathetically decipher our intended meaning (Smith (1759)). Consider a frustrated traveler at an airport asking "Is there any way to get to Terminal B faster?" They might equally say "I need to catch my flight—what's the quickest route to B?" or "Terminal B is so far—any shortcuts?" They could even ask indirectly, "How do I get to the gate for flight 718?" Each phrasing differs dramatically, yet all express the same underlying intent: finding the fastest path to their destination. Much communication, therefore, centers on extracting underlying intent from observable signals, with success depending on the recipient's ability to see past surface variation to grasp the purpose beneath.

This challenge of intent extraction becomes particularly critical for large language models (LLMs), where all interaction occurs through text, making the model's ability to understand user intent the first step in effective human-AI interaction. Despite the centrality of intent understanding, most model evaluation approaches focus on whether models can perform specific tasks. These evaluation frameworks assume users express their intentions clearly, but this assumption often fails in practice. Users frequently struggle to articulate their needs and may revise their requests multiple times until they express their intent clearly. This implies that for models to be reliable from a user perspective, they must not simply respond to literal text, but rather infer the underlying intent from limited textual cues, disregard unimportant surface variations, and respond to the user's true purpose.

In this paper, we propose a framework for measuring how well models capture users' underlying intentions. We treat intent as a latent variable that underlies every prompt: while the same purpose

can be expressed in many ways, a model that understands intent should produce consistent response distributions across surface variations, yet shift appropriately when purpose changes. We define intent comprehension as the property that responses remain invariant to phrasing when purpose is fixed, but vary systematically when purpose changes.

To operationalize this idea, we present a diagnostic method that decomposes variation of model outputs into three components: Intent Sensitivity, the share of variation due to changes in purpose or intent; Articulation Sensitivity, the share due to phrasing variations; and Model Uncertainty, the residual variation stemming from the model's inherent uncertainty. The variation in a model that understands user intent should exhibit high Intent Sensitivity and low Articulation Sensitivity, indicating consistent meaning extraction that disregards superficial linguistic cues.

Because intent is unobserved and generating equivalent prompts that differ only in form is non-trivial, we construct semantically equivalent prompts through cross-lingual translation, inspired by universal semantic structures across languages (Youn et al. (2016)). Starting from a base prompt, we translate it through a sequence of typologically diverse languages and back to English, inducing natural variation in phrasing while preserving intent. Translation serves this purpose well because its core objective aligns with our needs: altering surface form while maintaining meaning. Additionally, LLMs demonstrate strong automated translation capabilities, making this approach both theoretically motivated and practically feasible.

Using this pipeline, we evaluate five language models from the LLaMA and Gemma families of varying sizes across tasks in health, logistics, finance, travel, and social planning. Our findings show that within each family, the higher-parameter variants tend to attribute a larger share of output variability to changes in user intent (Intent Sensitivity), indicating stronger alignment with user goals and more coherent internal representations. However, this improvement is not uniform: the larger models do not consistently outperform smaller models across all domains, and its robustness advantages are often modest. Larger models also demonstrate greater sensitivity to prompt articulation, reflecting a trade-off between semantic generalization and responsiveness to surface cues. Furthermore, model robustness varies significantly across domains. These findings demonstrate our framework's diagnostic value and highlight the need for more targeted evaluations of models' ability to respond to user intent rather than varied textual cues.

**Intent in Computational Models.** The concept of intent has emerged as a fundamental construct in artificial intelligence to bridge observable behaviors and underlying cognitive states. In natural language processing, intent traditionally represents the underlying purpose or goal behind user utterances (Qin et al. (2021); Zhang and Wang (2022)), evolving from early slot-filling paradigms to sophisticated neural architectures (Louvan and Magnini (2020)). This aligns with philosophical foundations in Brentano's intentionality theory—the directedness of mental states toward objects (Jacob (2019))—and Gricean pragmatics, which emphasizes the role of communicative intentions in meaning (Grice (1957; 1989)). Recent work has formalized these intuitions through a Bayesian calculus, most notably Baker, Tenenbaum, and Saxe's inverse planning framework (Baker et al. (2007; 2009)), which models intent recognition as Bayesian inference over an agent's goals given observations of their actions under an assumption of approximate rationality.

**Contributions**   This paper makes three key contributions. First, we use these foundations to propose a formal definition of intent comprehension, specifically tailored to language models, grounded in the notion of response invariance to surface variations. Second, we introduce a variance decomposition method that quantifies the relative contributions of intent, phrasing, and uncertainty to model outputs, providing interpretable measures of model behavior. Third, we apply this method to real-world tasks across multiple domains, demonstrating how our framework reveals meaningful differences in the the ability of models to understand users intent.

## 2 CONCEPTUAL FRAMEWORK

In this section, we provide the conceptual foundation for our proposed measure in section 3. Let $T \in \mathcal{T}$ denote users' intent. This intent or purpose represents the communicative goal underlying a user's request to the model. Users articulate their intent through prompts $p_i$, which may vary widely in surface form even when seeking to convey the same underlying objective. For example, the intent of knowing the capital of France may be expressed through prompts such as "What's the capital of

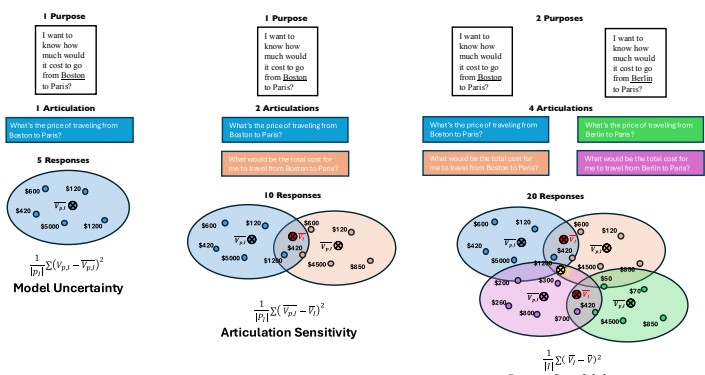

Figure 1: Conceptual Illustration of our decomposition Measures

France?" or "Which city is France's capital?". Prompts should be thought of as the full textual inputs provided to the model that captures the implied intent of the user. Finally, let $a \in \mathcal{A}$ be the model response.

We say that a language model understands the user's intent if it produces identical response distributions for requests or prompts that express the same underlying intent, and distinct response distributions for prompts that express different intents. We formalize this as follows:

**Definition 2.1 (Intent Comprehension)** *Let $\tau : \mathcal{P} \to \mathcal{T}$ be the* ground-truth *mapping that assigns each prompt an intent label. A language model that generates a response distribution $\pi$ over $\mathcal{A}$, is said to understand the user's intent if, for every pair of prompts $p_i, p_j \in \mathcal{P}$, the following properties hold:*

1. **Consistency**: *If $\tau(p_i) = \tau(p_j)$, then*

$$\pi(\cdot \mid p_i) \; = \; \pi(\cdot \mid p_j).$$

2. **Sensitivity**: *If $\tau(p_i) \neq \tau(p_j)$, then*

$$\pi(\cdot \mid p_i) \; \neq \; \pi(\cdot \mid p_j).$$

Our definition requires a model that understands user intent to distinguish between surface-level variations in articulation and substantive changes in user intent or purpose. In particular, a model that varies its response distribution in reaction to inconsequential articulation changes is likely overfitting to superficial correlations in the training data rather than genuinely understanding the user's underlying purpose.

**Remark 1** *Our definition of* Intent Comprehension *is directly tied to the broader notion of a* world model. *In robotics and model-based control, agents infer a latent state $s_t$ and learn dynamics/observation maps $(f, g)$ so that behavior depends on $s_t$ rather than raw measurements. Performance is typically judged by* correctness-centric *criteria such as one-step prediction error, trajectory likelihood, or task return under a known oracle. In our text-only setting, the prompt stream serves as observations, and the payoff-relevant latent state is the user's intent. As in robotics, a language model with a usable world model should (i) behave* invariantly *across paraphrases that preserve this state and (ii) shift its output distribution when the state (intent) changes. Crucially, unlike standard evaluations of world models—which assess both invariance across equivalent states and whether outcomes are* correct *relative to an external oracle—IC is* consistency-centric*: it tests whether the LLM can identify and respond to the latent state rather than the surface cues. IC therefore focuses on the first property of a world model: recognizing its latent state. See more in Appendix C.1.*

**Remark 2** *In our framework, intent captures what the user wants the model to say—specifically, the expected distribution over exact responses. Two prompts have the same intent if they reliably elicit the same output distribution from the model, even if they differ in wording or domain. For example, "Will a fair coin land heads or tails? Answer 1 for heads, 0 for tails" and "I drew a number from [-1, 1]. Is it positive or negative? Answer 1 for positive, 0 for negative" differ in content but share the same intent: both aim to produce a 50/50 distribution over identical outputs—'1' and '0'.*

Our criteria for whether a model has intent comprehension may be over-exacting. Precisely defining intent is inherently difficult due to its latency and potential ambiguity. Evaluating model responses adds another layer of complexity. This often requires estimation of the distribution of full-text responses, which are inherently high-dimensional (e.g., a detailed description of a driving itinerary from Boston to Miami.) Focusing on the distribution of responses, an evaluator may view distinct answers as carrying the same underlying meaning. For example, in response to the prompt "How much is $1 + 2$?", a model might answer "three" or "3." Although slight variations in the prompt may influence the likelihood of generating "three" versus "3," we should still regard the model as consistent.

To overcome these challenges, we propose a more relaxed notion of intent comprehension: a *sufficient* intent comprehension. This concept softens the standard definition by introducing an evaluator who judges prompts and responses. Specifically, we assume the existence of two functions: one that determines whether two prompts express the same intent, and another that assigns values to different responses. We say that a language model understands an the user intent if, whenever two prompts are judged to share the same intent, the distributions over evaluator-assigned values of responses remain identical. In this sense, we say that it's sufficient to say that a model understands the user's latent intent if the user cannot distinguish between model responses to the same intent.

Formally, let $\tilde{\tau} : \mathcal{P} \to \mathcal{T}$ map from prompts to intents (i.e., articulations to purposes), defined according to some evaluation criterion. Let $V : \mathcal{A} \to \mathbb{R}$ be a function that maps responses to values. Finally, let $\pi_V(\cdot \mid p)$ denote the distribution over response values induced by model responses to prompt $p$, as evaluated by $V$. The relaxed definition is as follows:

**Definition 2.2 (Sufficient Intent Comprehension with respect to an evaluator $\tilde{\tau}$ and $V$)** *Let $\tilde{\tau} : \mathcal{P} \to \mathcal{T}$ be a fixed intent–evaluator that assigns an intent label for each prompt, and let $\pi_V(\cdot \mid p)$ be the value distribution induced by the model, in response to prompt $p$. A language model possesses a sufficient intent comprehension with respect to $\tilde{\tau}$ and $V$ if for every pair of prompts $p_i, p_j \in \mathcal{P}$:*

  *1. (Sufficient Consistency) If $\tilde{\tau}(p_i) = \tilde{\tau}(p_j)$, then*
  $$\pi_V(\cdot \mid p_i) = \pi_V(\cdot \mid p_j).$$

  *2. (Sufficient Sensitivity) If $\tilde{\tau}(p_i) \neq \tilde{\tau}(p_j)$, then*
  $$\pi_V(\cdot \mid p_i) \neq \pi_V(\cdot \mid p_j).$$

Compared to Definition 2.1, this relaxed notion makes two changes: the intent map $\tilde{\tau}$ is evaluator–dependent, and consistency is enforced only on the pushforward distributions of evaluated responses $V(a)$, rather than on the full response distributions over $\mathcal{A}$.

**Remark 3** *Both Intent Comprehension and sufficient Intent Comprehension criteria focus on measuring consistency, rather than assessing whether a language model's responses are factually correct. This is because our primary goal is to evaluate how well the model understands user intent—not whether it produces the correct answer. This approach differs from traditional benchmarks, which typically rely on a binary notion of correctness, classifying answers as strictly right or wrong. To understand why consistency matters, consider a language model trained only on economic data from 2020. If asked about economic conditions in 2023, it may consistently produce outdated yet internally coherent answers across differently phrased questions, reflecting the information on which it was trained. While these answers are incorrect in light of present-day facts, their internal consistency suggests that the model is responding to the user's underlying intent, and not simply their textual variation.*

Our definition of a (sufficient) intent comprehension is motivated by LLM training data, which consists of human-generated text, replete with spurious correlations between writing style and user intent. Small stylistic changes—like tone or word choice—often co-occur with shifts in latent attributes of the writer, such as personality, mood, or communicative norms, even when the underlying intent and objective remains constant. This makes prompt phrasing a confounded signal, blending true intent with incidental articulation. As a result, a model may change its responses not because users change their intent, but because it has learned correlations between superficial cues and different responses. Evaluating whether an LLM understands the latent user intent, thus mirrors the causal inference problem: Can it separate variation due to intent from variation due to articulation? In a hypothetical world where prompt phrasing perfectly reveals intent and carries no noise, models would trivially satisfy our definition. But in reality, disentangling intent from correlation is non-trivial—making consistency a meaningful test of deeper understanding.

## 3 MEASURING (SUFFICIENT) INTENT COMPREHENSION

In this section, we present a simple method, motivated by our conceptual framework, for evaluating whether an LLM understands the user's latent intent. We assume access to a sample of prompts along with corresponding model responses. Some prompts are designed to share the same underlying intent, while others are not. In the experimental section, we provide a detailed description of how we construct such a sample.

Our evaluation approach is based on a variance decomposition of model responses. Specifically, we break down response variance into three components: *Intent Sensitivity* (IS), which captures the model's responsiveness to changes in the intent or purpose of the input prompt; *Articulation Sensitivity* (AS), which reflects the model's sensitivity to variations in how users express the same intent; and Model Uncertainty (MU), which accounts for the inherent ambiguity or variability in the model's responses.

To formalize these components, we consider a domain $D$ consisting of related purposes or intents $I \in D$. Each intent can be expressed through multiple prompts $p_I \in \tau(I)$, and we denote the model's value of response to a prompt $p$ as $v$. To enable comparability across tasks, domains, and models, we define the standardized response as $\tilde{v} = \frac{v}{\text{std}(v|D)}$. Furthermore, let $R_I^2$ be the standard coefficient of determination, the proportion of total variance explained by differences in intent, and $R_{p_I}^2$ be the proportion of variance explained by differences in prompt phrasing, holding intent $I$ fixed. With this notation, we define our three core measures, sketched in Figure 1, as:

$$
\begin{aligned}
IS &= Var_I(\mathrm{E}[\tilde{v}|I]) = R_I^2 && \text{(Intent Sensitivity)}, \\
AS &= \mathrm{E}_I \left[ Var_{p|I} \left( \mathrm{E} \left[ \tilde{v} \mid p_I \right] \right) \right] = \mathrm{E}_I \left[ R_{p_I}^2 \right] && \text{(Articulation Sensitivity)}, \\
MU &= \mathrm{E}_I \left[ \mathrm{E}_{p|I} \left( Var \left( \tilde{v} \mid p_I \right) \right) \right] = \mathrm{E}_I \left[ 1 - R_{p_I}^2 \right] && \text{(Model Uncertainty)}.
\end{aligned}
$$

The first term, IS, is the variance of the conditional expectation of responses given intent. It reflects the sufficient Intent Comprehension property in 2.2, quantifying how much the model's mean response shifts with changes in the user's underlying purpose. A model appropriately sensitive to intent should produce distinctly different responses for different tasks. Therefore, we expect IS to be high in models that understand the user's latent intent or purpose. The second term, $AS$, corresponds to the consistency property. It quantifies how much, on average, the model's mean response varies with changes in phrasing or wording of the prompt, holding the user's purpose fixed. A model that understands the user's latent intent, rather than being swayed by syntactic or stylistic differences, should exhibit low sensitivity to surface-level changes in articulation.

The third term, $MU$, captures the residual variance that remains after conditioning on both intent and phrasing. This component conflates at least three sources: (i) sampling stochasticity (e.g., due to temperature or nucleus sampling); (ii) epistemic uncertainty, where the model lacks a confident internal representation and spreads probability mass over multiple plausible answers; and (iii) aleatoric uncertainty, which is inherent to the task itself (e.g., when the prompt is "Tell me a random joke"). Whether a high MU is desirable depends on the context. For deterministic tasks with a clear correct answer (e.g., "What is 1+1?"), high MU reflects unwanted uncertainty. In contrast, for inherently open-ended or subjective tasks—such as those we examine in the experimental section—some degree of response variability is expected and even appropriate.

These three measures are all non-negative and sum to one: $IS + AS + MU = 1$. This identity ensures that each term represents the relative contribution of a distinct source of variation in model responses. Together, they provide a principled way to assess the robustness and interpretability of a model's behavior.

An alternative perspective on this decomposition is in terms of $R^2$: how much of the variation in model responses is predictable from different parts of the request. From this viewpoint, the normative target is clear: *there should be essentially no predictive signal in the paraphrase once intent is fixed*, and nearly all systematic variation should come from changes in intent. While we use an $R^2$-style "explained variation" framing to connect directly to the equations, the same idea can be operationalized with other performance metrics better matched to the output type—for example AUC for binary decisions, accuracy/F1 for discrete labels, or log loss/cross-entropy for probabilistic outputs. We emphasize the variance decomposition because it is model-agnostic, easy to implement, and intuitive to interpret, but other approach to operationalize the intent comprehension measures are possible.

We also define two summary statistics that condense the decomposition into intent-centric diagnostics. The first is the Meaningful Variability Share (MVS),

$$\text{MVS} = \frac{IS}{IS + AS},$$

which serves as a signal-to-noise ratio over the *explainable* portion of variability: among the variance attributable to request features—either genuine changes in intent ($IS$) or superficial changes in articulation ($AS$)—MVS measures how much is signal rather than surface noise. A high MVS indicates that most of the model's explainable variation reflects meaningful distinctions between user intents, whereas a low MVS suggests the model is overly influenced by superficial differences in wording, indicating a lack of intent understanding. Another interpretation is that, when decoding is effectively deterministic (temperature near zero) so that residual randomness is negligible, MVS approximates the two-way split between intent and articulation, because variability then stems only from intent and phrasing.

To align directly with our definition of sufficient intent comprehension—distributions should move with intent (sensitivity) and remain invariant to non-intent articulation (consistency)—we combine MVS with the overall share of total variance due to intent into a single index,

$$\text{ICI} = \frac{2\,\text{MVS} \cdot IS}{\text{MVS} + IS} \in [0, 1],$$

which we call the *Intent Comprehension Index (ICI)*. ICI is the harmonic mean of "intent purity" and "intent coverage." The MVS component captures the sufficient consistency by penalizing articulation-driven variability: if non-intent phrasing moves the distribution, $AS$ rises, MVS falls, and so does ICI. The $IS$ component captures the sufficient sensitivity clause by rewarding models whose outputs shift when intent changes: if the distribution barely moves with intent, $IS$ is small and ICI again falls. Therefore, the ICI summarizes the definition of intent comprehension.

This construction mirrors the classic precision–recall analogy in information retrieval. The $MVS$ term plays the role of precision because, among the variance that is systematically explained by features of the request (intent or articulation), it measures the fraction that is aligned with intent rather than surface form. In this analogy, $IS$ is the analogue of "true positives," while AS acts like "false positives": variance that is explained by the request, but driven by articulation rather than intent. The recall-like quantity, $\text{IS}/(\text{IS} + \text{AS} + \text{MU}) = \text{IS}$, instead asks what fraction of *all* observed variability in responses is driven by intent; the remaining components, $\text{AS} + \text{MU}$, are then analogous to "false negatives," capturing variation that does *not* move with changes in intent—either because the model is reacting to non-intent phrasing (AS) or because its behavior is intrinsically noisy (MU). Read this way, ICI is an F1-type score over units of variance: it is large exactly when intent-driven variation is both pure (high MVS) and covers a large share of total variability (high IS).

**Intuitive interpretation**. Our decomposition asks a simple question: when the model's answers change, why do they change? Intent Sensitivity captures variation driven by genuine differences in what users are trying to do. Articulation Sensitivity captures variation driven by superficial differences

in how users phrase the same request. Model Uncertainty is the residual: randomness from sampling, uncertainty in the model's beliefs, or genuine task ambiguity. The three components sum to one, forming a "pie chart" of the model's behavior. A model that truly understands intent will place most of the mass on IS, little on AS, and only as much MU as the task naturally requires. The ICI is high exactly when responses are primarily driven by intent rather than phrasing or noise.

> **Remark 4** *In practice, when estimating the components of the decomposition, we deliberately restrict attention to domains where requests share a common structure but differ in intent. This controlled setup reflects a conservative and more stringent approach to evaluating whether the model understands the user's intent. By fixing the overall structure of the query and varying only a key element that changes the underlying intent (such as modifying the income level in a tax-related prompt while keeping the rest of the wording unchanged), we eliminate spurious cues and force the model to rely on its understanding. If the model systematically adapts its output to such minimal yet meaningful changes, it suggests the presence of a coherent internal representation capable of capturing user intent and purpose.*

Our variance decomposition is one concrete operationalization of sufficient intent comprehension, but it is not the only way to quantify whether a model is responding to *intents* rather than to surface form. Conceptually, the core behavioral question we ask is whether prompts that share the same intent but differ in articulation (paraphrases, typos, different levels of detail) induce similar response distributions, while prompts that differ in intent induce systematically different distributions. The IS–AS–MU decomposition captures this by attributing variation either to changes in intent (IS), to changes in articulation conditional on intent (AS), or to residual model uncertainty (MU). Other scoring rules or distances over response distributions could in principle be plugged into the same logic; the key desideratum is that they separate variation due to intent from variation due to articulation and noise.

In the Appendix, we show how our decomposition framework extends to settings where the model produces free-form text (Appendix **??**) or discrete, categorical outputs (Appendix **??**). The key challenge in the discrete case is that variance is not naturally defined—any numeric encoding of categories is arbitrary. We address this by replacing the variance of the outcome v with its Shannon entropy H(v). Using the chain rule for entropy, we derive an analogous decomposition $H(v) = \text{IS} + \text{AS} + \text{MU}$. Here, $IS$ becomes the mutual information $I(I; v)$ between intent and output; AS becomes a conditional mutual information term that captures the additional information about $v$ conveyed by the articulation $p_I$ beyond intent; and MU becomes the remaining conditional entropy $H(v \mid I, p_I)$, representing residual unpredictability. Normalizing by $H(v)$ again yields unit-sum shares that play the same role as in our variance-based decomposition. Appendix **??** provides an illustrative application in which the model is tasked with ranking and approving loans and returns only categorical decisions, demonstrating how the entropy-based decomposition operates in practice.

## 4 EXPERIMENTS

In this section, we provide a brief description of how we construct our experiments; a detailed description is presented in Section F of the Appendix. We begin by designing an automatic question generator. Our questions focus on open-ended, guesstimation-style tasks. These are chosen because they naturally elicit a wide distribution of plausible responses, which is essential for disentangling intent sensitivity from articulation sensitivity and model uncertainty. We utilize an LLM (GPT-4.1) to construct 24 questions across five domains—transportation, personal finance, health and nutrition, logistics, and social planning—via a two-stage pipeline: first, we generate unit-specified templates with placeholders, and then we populate them with realistic values to define specific intents. For each task, we generate 12 distinct intents. This procedure yields a diverse and structured set of quantitative estimation tasks that reflect naturalistic usage while remaining well controlled.

To evaluate articulation sensitivity, we need to generate sets of prompts that convey the same intent. This is non-trivial, as creating prompts with exactly equivalent definitions is difficult even for humans. To address this, we generate equivalent prompts through cross-lingual translation chains, leveraging structural differences across languages to induce lexical and syntactic diversity while preserving meaning. GPT 4.1-based checks filter paraphrases to ensure intent equivalence, and we then select a

diverse final set using an embedding-based approach. Each paraphrase is paired with varying input values and repeatedly posed to different LLMs. Specifically, for each of the 24 tasks, we generate 10 paraphrases. We then submit these prompt variations to five different language models—LLaMA 3.2 3B, LLaMA 3.1 8B, LLaMA 3.3 70B (Aaron Grattafiori et al. (2024)), and Google's Gemma 3 12B-IT, and 27B-IT (Gemma Team: Aishwarya Kamath et al. (2025a))—requesting 25 responses for each prompt–intent pair. Unless otherwise noted, we fix the sampling temperature at 1 for all models. This places each model in a single, untuned "natural" decoding regime, so that differences in IS/AS/MU reflect the models themselves rather than model-specific temperature tuning; Appendix G explores other temperature setting. In total, we obtain 1,772,492 model responses, averaging 354,499 per model[1]. Finally, we perform the variance decomposition described in Section 3. To mitigate finite-sample issues, we apply ANOVA for the variance decomposition, which we discuss further in Section D of the Appendix. Human evaluation, described in Appendix B, verifies that our paraphrases largely preserve intent and that our numeric extraction is reliable, with high inter-annotator agreement for both checks.

## 4.1 RESULTS

We now turn to explore our results. Figure 2 presents our key findings across models. The figure shows the average Intent Sensitivity (IS), Articulation Sensitivity (AS), and Model Uncertainty (MU) across all tasks and across models. The figure reveals that the two smallest models, LLaMA 8B and LLaMA 3B, exhibit very high MU across tasks, accompanied by both low IS and low AS. This is accompanied by higher variance in general, as shown in Figure 7. In contrast, the three larger models—LLaMA 70B, Gemma 27B, and Gemma 12B—show comparable results across these three components, with the largest share attributed to IS. LLaMA 70B shows slightly higher MU. Within our evaluation, this indicates that models with more parameters in these families are not uniformly superior, and that high IS can be achieved by a 12B-parameter Gemma model despite its smaller size relative to LLaMA 70B. As the two families differ in training practices and training data this indicate that high IS can be achieved within smaller models.

Figure 3[2] presents our results for the Intent Comprehension Index (ICI), Meaningful Variability Share (MVS), and IS across models. The blue line illustrates that, in this five-model set, the higher-parameter LLaMA and Gemma variants tend to yield higher IS, indicating that changes in user intention are more effectively reflected in adjustments to the response distribution. Specifically, the smaller

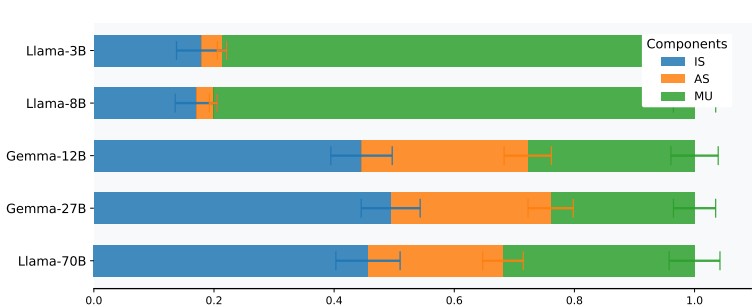

Figure 2: The Average Variance Decomposition Across Models, shaded area is 95% CI

LLaMA models account for approximately 20% of the response variation due to changes in intention, while the larger models account for around 50%. Because these models differ in architecture and training regimes as well as parameter count, we treat this as a descriptive association rather than a causal scaling law.

The second component, MVS, shown by the orange line, demonstrates an inverse pattern: the two smaller models exhibit higher MVS than their larger counterparts. This implies that smaller models respond relatively more to changes in intent than to changes in prompt articulation, aligning with the general lower responsiveness suggested by their Intent Sensitivity scores. Interestingly, larger models, in are evaluation, show greater sensitivity to articulation shifts. This suggests that some larger models are more sensitive to prompt text, therefore more responsive to intent, but at the cost of greater overall

---

[1] We only keep track of valid responses, but sometimes the model fails to reach the desired set of responses within 125 attempts, and in such cases, we move on to the next section.

[2] Full distribution of the component for each tasks, by model, is in figure 8 in the Appendix

susceptibility to surface-level prompt variations, reflecting overfitting rather than an emerging deeper understanding in larger models.

Finally, examining the Intent Comprehension Index, we observe higher values for larger models, but the improvement appears to plateau—the difference between the 70B model and the 12B model is modest. This suggests that intent comprehension is not necessarily an emergent property that scales linearly with model size, and that substantial improvements in understanding user intent can be achieved without requiring the largest available models.

In Figure 4, we examine heterogeneity in model performance across the five topical areas. The results indicate that larger models are not uniformly superior across domains; instead, their sensitivity varies by topic. For example, the largest model, LLaMA 70B, achieves the highest IS scores in Health and Nutrition and Transportation, whereas LLaMA 27B performs best in Logistics, Personal Finance, and Social Planning. The figure further highlights differences in the share of AS: questions in Social Planning are consistently more sensitive to writing style across models than those in Health and Nutrition, Transportation, or Personal Finance. Taken together, these findings suggest that model capability is domain-dependent. In this way, intent comprehension is not a uniform property of models but rather varies systematically across topical areas.

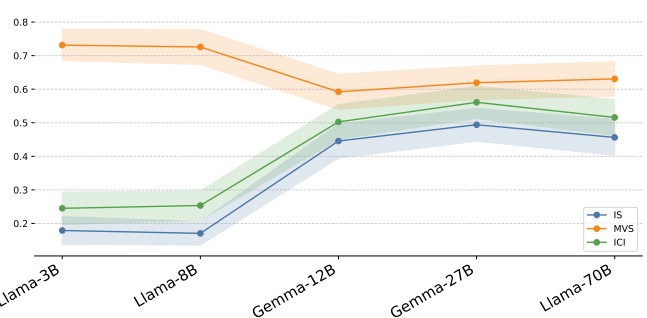

Figure 3: ICI, MVS and IS across models, shaded area is 95% CI

The differences we observe across models reflect not only scale but also model family. These family-level gaps are more plausibly attributed to training procedures than to data access: both LLaMA 3 and Gemma 3 are trained on web-scale, multi-trillion-token mixtures (Meta AI (2024), Gemma Team: Aishwarya Kamath et al. (2025b)). Yet their training pipelines diverge sharply. LLaMA 3 relies primarily on supervised instruction tuning and preference-based alignment, whereas Gemma 3 incorporates knowledge distillation during pre-training and follows a structured four-stage post-training recipe: distillation from a larger instruct teacher, RLHF, RL from machine feedback for math, and RL from execution feedback for code. This contrast suggests that training design—rather than size or corpus alone—can substantially influence a model's ability to infer and match a user's latent intent, precisely the behavior our intent measure targets.

## 5 DISCUSSION AND LIMITATIONS

In this project, we present a formal framework to assess whether LLMs understand the user's latent intent. Instead of focusing solely on whether the model responds correctly to a question, we decompose the response variability of the model into user intent (Intent Sensitivity), irrelevant information (Articulation Sensitivity), and intrinsic randomness (Model Uncertainty). Applying this framework across models of varying sizes and across families and domains, we observe that larger models in our set of models generally exhibit a more robust understanding of users' intent —but an increased number of parameters does not guarantee better intent comprehension. Difference across llama and gemma model also indicate that different training approaches and data sources may contribute to the model sensitivity to the intent and articulation.

In some cases, we find that smaller models outperform larger ones, and robustness varies across different domains, underscoring the need for evaluation across diverse contexts. Our findings emphasize semantic consistency and generalization as key indicators of model quality, offering a scalable method to distinguish between true understanding and pattern matching.

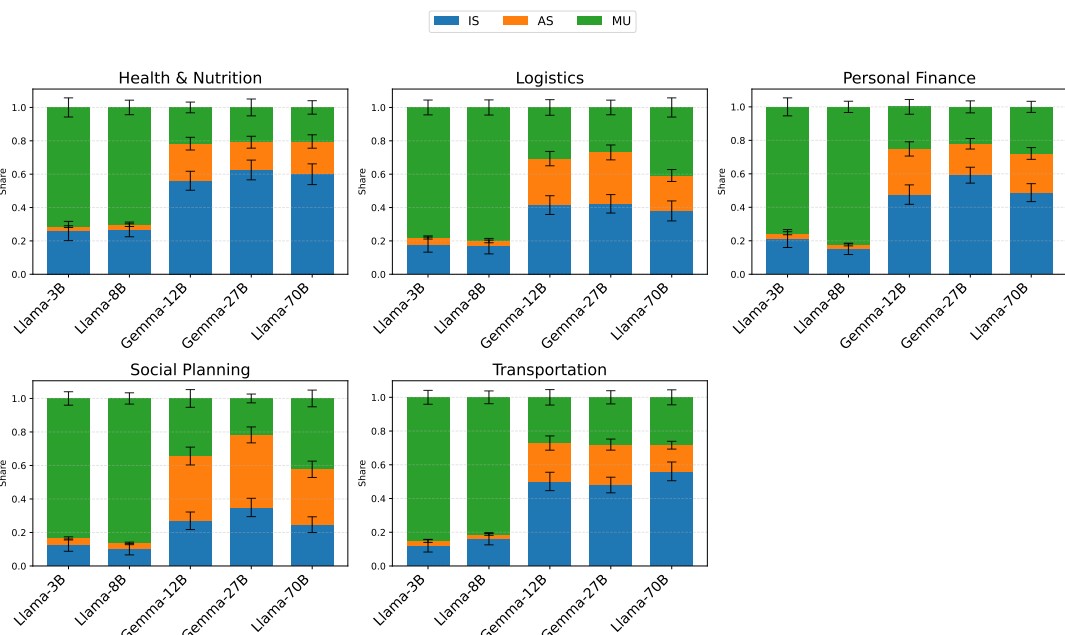

Figure 4: Variance Decomposition across the different topics, with black bars indicating the 95% confidence intervals from using bootstrap with 100 repetitions.

We emphasize that our notion of intent comprehension is not meant to replace standard accuracy metrics, but to complement them. Accuracy evaluates whether a model's answers are correct, whereas our measure quantifies whether its responses track changes in a user's underlying intent while remaining stable across paraphrases. In this sense, our framework helps distinguish genuine sensitivity to intent from brittle pattern matching on surface form. See Appendix E for a detailed discussion on incorporating correctness in our framework.

For practitioners, the IS–AS–MU decomposition offers a comprehensive risk assessment framework suitable for informed decision-making in LLM-enabled products. Accuracy tests alone tell a developer or user whether a model can answer a benchmark question; our metric tells them how the model will behave when real users inevitably rephrase, typo, or embed multiple requests in one prompt. Our metric also informs practitioners on potential fairness issues. Articulation sensitivity often correlates with dialect, accent, or education level; a model whose answers swing with phrasing differences can systematically disadvantage certain groups. Tracking AS gives model producers a concrete, quantitative way to demonstrate equity, complementing broader accuracy and bias checks.

Our measurement approach has several limitations. First, we only consider the first and second moments of response distributions. Although some decision-makers may prefer higher-order statistics or full distributional analyses, our method strikes a balance between complexity and practicality: mean and variance already capture key risks for many applications. Second, in settings where evaluators must assign qualitative categories—such as "great," "ok," "mid," "bad," or "worst"—subjective judgments can lead different annotators to produce different orderings. In such cases, results should be accompanied by sensitivity analyses across scoring rules or annotators to ensure robustness.

Our research highlights the importance of understanding the intent that drives users to utilize LLMs and other AI models. In this study, we show that LLMs often do not fully understand the user intent and respond to differences in superficial phrasing. Future research could more deeply and directly quantify user intent, processes through which intent emerges from preferences or through interaction with generative models, and the inherent limitations of fully expressing complex and original intent. Improved intent identification will enable to better measure models' ability to capture and respond appropriately.

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

# Appendix

## A WORKING WITH NON-NUMERIC RESPONSES

The main text focuses on settings where model outputs are numerical, for which the variance-based decomposition provides a direct and transparent operationalization of our definition of (sufficient) Intent Comprehension. In many applications, however, the desired outputs are discrete, categorical, or fully free-form text. This section outlines how the same conceptual framework extends to these cases by replacing raw responses with an appropriate representation extracted from the model's output. We first describe how to handle unrestricted natural-language responses through value-extraction functions such as labelers, scoring models, and embedding-based representations. We then introduce an analogue of our variance decomposition for discrete and categorical settings based on an entropy-based decomposition of variability.

### A.1 FREE-FORM RESPONSES

Our theoretical framework in Section 3 does not assume that model outputs are numeric or categorical. When models produce unrestricted natural-language responses—explanations, multi-sentence arguments, or other open-ended text—we require a mapping from raw text to an analyzable quantity. This mapping is the value function $V$, after which the variance decomposition proceeds exactly as in the numeric case.

The function $V(a)$ serves to extract the aspect of the response relevant to evaluation. Depending on the application, $V$ may be implemented through human annotation, a domain-specific scoring rule, an evaluator model, or an embedding-based representation. If the analyst wishes to track a specific semantic property (such as stance, polarity, or task success), $V(a)$ can return a scalar or discrete label derived from the text. For more open-ended generation, $V(a)$ may be a continuous embedding vector capturing semantic or stylistic content. In such cases one may analyze individual embedding coordinates, low-dimensional projections, or task-specific directions, each of which functions as the response variable whose variation is decomposed across intent and articulation.

The only requirement is that the representation induced by $V$ meaningfully captures the dimension along which intent comprehension is to be assessed. When this dimension is well defined, $V$ may be a deterministic extractor; when it is diffuse or multidimensional, embeddings offer a natural surrogate. In all cases, once $V(a)$ is specified, the decomposition attributes variation in this induced representation to changes in intent, changes in articulation, or residual randomness in exactly the same manner as with scalar outputs.

### A.2 DECOMPOSITION FOR DISCRETE OUTPUTS

In the main text, we use the variance decomposition for our measure

$$\mathrm{Var}(v) \;=\; \underbrace{\mathrm{Var}\big(\mathrm{E}[v \mid I]\big)}_{IS} \;+\; \underbrace{\mathrm{E}_I\big[\mathrm{Var}(\mathrm{E}[v \mid p_I] \mid I)\big]}_{AS} \;+\; \underbrace{\mathrm{E}_I\big[\mathrm{E}\big[\mathrm{Var}(v \mid p_I) \mid I\big]\big]}_{MU},$$

which is appropriate when $v$ is continuous (or ordinal), as $\mathrm{Var}(\cdot)$ is then well-defined. For categorical or otherwise non-ordinal outcomes, however, variance cannot be uniquely specified without imposing an arbitrary numerical scale. For example, consider a travel agency that uses a chatbot to recommend destinations based on user preferences. From the agency's perspective, the recommendations correspond to discrete destinations. To handle such cases, we propose an information-theoretic analogue, derived directly by applying the chain rule of entropy twice.

For a discrete random variable $X$ with distribution $P(X)$ we write $H(X) = -\sum_x P(X = x) \log P(X = x)$ for its Shannon entropy. We denote the mutual information between random variables $X$ and $Y$ as $I(X; Y) = H(X) - H(X \mid Y) = H(Y) - H(Y \mid X)$, where $H(Y|X)$ is the conditional entropy of $Y$ given $X$, $H(X \mid Y = y) = -\sum_x P_{X|Y}(x \mid y) \log P_{X|Y}(x \mid y)$. Let $v$ be a discrete outcome variable, and denote again $I$ as intent and $p_I$ as the prompt constructed from intent, as defined in the main text.

With these definitions, we obtain the decomposition

$$H(v) \;=\; \underbrace{I(I;v)}_{IS} \;+\; \underbrace{\mathrm{E}_i\big[I(p_I;v \mid I=i)\big]}_{AS} \;+\; \underbrace{\mathrm{E}_i\big[H(v \mid p_I, I=i)\big]}_{MU}.$$

where $IS$ (Intent Sensitivity) is the mutual information between the intent $I$ and the model output quantifies how much output reveals about the intended meaning before the prompt is supplied. $AS$ (Articulation Sensitivity) is now the conditional mutual information that measures the extra information carried by the prompt $p_I$, given the intent. Finally, $MU$ (Model uncertainty) is the remaining conditional entropy that captures irreducible uncertainty in the model's prediction once both intent and prompt are known.

We can further divide by the total entropy to get a unit-sum normalization that facilitates comparison across models:

$$1 = \frac{I(I;v)}{H(v)} \;+\; \mathrm{E}_i\left[\frac{I(p_I;v \mid I=i)}{H(v)}\right] \;+\; \mathrm{E}_i\left[\frac{H(v \mid p_I, I=i)}{H(v)}\right].$$

This expression mirrors the continuous-outcome variance decomposition in structure while remaining well-defined for any discrete output space.

### A.2.1 EXAMPLE OF USAGE FOR THE DISCRETE CASE

We illustrate the discrete decomposition by applying it in a manner analogous to the variance decomposition approach we used in the main text. As an example, we focus on decision problems in consumer credit.

As in the main analysis, the experimental pipeline consists of synthetic intent generation, multilingual paraphrasing of task prompts, large-scale sampling of model outputs across all intent–paraphrase combinations, and an information-theoretic entropy decomposition applied separately for each model and task.

We evaluate two consumer-credit decision problems. The first task, loan approval, is a binary decision problem in which the model outputs $v \in {0, 1}$, where $1$ indicates approval and $0$ rejection. The second task, risk rating, is an ordinal classification problem in which the model assigns a risk score $v \in \{1, 2, 3, 4, 5\}$, with $1$ denoting very low risk and $5$ very high risk. For both tasks, the prompt specifies a detailed natural-language description of the bank's decision policy, and the output label is parsed from the model's response. Any output whose extracted label falls outside the valid set is discarded and the prompt is resampled.

INTENT GENERATION    For each task, we generate a small synthetic set of *intents*—applicant profiles that serve as proxies for underlying world states. Intents are generated automatically using ChatGPT 4.1. The prompt instructs the model to produce twelve diverse and realistic loan applicant profiles, each containing specified attributes (age, income, existing debt, employment characteristics, credit history, loan amount, and loan term). The model returns a JSON list, from which we extract twelve applicant profiles per task. These profiles are shared across all evaluation models and all downstream prompting steps.

PARAPHRASE GENERATION    Paraphrased versions of each base prompt are constructed via back-translation, following the method described in the main text. Each task has a base English prompt template that describes the decision-making role, the policy criteria, the format of the applicant profile, and the required output label. A placeholder token `applicant_profile` is replaced with an intent-specific description. Translations and back-translations—performed again using ChatGPT 4.1—yield candidate paraphrased templates. To ensure paraphrase fidelity, we apply the automatic equivalence check used in the main experiment.

SAMPLING DESIGN    Given the sets of intents and paraphrased templates, we form a full grid of evaluation conditions over tasks, intents, paraphrases, and models. Each task combines twelve intents with ten paraphrases, yielding $12 \times 10 = 120$ prompt templates per task. For each template and each model, we repeatedly sample outputs by instantiating the template with the corresponding applicant

profile and submitting the resulting prompt. We evaluate the same five instruction-tuned models used in the main experiment. For each $task, intent, paraphrase, model$ tuple, we target 25 valid samples, where validity requires that the parsed output lies within the task's label set. Across both tasks, intents, and paraphrases, this yields 6,000 samples per model and 30,000 samples in total.

EMPIRICAL DECOMPOSITION    To measure intent comprehension in the discrete case, we apply empirical plug-in estimators for the decomposition in Appendix **??**. The observed data consist of $(v_n, i_n, p_n)_{n=1}^{N}$, where $v_n$ is the discrete model output, $i_n$ the intent index, and $p_n$ the paraphrase index. For any $(v, i, p)$, define the empirical joint distribution $\hat{p}(v, i, p) = \frac{1}{N} \sum_{n=1}^{N} \mathbf{1} v_n = v, ; i_n = i, ; p_n = p$, with the corresponding marginals obtained by summation and conditional distributions formed whenever denominators are nonzero. Using these empirical distributions, we compute plug-in estimators for the relevant entropies:

$$\widehat{H}(v) = \sum_v \hat{p}(v) \log \hat{p}(v),$$

$$\widehat{H}(v \mid I) = \sum_i \hat{p}(i)\Big(* \sum_v \hat{p}(v \mid i) \log \hat{p}(v \mid i)\Big),$$

$$\widehat{H}(v \mid p_I, I) = \sum_{i,p} \hat{p}(i, p)\Big(* \sum_v \hat{p}(v \mid i, p) \log \hat{p}(v \mid i, p)\Big).$$

The empirical mutual informations are

$$\widehat{I}(I; v) = \widehat{H}(v) - \widehat{H}(v \mid I), \qquad \widehat{I}(p_I; v \mid I) = \widehat{H}(v \mid I) - \widehat{H}(v \mid p_I, I).$$

Finally, we compute the empirical shares

$$\widehat{IS} = \frac{\widehat{I}(I; v)}{\widehat{H}(v)}, \qquad \widehat{AS} = \frac{\widehat{I}(p_I; v \mid I)}{\widehat{H}(v)}, \qquad \widehat{MU} = \frac{\widehat{H}(v \mid p_I, I)}{\widehat{H}(v)},$$

with all shares set to zero whenever $\widehat{H}(v) = 0$. We construct 95% confidence intervals using the percentile bootstrap with $B = 100$ replications.

**Results**    Figures 5 and 5 report the information-theoretic decomposition for *loan approval* and *risk rating*, respectively. Across both tasks, we observe a clear separation between model families and scales. The smaller LLaMA models allocate a large share of total label entropy to model uncertainty and relatively little to intent sensitivity, indicating that the realized decision label remains hard to predict even after conditioning on the applicant profile (intent) and the specific paraphrase. In other words, much of their variation looks like "intra-condition wobble" rather than systematic differentiation across applicants. By contrast, Gemma-12B and Gemma-27B exhibit substantially higher IS, implying that most of the explainable variation in discrete decisions is driven by changes in applicant profiles rather than by superficial prompt differences; these models behave more like stable decision rules that react to the underlying state.

A notable nuance is that LLaMA-70B tends to sit between these extremes: it shows strong IS (it reacts to intent), but also a visibly non-trivial articulation sensitivity, component. In this discrete setting, AS has a concrete interpretation: conditional on the same applicant profile, some paraphrases systematically tilt the predicted label distribution. For a high-stakes decision framing, this is exactly the failure mode our framework is designed to surface—outcomes can become partially a function of how the request is phrased rather than what the applicant profile implies.

Comparing tasks also hints why articulation effects can matter more in ordinal prediction than in binary decisions. The risk rating task introduces "adjacent-category ambiguity": even when a model has the right qualitative assessment, mapping that assessment into one of five discrete bins (e.g., 3 vs. 4) can be more susceptible to framing. Thus, increases in AS for risk rating are especially informative: they suggest not just noise, but potentially a prompt-dependent calibration of the decision threshold on the ordinal scale.

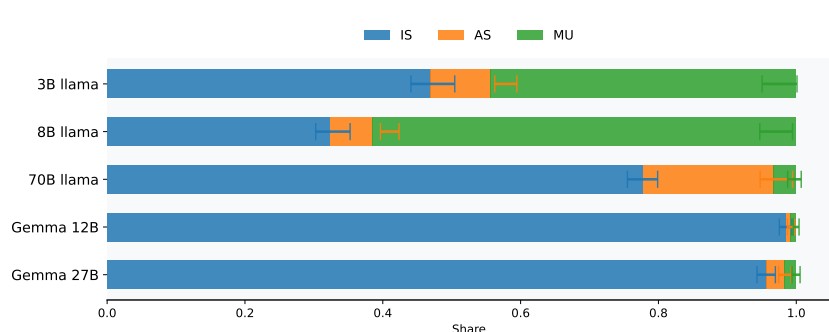

Figure 5: Loan Approval - Information Theoretic Decomposition of Intent Comprehension

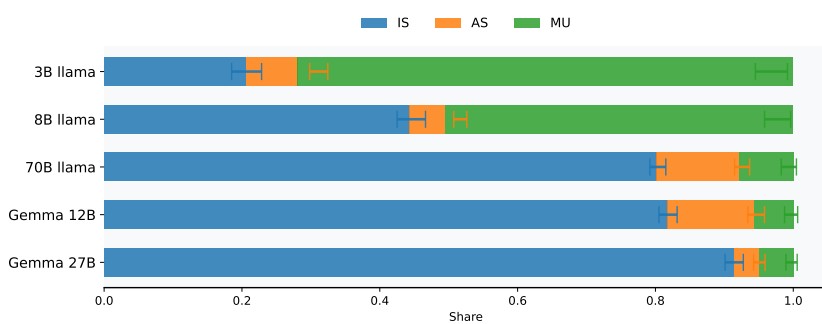

Figure 6: Risk Rating - Information Theoretic Decomposition of Intent Comprehension

Across both loan tasks, the decomposition pinpoints how intent comprehension fails when mapping an applicant profile to an outcome. High MU means the model has not formed a stable applicant→decision rule: even holding the candidate attributes and the exact articulation fixed, it vacillates, consistent with an internally ambiguous or weakly specified credit assessment. High AS, in contrast, directly reflects imperfect intent comprehension: when the candidate is held fixed, paraphrases that should be equivalent nonetheless shift the output, implying the model is not reliably reacting to the applicant profile but is letting unintended, surface-level factors (framing, emphasis, narrative style, wording) enter the decision rule. In other words, the model behaves less like it extracted "what is this applicant's risk?" and more like it is pattern-matching on presentation artifacts.

**Prompt Templates for the discrete case** The following list the exact prompt templates used in the experiment. Placeholders enclosed in braces, such as `{applicant_profile}`, are instantiated programmatically at runtime.

BASE TASK PROMPTS For the `loan_approval` task, the base prompt template used when querying evaluation models is:

```
You are a conservative loan underwriter at a retail bank.
The bank's policy:

Approve loans only if the borrower has a high likelihood of repaying on time.
Reject loans with substantial risk of default.
Use only the information below. Ignore any information not provided.
Applicant profile:
{applicant_profile}

Decision:
Should the bank approve this loan?
```

```
Answer with a single digit:
1 = approve
0 = reject
Do not output anything else.
```

For the `risk_rating` task, the base prompt template is:

```
You are a risk analyst at a bank.
Rate the credit risk of the applicant for the specified loan.

Applicant profile:
{applicant_profile}

Rating:
On a scale from 1 to 5, where
1 = very low risk,
2 = low risk,
3 = moderate risk,
4 = high risk,
5 = very high risk,

answer with a single integer between 1 and 5, and nothing else.
```

INTENT-GENERATION PROMPTS    Intents are generated once per task using an Azure-hosted model
with the following system message:

```
You generate realistic bank loan applicant profiles for credit assessment.
Each profile should include:

Age
Annual income
Existing debt
Employment type and tenure
Credit history (late payments, defaults, etc.)
Loan amount and term
Keep each profile in 5-8 bullet points, simple and plausible.
```

and the following user message:

```
Generate 12 diverse applicant profiles that vary widely in income, debt-to-income rat
```

The JSON returned by this interaction is parsed, and the value of the `"applicant_profile"`
field in each object is used as an intent.

## B    HUMAN EVALUATION

We evaluate the two main components of our pipeline: whether model-generated paraphrases faithfully
preserve the intent of the original prompts, and whether our numerical-value extractor, as discussed
in section 4 and Appendix F, and accurately recovers the quantitative estimates produced by the
model. For both evaluations, we construct a human-assessment dataset by uniformly sampling 200
items for each task. For the intent-preservation evaluation, we uniformly draw pairs of original
prompts and their corresponding paraphrases generated by the model. Each paraphrase is compared
to its originating intent-defining prompt. For the extractor evaluation, we uniformly sample model-
generated answers and compare the extractor's cleaned numerical output to the intended numerical
value expressed in the answer. In both settings, two independent human annotators review each
sampled item, producing binary judgments: a paraphrase is labeled as either intent-preserving or not,
and an extraction is labeled as either correct or incorrect.

Overall, the human evaluations provide strong evidence that both components of the pipeline behave reliably. In the intent-preservation task, both annotators judged the vast majority of paraphrases as faithfully preserving the original intent, with Annotator 1 marking 95% of items as intent-preserving and Annotator 2 marking 91% as such, where in 98.5% of cases, at least one of the annotator indicate the intent was preserved. The annotators agreed on 89% of cases, reflecting consistent judgments, and disagreements were concentrated in borderline paraphrases involving minor stylistic shifts or slight differences in emphasis. Because negative labels were extremely rare, Cohen's $\kappa$ is mechanically low ($\kappa \approx 0.16$) despite the high raw agreement; this effect is well-known in settings with extreme class imbalance and does not indicate substantive disagreement between annotators. Table **??** is the confusing matrix across the two annotators.

| Annotator 2
Annotator 1 | Different | Same |
| --- | --- | --- |
| Different | 3 | 7 |
| Same | 15 | 175 |

Table 1: Confusion Matrix Between the Two Annotators

The evaluation of the numerical-value extractor shows similarly encouraging results. Annotator 1 labeled 94% of extractions as correct, and Annotator 2 labeled 91% as correct, and the two annotators agreed on 93% of items. Disagreements were limited and typically arose when the model provided a range (e.g., "around 5–10") or it's reasoning was cut by the token limit, leading annotators to differ slightly in whether the extractor should be credited for selecting a particular value. Cohen's $\kappa$ is approximately 0.50, interpreted as moderate agreement and consistent with the very high raw agreement. The table below shows the confusion matrix for both annotators.

| Annotator 2
Annotator 1 | Incorrect | Correct |
| --- | --- | --- |
| Incorrect | 8 | 4 |
| Correct | 10 | 178 |

Table 2: Confusion Matrix Between the Two Annotators

Taken together, these results indicate that both the paraphrase-generation and numerical-extraction components of the pipeline perform reliably in practice. The high agreement rates across two independent annotators, supported by chance-corrected measures where appropriate, demonstrate that paraphrases almost always preserve intent and that numerical values are extracted correctly in the vast majority of model outputs.

## C    RELATED LITERATURE

**LLMs Prompt Robustness**   Recent papers have investigated the sensitivity of large language models (LLMs) to prompting using a variety of frameworks and metrics. Zhuo et al. (2024) introduce the ProSA framework, which quantifies prompt sensitivity through a novel metric called PromptSensiScore (PSS). PSS measures the average variation in performance across semantically equivalent prompt variants at the instance level, offering insight into how much a model's output changes with different prompt formulations. Their findings show that prompt sensitivity varies across tasks, models, and prompt types, with particularly high sensitivity observed in reasoning and creative tasks. Similarly, Sclar et al. (2023) examine the role of "spurious features" in prompt formatting—such as punctuation, spacing, and capitalization—and propose FORMATSPREAD, a metric that captures the spread in task accuracy across equivalent prompt formats. FORMATSPREAD is defined as the difference in performance between the best and worst format, and their results reveal that such superficial changes can lead to performance swings of up to 76 accuracy points—effects that are not mitigated by model size or few-shot learning. Brucks and Toubia (2025) take a different perspective, framing prompt sensitivity as a methodological artifact inspired by the concept of choice architecture.

Using full-factorial experiments, they show that prompt order, labeling, framing, and justification systematically bias model responses—and that even instructing the model to ignore these features does not eliminate the effect. Their central claim is that no prompt is neutral. To address this, they recommend aggregating responses across multiple prompts, akin to ensemble methods, to counteract individual prompt biases.

Our behavioral framing complements and extends this literature by offering a diagnostic framework that quantifies prompt sensitivity in terms of structured variance. Unlike prior work that focuses on scalar measures of how model responses change, we take seriously the idea that model outputs should be stochastic—but emphasize that the source of this randomness should not arise from arbitrary variations in prompt phrasing. As we know, no other measure of prompt sensitivity linked the importance of robustness to the world model and focused on the variance aspect of responses.

**Measuring Semantic Robustness and Fairness.** Our decomposition also offers practical diagnostic value. High *articulation sensitivity* suggests that small variations in phrasing disproportionately affect model outputs, raising concerns for *fairness* and *accessibility*. Prior work shows that models may perform differently for users with dialectal, non-standard, or less "typical" phrasing (Bolukbasi et al., 2016; Si et al., 2022; Tan and Celis, 2021; Guo et al., 2024). Our framework allows developers to quantify this fragility and track improvements over time. In this respect, our method serves as a form of semantic reliability auditing, applicable in safety-critical domains such as healthcare, finance, and legal reasoning, where output variability under minor phrasing shifts can lead to unacceptable inconsistency.

**Translation Chains** Back-translation is a data augmentation technique in natural language processing, particularly in neural machine translation ( Sennrich et al. (2016)). Back-translation involves translating monolingual target-language text into the source language using a reverse translation model, and then using the resulting synthetic parallel data to train the forward model. Subsequent work has extended the approach to include round-trip translation and multilingual back-translation, where intermediate languages are introduced to further diversify the training data and improve generalization (Fadaee et al., 2017; Edunov et al., 2018; Xie et al., 2020; Youn et al., 2016). These variants exploit linguistic variation introduced during translation to create richer training distributions, which have been beneficial not only for translation tasks but also for classification, question answering, and style transfer. In our setup, we do not use translation chains as a data augmentation tool, but instead think of them as a way to diversify language while preserving meaning.

## C.1 RELATION TO OTHER WORK ON MEASURING WORLD MODELS

As discussed in Remark 1, our definition of Intent Comprehension, is highly related to world models. Our notion of a world model draws inspiration from research in reinforcement learning (RL) and model-based control, where a world model captures the dynamics of an environment and supports planning. In these contexts, a world model is typically a learned transition function or latent representation that abstracts the environment's relevant state space [Ha and Schmidhuber (2018b); Hafner et al. (2019); Ha and Schmidhuber (2018a)].

Analogously, we treat user intent as a latent variable and evaluate whether a model can infer and represent this variable consistently across diverse inputs. This is conceptually related to state abstraction in RL Li et al. (2006), where different observations map to the same underlying state if they yield equivalent value functions. In our case, different prompts map to the same intent if they elicit the same output distribution (or value-weighted distribution).

Recent work has examined whether generative models develop an internal world model in the context of games. Toshniwal et al. (2022) and Li et al. (2023) helped establish games—such as chess and Othello—as testbeds for evaluating the emergence of world models. A common approach in this literature involves using probes to assess whether a model's internal representations encode latent game states. In contrast, our evaluation metrics are model-agnostic: rather than probing representations or relying on an external notion of state feasibility, we focus on the internal consistency of a model's responses as an indicator of world model quality.

Most closely aligned with our work is Vafa et al. (2024), who test an LLM's world model by asking whether it recovers the deterministic finite automaton (DFA) governing a sequence–generation

task. Leveraging the Myhill–Nerode theorem, they introduce two metrics: (i) sequence compression—do any two prefixes that land in the same DFA state admit the same set of valid continuations? and (ii) sequence distinction—do prefixes that reach different states generate different permissible continuations?

Both their framework and ours impose a common behavioral mandate: (1) inputs sharing a latent condition (DFA state or user intent) must elicit indistinguishable outputs, and (2) inputs differing in that condition must yield measurably different outputs. Crucially, both approaches assess world-model quality based solely on outputs, without inspecting internal parameters.

The divide lies in how "correctness" is anchored. Vafa et al. (2024) use a built-in, binary oracle—the known DFA—to label continuations as right or wrong at the syntax level. Our variance-decomposition framework instead delegates that judgment to an external evaluator $V$, which maps a $\langle$prompt, response$\rangle$ pair to latent intent and task value. $V$ can imitate a hard 0/1 oracle, but it can also apply graded semantic scores. Importantly, our test goes further: when $V$ deems two prompts equivalent, we require the entire distribution of (possibly stochastic) responses to match across those prompts. If $V$ itself enforces strict correctness, this collapses to zero model uncertainty—any variability in outputs must stem from differing intents. Thus, while both methods rely on an evaluator, Keyon's oracle is intrinsic and binary, whereas ours is external, tunable, and distribution-aware.

Finally, our notion of a IC demands distributional equality across equivalent prompts, whereas the DFA test assumes deterministic continuations. Extending Myhill–Nerode to stochastic automata would miss the point we target: open-ended tasks with no unique correct answer. In such settings, the likelihoods assigned to alternative continuations encode the model's assumptions. A genuine world model, therefore, aligns those likelihoods whenever the underlying intent is the same, preserving semantic sufficiency even under uncertainty.

## D ESTIMATING VARIANCE DECOMPOSITION VIA ANOVA

Our goal is to decompose variability in numeric responses into three components: IS, AS, and MU. A natural starting point is the variance decomposition

$$\mathrm{Var}(Y) = \mathrm{Var}\big(\mathbb{E}[Y \mid i, p]\big) + \mathbb{E}\big[\mathrm{Var}(Y \mid i, p)\big],$$

where $(i, p)$ indexes task and prompt. A naive sample-analogue implementation replaces the conditional means $\mathbb{E}[Y \mid i, p]$ with cell-wise sample means and the conditional variances with cell-wise sample variances, and then computes the variance of these estimated means across tasks and prompts. However, this plug-in approach can induce finite-sample bias when cells are sparse or unbalanced.

To see the problem, let $\mu_{ip} = \mathbb{E}[Y \mid i, p]$ and let $\hat{\mu}_{ip}$ denote the sample mean in cell $(i, p)$ based on $n_{ip}$ draws. Then

$$\mathrm{Var}(\hat{\mu}_{ip}) = \mathrm{Var}(\mu_{ip}) + \mathbb{E}[\mathrm{Var}(\hat{\mu}_{ip} \mid \mu_{ip})] \approx \mathrm{Var}(\mu_{ip}) + \frac{\sigma_R^2}{n_{ip}},$$

so the variance of *estimated* cell means mechanically includes sampling noise on the order of $\sigma_R^2/n_{ip}$. When many cells have small $n_{ip}$, this additional term inflates the apparent between-cell variation and therefore biases upward the estimated contributions of IS and AS. Unbalanced designs exacerbate the issue: cells with very small $n_{ip}$ have highly variable $\hat{\mu}_{ip}$ but still enter equally in a simple variance-of-means calculation. In contrast, the within-cell component can be biased downward if the between-cell inflation is not correctly accounted for. In short, naive variance decomposition treats noisy cell means as if they were observed without error.

As the number of samples per cell grows, the sampling variance of $\hat{\mu}_{ip}$ shrinks and the naive decomposition approaches the population decomposition. In the limit $n_{ip} \to \infty$ for all $(i, p)$, the plug-in estimator is consistent. Practically, however, we work with finite and often heterogeneous $n_{ip}$. Precision improves with (i) the number of draws per task–prompt cell, and (ii) the diversity of paraphrases per task, which helps distinguish task-level and prompt-level variability rather than confounding them with noise. As a rough rule of thumb, naive cell-wise decompositions are reasonably stable only when most $(i, p)$ cells have at least 20–30 responses and very few cells have fewer than about 5; below this range, between-cell variance estimates become highly sensitive to sampling noise, and partial pooling is strongly preferable.

To address these issues directly, we use a hierarchical (mixed-effects) specification that *borrows strength* across related cells via *partial pooling*: group means are shrunk toward a common mean in proportion to their sampling variance and group size. This stabilizes variance components, especially when some tasks or prompt variations have few observations.

Index tasks by $i = 1, \ldots, I$, prompt variations (nested within tasks) by $p = 1, \ldots, P_i$, and individual observations within a prompt by $k$. We model the standardized response $y_{ipk}$ as

$$y_{ipk} = \mu + u_i + v_{ip} + \varepsilon_{ipk}, \qquad u_i \sim \mathcal{N}(0, \sigma_I^2), \ \ v_{ip} \sim \mathcal{N}(0, \sigma_A^2), \ \ \varepsilon_{ipk} \sim \mathcal{N}(0, \sigma_R^2), \ \ (1)$$

with $v_{ip}$ nested in task $i$. Here $\mu$ is a global intercept, $u_i$ captures intent-level variation, IS, $v_{ip}$ captures prompt-level heterogeneity within intent, AS, and $\varepsilon_{ipk}$ is idiosyncratic noise, MU). Partial pooling arises because the random effects $(u_i, v_{ip})$ are estimated jointly: noisy cells with small $n_{ip}$ are shrunk more aggressively toward the global mean, preventing them from exerting disproportionate influence on the between-cell variance.

Let $\sigma_T^2 = \sigma_I^2 + \sigma_A^2 + \sigma_R^2$ denote the total variance in model responses. We report variance *shares*

$$\text{IS} = \frac{\sigma_I^2}{\sigma_T^2}, \qquad \text{AS} = \frac{\sigma_A^2}{\sigma_T^2}, \qquad \text{MU} = \frac{\sigma_R^2}{\sigma_T^2},$$

which sum to 1 and are invariant to scale. If we standardize $y$ before fitting, these coincide with absolute contributions on the standardized scale. We estimate $(\mu, \sigma_I^2, \sigma_A^2, \sigma_R^2)$ by *Restricted Maximum Likelihood* (REML), which reduces the small-sample bias of variance component MLEs. Practically, we fit (1) via `lme4::lmer` (through `pymer4`). Finally, we report uncertainty for the *shares* using a bootstrap that respects the nesting: we resample at the task level, then within tasks at the prompt level, and then within prompts at the response level.

Throughout our estimation we report bootstrap standard errors for all figures, from 100 repetitions.

# E  INCORPORATING CORRECTNESS INTO THE FRAMEWORK

Our main experiments focus on numeric responses, where the valuation function $V$ maps a model output $a$ (possibly together with the prompt $p$) to a scalar value $Y = V(a, p)$ (e.g., a probability, price, or duration). When ground-truth labels are available, the same framework can enhance the standard analysis of *correctness*. This appendix explains how and also clarifies what is lost when we collapse multiple plausible answers into a single correctness label.

## E.1  BINARY CORRECTNESS

Suppose we have a notion of correctness at the level of the evaluator. For a given prompt $p$ and response $a$, let

$$V_{\text{corr}}(a, p) \in \{0, 1\}$$

indicate whether the response is correct (1) or incorrect (0) according to ground truth or human judgment. We can then apply the variance decomposition (or the decomposition suggested in appendix **??** for discrete values) to the random variable

$$Y_{\text{corr}} = V_{\text{corr}}(A, P),$$

where $A$ is the model's sampled answer and $P$ ranges over prompts in our design.

All definitions carry over directly:

- $\text{IS}_{\text{corr}}$ measures how much of the variation in *correctness* is explained by changes in intent, holding articulation fixed.
- $\text{AS}_{\text{corr}}$ measures how much variation in *correctness* is explained by changes in articulation, holding intent fixed.
- $\text{MU}_{\text{corr}}$ captures residual variation in *correctness* not explained by either.

In degenerate cases where a model is always correct (or always wrong) on a given task, $Y_{\text{corr}}$ is constant and there is no variation to explain. This is appropriate: when correctness is deterministic,

there is no remaining uncertainty about where errors come from and we cna't distinc between cases where the model is simply matching patterns to cases where the model reacting to the underlying user intent. The decomposition is most informative in the realistic regime where correctness lies strictly between $0$ and $1$, and we wish to understand whether failures are driven by intent, articulation, or residual noise.

In this sense, using correctness as the valuation $V$ does not replace our original analysis on raw numerical answers, but complements it. When we apply the decomposition to raw answers, we study *how the model's responses change with intent and articulation.* When we apply it to correctness indicators, we instead study *how the probability of being correct changes with intent and articulation,* that is, whether errors are concentrated on certain intents, certain phrasings, or are largely unexplained noise.

### E.2    MULTIPLE ACCEPTABLE ANSWERS AND PARTIAL CREDIT

In many settings there is not a single uniquely correct answer, but instead a set of *acceptable* answers $\mathcal{A}_{\mathrm{acc}}(p)$ for a prompt $p$ (e.g., a range of reasonable numerical estimates, or several semantically equivalent textual completions). Our framework can accommodate this in several ways.

A simple approach is to define a binary acceptability indicator

$$V_{\mathrm{acc}}(a, p) = \mathbf{1}\{a \in \mathcal{A}_{\mathrm{acc}}(p)\},$$

and apply the same decomposition to

$$Y_{\mathrm{acc}} = V_{\mathrm{acc}}(A, P).$$

The resulting $\mathrm{IS}_{\mathrm{acc}}$, $\mathrm{AS}_{\mathrm{acc}}$, and $\mathrm{MU}_{\mathrm{acc}}$ then describe how the *probability of producing an acceptable answer* varies with intent and articulation.

When numeric magnitude also matters, we can combine acceptability and value. For example, in a guesstimation task we might consider

$$V_{\mathrm{hyb}}(a, p) \;=\; V(a) \cdot \mathbf{1}\{a \in \mathcal{A}_{\mathrm{acc}}(p)\},$$

where $V(a)$ is the relvent numeric value in the response. so that only acceptable answers contribute their numeric value, while unacceptable answers are mapped to zero. More generally, one can use a *graded scoring function*

$$V_{\mathrm{score}}(a, p) \in [0, 1],$$

which assigns partial credit based on distance to the correct value or semantic similarity to a reference answer. In all these cases, our variance decomposition applies unchanged to the induced random variable $Y = V(A, P)$.

In cases where there are multiple plausible answers, simply encoding correctness as a binary indicator may miss important structure. Collapsing an entire set $\mathcal{A}_{\mathrm{acc}}(p)$ into the value $1$ and everything else into $0$ discards information about *how* the model's responses are distributed within the acceptable set and how they deviate when they are not acceptable. For example, two models might have the same overall correctness rate, but one produces a tight distribution around a canonical answer while the other oscillates between several qualitatively different, yet still acceptable, solutions. A binary $V_{\mathrm{acc}}$ does not distinguish these behaviors.

Our framework is flexible to this choice. If the evaluation goal is to separate *successes from failures*, correctness-based encodings such as $V_{\mathrm{corr}}$ or $V_{\mathrm{acc}}$ are appropriate, and the decomposition reveals whether variability in success is driven by intent, articulation, or model uncertainty. If, instead, the goal is to understand how the model navigates a space of multiple plausible answers, one can retain a richer $V$ (e.g., a continuous score or a calibrated utility) and apply the decomposition to that quantity rather than to a binary indicator.

Thus, accuracy-based evaluation and our intent-comprehension metrics are not competing notions. Accuracy tells us *how often* the model is right; our framework, applied either to raw answers or to correctness-encoded values, helps explain *why* it succeeds or fails, by attributing variation to intent, articulation, and residual uncertainty, while leaving room to capture the structure of multiple plausible answers when that structure is evaluatively relevant.

## F  EXPERIMENTAL DETAILS

In this section, we describe how we construct our experiments. We first describe how we construct our set of tasks and intents, and then discuss how we construct an intent-equivariant set of prompts.

**Constructing the set of tasks** To construct our evaluation metric, we focus on generating open-ended, guesstimation-type questions, using a two-stage LLM workflow. We focus on these types of questions as our IC metric for two reasons. First, our measure is defined over response distributions. To identify whether variation is driven by intent (IS) rather than articulation (AS) or model uncertainty (MU), we need tasks that produce genuine dispersion in plausible answers[3]. Open-ended questions, with various sets of answers, also match real usage, where users ask for estimates, recommendations, and contextual judgments; Together, this yields a naturalistic yet controlled setting in which IS, AS, and MU are identifiable and informative.

We focus on 5 areas of interest: Transportation, Personal Finance, health and nutrition, logistics, and social planning. and use an automatic procedure to generate 24 questions for each topic. To construct our dataset of guesstimation-style prompts, we implemented an automated pipeline that leverages large language models guided by structured templates and semantic constraints. Each generated question contains exactly one {placeholder} token, which is later substituted with concrete values. The system prompt enforces that questions are self-contained, specify explicit reporting units (e.g., "dollars per year," "kWh per month"), and use the placeholder to denote a general semantic role such as a material, process, or population subgroup. To ensure coverage across domains, we draw from a library of question templates spanning categories such as counts, rates, intensities, costs, and probabilities (see Appendix I.1). For each placeholder, a second prompt generates a set of plausible replacements, or intents, which are short noun phrases consistent with the semantic role and unit of the question. This two-step process—first generating unit-specified question structures and then populating them with realistic values—produces a diverse set of open-ended, quantitative guesstimation questions (see Appendix J for the exact wording of the prompts).

**Creating a Set of Intent-Equivalent Prompts** Given the set of categories and tasks, our next challenge is to generate a collection of prompts that express the *same intent* but differ in surface form and semantics. This is a non-trivial challenge. Even for humans, writing multiple prompts that convey exactly the same intent without introducing subtle shifts in meaning is difficult. Language models are highly sensitive to linguistic cues, and small differences in phrasing can unintentionally signal different goals or assumptions. This makes it essential to generate paraphrases that are semantically faithful to the original prompt but lexically and syntactically distinct.

To tackle this, we adopt a two-stage approach. Our primary method relies on *cross-lingual translation*, a process inherently designed to preserve meaning while altering surface expression (e.g., Youn et al. (2016)). Translation captures the core idea of transferring intent across languages while abstracting away from specific phrasings. LLMs have demonstrated impressive performance in this task, often on par with human translators in consistency and fluency (Karpinska and Iyyer (2023); Yan et al. (2024a;b)). We leverage this ability by using GPT-4.1 to perform the translations: starting with an original English prompt, we translate it sequentially through two randomly selected intermediate languages and then back into English. The result is a paraphrase that retains the original intent but exhibits different syntactic and semantic features due to translation-induced variation. Different languages vary in how they structure, resulting in varying cross-translations as we vary the source languages into and from which we translate the prompts (Lewis et al. (2023)).

To maximize semantic divergence while preserving intent, we randomly sample intermediate languages from a diverse set.[4] Each translation chain is required to include at least one of Chinese, Japanese, or Arabic, given their significant structural and lexical distance from English (Chiswick and Miller (2005); Lewis et al. (2023)), which helps introduce greater variation in the back-translated output.

To ensure the resulting prompts still reflect the original intent, we also use GPT-4.1 as an additional check to verify that the intent remains unchanged. Given the original and translated prompt, the

---

[3]Single-answer tasks cam make the IC estimator ill-conditioned if the model returns just a single response, because there is no variance to decompose; the shares (and ratios like MVS, ICI) become numerically unstable or undefined, and the standardized scale becomes ill-posed.

[4]Chinese, Japanese, Arabic, Korean, Portuguese, Spanish, German, Russian, Italian, French, Hindi

model assesses whether they express the same underlying request. Only those pairs judged equivalent are retained. To further enrich the diversity of the prompt set, we generate 500 paraphrases for each original question, embed all prompts using sentence embeddings, and apply a greedy selection algorithm to identify the 10 most diverse prompts—those with the highest mutual semantic distance. This final step ensures that our prompt set not only shares intent but also spans a wide semantic space, enabling robust evaluation of the model's Articulation Sensitivity.

After generating intent-equivalent prompts, we present them to the language model under evaluation. Each prompt is paired with three input values that vary the prompt's main intent — for example, income level in a tax question. Then, for each of the 24 prompts, at each of the 5 topics, and for each of the 12 values, we prompt the candidate LLM and elicit 25 model responses to capture the within-prompt variation.

As discussed in the definition of a sufficient IC, we need to determine how to evaluate the models' responses. We focus on extracting a clear bottom-line value: if the model provides a range, we take the average. Responses without a definitive answer are discarded, and we continue generating responses until we obtain the desired 25 responses. All outputs are generated with a temperature of 1 to accurately reflect the model's response distribution [5]. To extract a usable numeric value from each response, we employ GPT-4.1-mini as a post-processor that identifies and retrieves the relevant quantity from the model's output. In order to assure stability of the ANOVA estimation, for each model–question pair, we winsorize the outcome by removing responses whose numeric estimate lies more than 3 standard deviations from the mean response for that intent, across all paraphrase.

Finally, in our analysis, we evaluate five models: Meta's LLaMA 3.2 3B Instruct,3.1 8B Instruct and LLaMA 3.3 70B Instruct, as well as Google's Gemma 3 12B-IT and 27B-IT. We use the OpenRouter API to generate multiple responses from each model.

## G   ADDITIONAL FIGURES

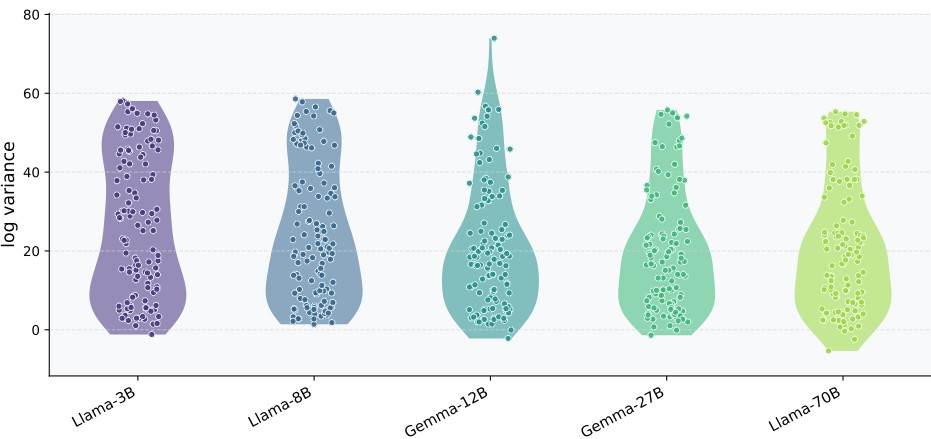

Figure 7: Log variance distribution across the 100 tasks, by model.

---

[5]in the Appendix H we show the effect of temperature on the results

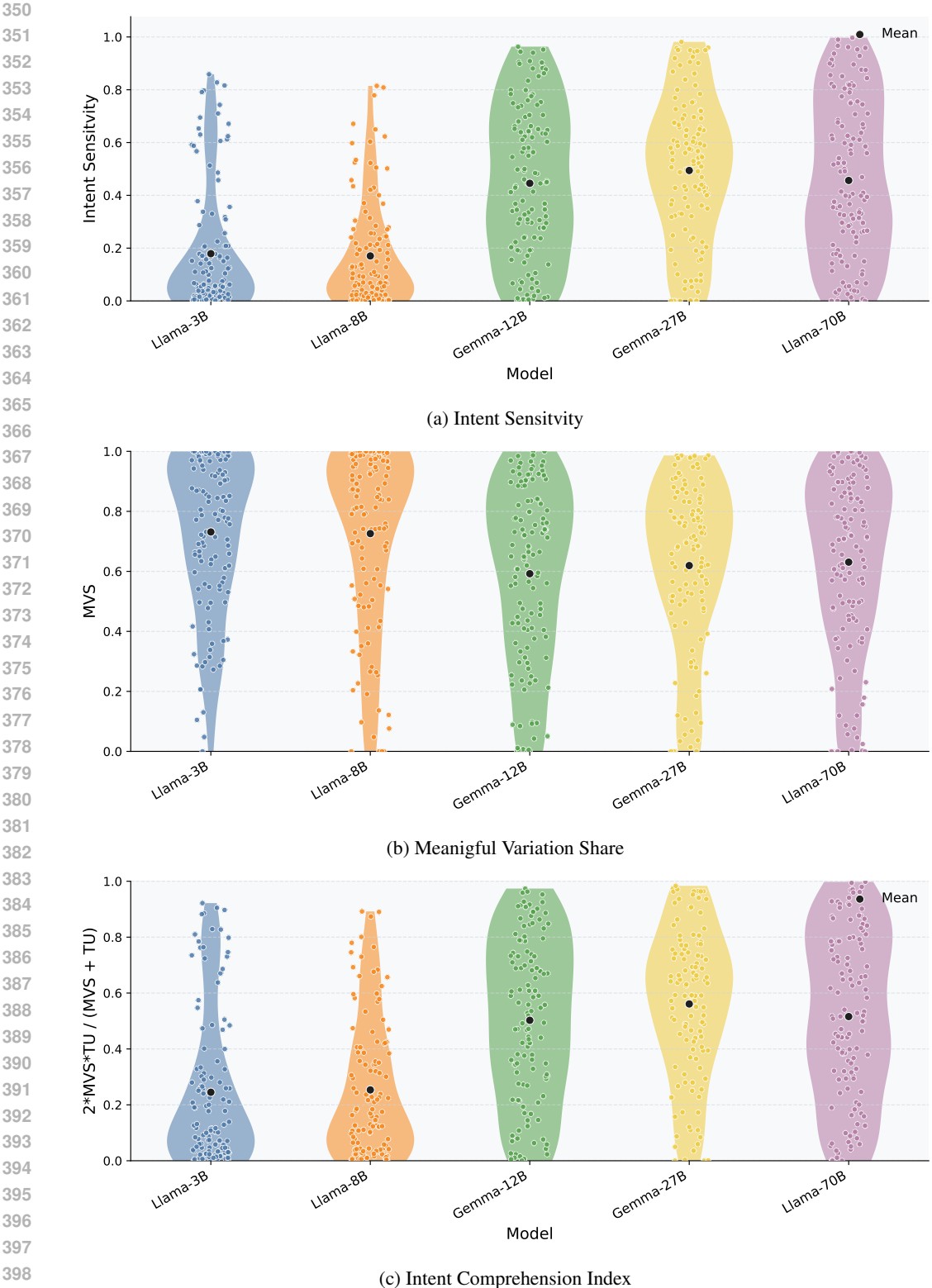

(a) Intent Sensitvity

(b) Meanigful Variation Share

(c) Intent Comprehension Index

Figure 8: Distribution of components across tasks

# H  TEMPERATURE

The main results report model performance under the natural temperature of 1. Figure 2 illustrates this with a spider graph estimated on 5 questions for each of the 5 topics. To explore the effect of sampling temperature, Figures 9 and 11 compare the same tasks estimated with temperature 0.2 versus temperature 1. Reducing the temperature decreases model uncertainty across all models, with the effect particularly pronounced for the smaller models. the figure further suggests that, within this set, the higher-parameter LLaMA and Gemma models are more sensitive to surface text, whereas the smaller LLaMA models appear relatively more responsive to meaningful changes in intent. Figure 10 presents the corresponding results for temperature 0.5[6]. As expected, its behavior lies between the two other temperature settings.

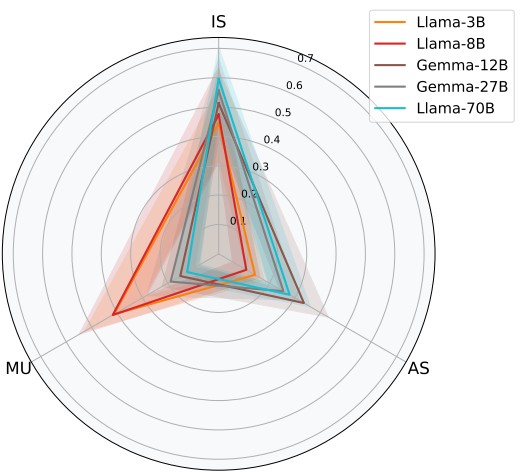

Figure 9: Variance decomposition estimated with temperature 0.2

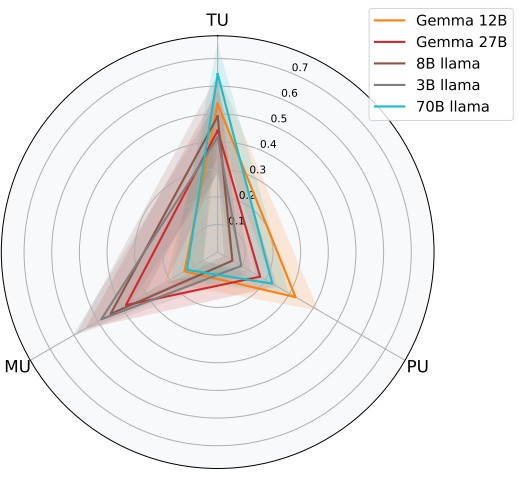

Figure 10: Variance decomposition estimated with temperature 0.5

---

[6]The Gemma 27B model was run at a temperature of 0.55 due to instability at 0.5 on OpenRouter.

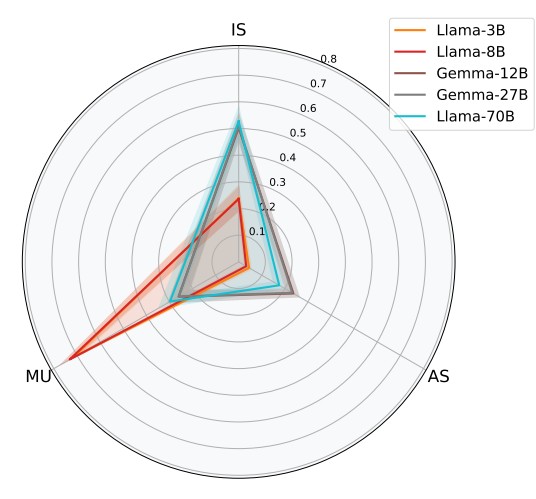

Figure 11: Variance decomposition estimated with temperature 1.

# I  PROMPTS

## I.1  EXAMPLE QUESTIONS

### EXAMPLE QUESTION TEMPLATES

#### PREVALENCE / COUNTS

1. How many {placeholder} are there in a typical city?
2. Approximately how many {placeholder} are produced each farm?
3. What is the total number of {placeholder} registered voters?
4. Roughly how many {placeholder} exist worldwide?
5. How many new {placeholder} were added in 2022?

#### AVERAGES / MEANS

1. What is the average {placeholder} per high school student?
2. What is the mean {placeholder} recorded each year?
3. On average, how much {placeholder} does an individual consume in a day?
4. What is the typical {placeholder} per unit of output?
5. What is the long-run average {placeholder} for a US county?

#### RATES / SHARES / PERCENTAGES

1. By what percentage did {placeholder} change between 2010 and 2020?
2. What fraction of total {scope} is accounted for by {placeholder}?
3. What is the annual growth rate of {placeholder}?
4. What proportion of US population owns at least one {placeholder}?
5. What share of all household expenditures goes to {placeholder}?

#### TOTALS / AGGREGATES

1. Estimate the total {placeholder} required over 5 years horizon.
2. What is the cumulative {placeholder} expected over the next 5 years?
3. What is the projected lifetime total of {placeholder} for a typical student?
4. What is the overall stock of {placeholder} currently in use?
5. How much {placeholder} will be needed to cover national healthcare system for the next decade?

#### INTENSITIES / PER-UNIT METRICS

1. What is the {placeholder} per unit of output?
2. How much {placeholder} is required per kilometre travelled?
3. What is the {placeholder} per capita in Japan?
4. What is the carbon intensity measured as {placeholder} per kWh?
5. What is the average {placeholder} per square metre?

#### MAXIMA / MINIMA / RECORDS

1. What is the maximum recorded {placeholder} in a single year?
2. What is the historical peak {placeholder} observed since 2020?
3. What is the lowest recorded {placeholder}?
4. What is the record-high {placeholder} achieved by a single entity?
5. What is the upper bound of {placeholder} under current regulations?

COSTS / MONETARY

1. What is the average cost of {placeholder} per squre feet?

2. What is the expected budget share spent on {placeholder} each decade?

3. What is the median price paid for {placeholder} in the US?

4. How much investment is required for one unit of {placeholder}?

5. What is the total expenditure on {placeholder} in 2015?

DURATIONS / TIMES

1. What is the average time needed to complete {placeholder}?

2. How long does it typically take for {placeholder} to reach completion?

3. What is the mean lifetime of a {placeholder}?

4. What is the expected waiting time until {placeholder} occurs?

5. What is the typical duration of {placeholder} in the manufacturing sector?

DENSITIES / CONCENTRATIONS

1. What is the density of {placeholder} per square kilometre?

2. What is the concentration of {placeholder} per litre in the sample?

3. What is the average number of {placeholder} per household?

4. What is the typical {placeholder} per lane-kilometre of road?

5. What is the median {placeholder} per employee in the sector?

PROBABILISTIC / RISK

1. What is the probability that {placeholder} occurs within a given year?

2. What is the expected frequency of {placeholder} per decade?

3. What is the chance of observing at least one {placeholder} in a week?

4. What is the return probability to {placeholder} after 5 years?

5. What is the expected failure rate expressed as {placeholder} per 1,000 units?

RESOURCE / INPUT COEFFICIENTS

1. How much {placeholder} is consumed per tonne of output?

2. What is the marginal {placeholder} required for one additional unit?

3. What is the input output coefficient of {placeholder} to gross output?

4. What is the elasticity of {placeholder} with respect to price?

5. What is the shadow cost of one unit of {placeholder}?

VOLUME & CAPACITY

1. How many {placeholder} would it take to fill a standard school bus?

2. What is the total volume of {placeholder} that flows over Niagara Falls in a single day?

3. If all the {placeholder} consumed in the United States in a year were put in a single container, how large would it be?

4. What is the total annual production of {placeholder} in France, in liters?

5. How many bathtubs could you fill with the amount of {placeholder} consumed globally each day?

WEIGHT & MASS

1. What is the total weight of all the {placeholder} on Earth?

2. Estimate the total mass of all {placeholder} currently in the Netherlands.

3. What is the weight of all the {placeholder} produced by New York City each week?

4. What is the total weight of all the {placeholder} in the state of Texas?

5. Estimate the total mass of all {placeholder} currently airborne over the United States.

LENGTH, DISTANCE & AREA

1. What is the total length, in miles, of all the {placeholder} in Germany?

2. If you laid every {placeholder} eaten in America on July 4th end-to-end, how far would the line stretch?

3. What is the total surface area of all the {placeholder} in China?

4. How many times does an average {placeholder} rotate during its operational lifetime?

5. Estimate the total length of all the {placeholder} sold in North America each holiday season.

FINANCIAL & ECONOMIC

1. How much money, in loose change, is currently in all the {placeholder} in the United States?

2. What is the total cost to fuel all the {placeholder} in California for one day?

3. What is the total annual revenue of all the {placeholder} operating in the United States?

4. How much money is spent on {placeholder} in Canada annually?

5. What is the total market value of all items listed as {placeholder} on eBay worldwide?

TIME & DURATION

1. How many total hours do all people in the United States spend doing {placeholder} each year?

2. How long would it take one person to watch every {placeholder} on YouTube?

3. On average, how many times a day does a person in Japan check their {placeholder}?

4. Estimate the total person-years spent on {placeholder} globally each day.

5. What is the average wait time for a {placeholder} in London during peak hours?

RATES & FREQUENCY

1. Estimate the total number of {placeholder} sent in India every day.

2. How many {placeholder} are sold in the United Kingdom each year?

3. How many {placeholder} are fixed in Chicago each year?

4. What is the consumption rate of {placeholder}, in units per second, in the United States on a Friday night?

5. How many {placeholder} are uploaded to Instagram worldwide every minute?

POPULATION & PROFESSION

1. How many {placeholder} are there in the state of Illinois?

2. Estimate the number of {placeholder} in Brazil.

3. What is the total number of {placeholder} on all the people in Japan?

4. Estimate the total number of people currently airborne in {placeholder} around the world.

5. How many {placeholder} work in Paris?

EVERYDAY OBJECTS & CONSUMPTION

1. How many pairs of {placeholder} does the average person in America own in their lifetime?

2. Estimate the total number of {placeholder} used by all babies in the United States in one year.

3. What is the total amount of {placeholder} consumed in the United States annually?

4. How many gallons of {placeholder} are used by the U.S. newspaper industry annually?

5. How many words does the average person read per day on their {placeholder}?

INFRASTRUCTURE & URBAN

1. What is the total number of {placeholder} in all the buildings of downtown Manhattan?

2. How many {placeholder} are there in the entire city of Tokyo?

3. Estimate the total number of {placeholder} in New York's Central Park?

4. What is the total number of {placeholder} in the Empire State Building?

5. Estimate the total annual electricity consumption of all the {placeholder} in the world.

CREATIVE & ABSTRACT

1. How many individual {placeholder} are on a professional soccer field?

2. How many {placeholder} could you fit in Grand Central Terminal's main concourse?

3. What is the total number of {placeholder} manufactured globally in a single day?

4. What is the total population of {placeholder} in Venice?

5. How many {placeholder} would it take to build a chain to the Moon?

PROBABILISTIC & ODDS

1. What is the probability that a randomly selected person in the United States has {placeholder}?

2. What are the odds of a flight being delayed at {placeholder} International Airport?

3. What is the daily probability that the {placeholder} in a major city experiences a major outage?

4. What is the chance that a new {placeholder} in the United States fails within its first year?

5. If you pick a random word from the New York Times, what is the probability it is the word '{placeholder}'?

6. What is the likelihood of experiencing {placeholder} in London on a given day in July?

7. What is the probability that a car driving one mile on a US highway will get a {placeholder}?

8. Estimate the chance that a randomly selected email in an average inbox is {placeholder}.

9. What is the annual probability of a {placeholder} causing significant damage in California?

10. What are the odds that a randomly chosen {placeholder} from a large supermarket is expired?

## J PROMPT SPECIFICATIONS AND DATA COLLECTION PROCEDURE

This section describes all prompts and control settings used in data collection.

### J.1 SYSTEM PROMPT USED FOR ANSWER GENERATION

For every question posed to a model (the *estimation task*), we attach the following `system` message and then a `user` message containing the question text:

```
You are a helpful assistant designed to answer users
questions that involve estimating real-world quantities.
When asked for a numerical value (e.g., average, frequency,
duration, or count), always provide your best-guess
estimate, even if you lack exact data.  Avoid generic
refusals like "I don't have that information."  If needed,
```

```
rely on general knowledge, plausible assumptions, or known
ranges from similar contexts.  The goal is to be useful by
offering a grounded and reasoned numerical estimate.  Return
a single numeric value in your response, followed by the
appropriate unit of measurement (e.g., 3 days, 3 kg, 10 mg).
```

## J.2    BACK–TRANSLATION PROMPTING AND DIVERSITY PROCEDURE

We generate alternative wordings of a seed question via back–translation. Each back–translation hop uses the same `system` translator instruction:

```
You are a professional translator.  Translate the user text
to {target_lang}, preserving {placeholder}.
Preserve meaning exactly.  Do not add, remove facts and
information.
Keep sentence boundaries and speaker perspective the same.
Return *only* the translation--no commentary.
```

The corresponding `user` content is the current question snippet (potentially containing a literal {placeholder} token). We first translate English $\rightarrow$ intermediate language(s), then back to English using the same instruction with {target_lang} = English. Placeholders enclosed in curly braces are preserved verbatim through all hops.

**Language chain.**    We use a chain of $H$ languages (default $H = 2$) selected to encourage script and typological variation. One of {Chinese, Japanese, Arabic} is always included; the remaining hop(s) are sampled from: { Chinese, Japanese, Arabic, Korean, Portuguese, Spanish, German, Russian, Italian, French, Hindi }. (Exact draws are randomized per back–translation.)

EQUIVALENCE FILTER.

```
system:  You are an expert at judging whether two English
sentences mean exactly the same thing.
user:  Do these two sentences convey the same meaning and
intent?
Sentence A: "..."
Sentence B: "..."
Respond only with "Yes" or "No".
```

When a concrete value for {placeholder} is available, the comparison is done after substituting that value in both sentences (the raw candidate must still contain the literal {placeholder} token).

SPEAKER/PERSPECTIVE FILTER.

```
system:  You are an expert in grammar and meaning.  Your
job is to assess whether two English sentences use the same
grammatical subject or speaker perspective.  That includes
whether both use "I", "we", "you", passive voice, or the
same named subject (e.g., "the city", "Jiji").
user:  Do these two sentences use the same speaker or
subject perspective?
Sentence A: "..."
Sentence B: "..."
Respond only with "Yes" or "No".  If the speaker changes or
a name is added or removed, respond "No".
```

### J.2.1 LIST OF QUESTION PROMPTS AND EXTRACTIONS

### HEALTH & NUTRITION

**Question:** On average, what is the energy expenditure from {placeholder} for an adult during one hour, in kilocalories per hour?
**Values:** ["walking", "running", "cycling", "swimming", "yoga", "dancing", "gardening", "standing", "typing", "reading", "cleaning", "driving"], You extract a single numeric value from an answer string. Return only the number and ensure it matches the requested unit.
**Extraction Prompt:** Extract a single numeric value representing kilocalories per hour. Return "None" if there is no numerical value.

Answer text: {answer_text}

---

**Question:** Estimate the average amount of dietary fiber contained in one serving of {placeholder}, in grams per serving.
**Values:** ["black beans", "oatmeal", "whole wheat bread", "broccoli", "chia seeds", "raspberries", "almonds", "sweet potato", "green peas", "avocado", "brown rice", "carrots"], You extract a single numeric value from an answer string. Return only the number and ensure it matches the requested unit.
**Extraction Prompt:** Extract a single numeric value representing grams per serving. Return "None" if there is no numerical value.

Answer text: {answer_text}

---

**Question:** Estimate the average number of times a typical household uses a {placeholder} per week, in uses per week.
**Values:** ["microwave", "dishwasher", "vacuum cleaner", "washing machine", "television", "oven", "toaster", "coffee maker", "blender", "refrigerator door", "air conditioner", "hair dryer"], You extract a single numeric value from an answer string. Return only the number and ensure it matches the requested unit.
**Extraction Prompt:** Extract a single numeric value representing the number of uses per week; unit: 'uses per week'. Return "None" if there is no numerical value.

Answer text: {answer_text}

---

**Question:** Estimate the average annual revenue generated by {placeholder} in the health and nutrition industry in the United States, in dollars per year.
**Values:** ["dietary supplements", "organic food products", "vitamin sales", "weight loss programs", "nutrition consulting services", "sports nutrition drinks", "meal replacement shakes", "functional foods", "natural health stores", "health coaching services", "nutritional app subscriptions", "protein powder brands"], You extract a single numeric value from an answer string. Return only the number and ensure it matches the requested unit.
**Extraction Prompt:** Extract a single numeric value reported in dollars per year. Return "None" if there is no numerical value.

Answer text: {answer_text}

---

**Question:** Estimate the average body mass index (BMI) of {placeholder} in the United States, in kilograms per square meter (kg/m¬).
**Values:** ["adults", "children", "teenagers", "elderly men", "pregnant women", "college students", "African Americans", "Asian Americans", "male athletes", "female nurses", "Hispanic women", "middle-aged adults"], You extract a single numeric value from an answer string. Return only the number and ensure it matches the requested unit.
**Extraction Prompt:** Extract a single numeric value and ensure the unit is kilograms per square meter (kg/m¬). Return "None" if there is no numerical value.

Answer text: {answer_text}

**Question:** Estimate the average shelf life of {placeholder} in a typical American grocery store, in days per item.
**Values:** ["milk", "eggs", "lettuce", "yogurt", "chicken breast", "ground beef", "bread loaf", "apples", "bananas", "cheese block", "carrots", "spinach"], You extract a single numeric value from an answer string. Return only the number and ensure it matches the requested unit.
**Extraction Prompt:** Extract a single numeric value representing average shelf life; unit must be 'days per item'. Return "None" if there is no numerical value.

Answer text: {answer_text}

---

**Question:** Estimate the average amount of vitamin C contained in one medium-sized {placeholder}, in milligrams per item.
**Values:** ["orange", "kiwi", "strawberry", "broccoli floret", "bell pepper", "tomato", "grapefruit", "mango", "papaya", "brussels sprout", "cabbage leaf", "pineapple slice"], You extract a single numeric value from an answer string. Return only the number and ensure it matches the requested unit.
**Extraction Prompt:** Extract a single numeric value and report it in milligrams per item. Return "None" if there is no numerical value.

Answer text: {answer_text}

---

**Question:** Estimate the average annual production of {placeholder} in the United States, in metric tons per year.
**Values:** ["corn", "soybeans", "wheat", "cotton", "rice", "sugar beets", "potatoes", "tomatoes", "coal", "steel", "aluminum", "cement"], You extract a single numeric value from an answer string. Return only the number and ensure it matches the requested unit.
**Extraction Prompt:** Extract a single numeric value representing annual production. Only report in 'metric tons per year'. Return "None" if there is no numerical value.

Answer text: {answer_text}

---

**Question:** Estimate the average maintenance cost of a {placeholder} used in a typical American hospital, in dollars per device per year.
**Values:** ["MRI machine", "X-ray machine", "ultrasound scanner", "ventilator", "defibrillator", "ECG monitor", "infusion pump", "anesthesia machine", "patient monitor", "CT scanner", "sterilizer", "dialysis machine"], You extract a single numeric value from an answer string. Return only the number and ensure it matches the requested unit.
**Extraction Prompt:** Extract a single numeric value and report it using the unit 'dollars per device per year'. Return "None" if there is no numerical value.

Answer text: {answer_text}

---

**Question:** Estimate the average failure rate for {placeholder} in clinical nutrition settings, in percent (%) per year.
**Values:** ["infusion pumps", "enteral feeding tubes", "peripheral IV catheters", "central venous catheters", "parenteral nutrition bags", "glucometers", "electronic medication carts", "feeding pumps", "blood glucose monitors", "nutrition software systems", "IV fluid warmers", "nutritional supplement dispensers"], You extract a single numeric value from an answer string. Return only the number and ensure it matches the requested unit.
**Extraction Prompt:** Extract a single numeric value and report it in percent (%) per year. Return "None" if there is no numerical value.

Answer text: {answer_text}

---

**Question:** Estimate the average annual revenue generated by {placeholder} per grocery store in the United States, in dollars per year.

**Values:** ["fresh produce sales", "dairy products", "bakery items", "meat department", "seafood sales", "beverage section", "frozen foods", "prepared meals", "snack foods", "organic products", "household goods", "health and beauty aids"], You extract a single numeric value from an answer string. Return only the number and ensure it matches the requested unit.
**Extraction Prompt:** Extract a single numeric value expressed in 'dollars per year'. Return "None" if there is no numerical value.

Answer text: {answer_text}

---

**Question:** Estimate the average annual treatment cost, in dollars per year, for a patient with {placeholder}.
**Values:** ["diabetes", "hypertension", "asthma", "rheumatoid arthritis", "multiple sclerosis", "chronic kidney disease", "heart failure", "COPD", "psoriasis", "Parkinson's disease", "HIV infection", "breast cancer"], You extract a single numeric value from an answer string. Return only the number and ensure it matches the requested unit.
**Extraction Prompt:** Extract a single numeric value representing cost and ensure the unit is 'dollars per year'. Return "None" if there is no numerical value.

Answer text: {answer_text}

---

**Question:** What is the estimated annual economic cost of {placeholder} in the United States, measured in dollars per year?
**Values:** ["Diet-related chronic diseases", "foodborne illnesses", "drug abuse", "workplace injuries", "medical errors", "Childhood obesity", "Adult obesity", "Micronutrient deficiencies", "chronic diseases", "environmental pollution", "Type 2 diabetes", "High sodium consumption"], You extract a single numeric value from an answer string. Return only the number and ensure it matches the requested unit.
**Extraction Prompt:** Extract a single numeric value and report it in 'dollars per year'. Return "None" if there is no numerical value.

Answer text: {answer_text}

---

**Question:** What is the estimated average indoor concentration of {placeholder} in residential homes, measured in micrograms per cubic meter?
**Values:** ["particulate matter", "formaldehyde", "benzene", "toluene", "nitrogen dioxide", "carbon monoxide", "ammonia", "radon", "ozone", "acetaldehyde", "volatile organic compounds", "mold spores"], You extract a single numeric value from an answer string. Return only the number and ensure it matches the requested unit.
**Extraction Prompt:** Extract a single numeric value. The unit must be micrograms per cubic meter. Return "None" if there is no numerical value.

Answer text: {answer_text}

---

**Question:** What is the average number of meals prepared using {placeholder} per household per month? (unit: meals per month)
**Values:** ["weekdays", "weekends", "holidays", "summer", "winter", "schooldays", "vacation", "festivals", "spring", "autumn", "busy days", "quiet days"], You extract a single numeric value from an answer string. Return only the number and ensure it matches the requested unit.
**Extraction Prompt:** Extract a single numeric value representing the average number of meals prepared using the specified time window/type, reported in meals per month. Return "None" if there is no numerical value.

Answer text: {answer_text}

---

**Question:** What is the estimated annual revenue generated from {placeholder} in the United States, measured in dollars per year?
**Values:** ["online advertising", "pharmaceutical sales", "movie ticket sales", "automobile manu-

facturing", "telecommunications services", "video game industry", "fast food chains", "streaming subscriptions", "health insurance premiums", "retail e-commerce", "professional sports leagues", "music industry"], You extract a single numeric value from an answer string. Return only the number and ensure it matches the requested unit.

**Extraction Prompt:** Extract a single numeric value representing the annual revenue from {placeholder} in dollars per year. Only report a number and 'dollars per year' as the unit. Return "None" if there is no numerical value.

Answer text: {answer_text}

---

**Question:** What fraction of total public health spending is allocated to {placeholder}, measured in percent (%)?
**Values:** ["mental health services", "immunization programs", "maternal care", "chronic disease management", "substance abuse prevention", "emergency preparedness", "HIV/AIDS treatment", "elderly care", "primary care initiatives", "health education campaigns", "rural health outreach", "tuberculosis control"], You extract a single numeric value from an answer string. Return only the number and ensure it matches the requested unit.
**Extraction Prompt:** Extract a single numeric value representing the fraction as a percentage (%). Return "None" if there is no numerical value.

Answer text: {answer_text}

---

**Question:** What is the average annual interest paid on {placeholder} by U.S. households, measured in dollars per year?
**Values:** ["credit cards", "mortgages", "auto loans", "student loans", "personal loans", "home equity lines", "payday loans", "installment loans", "private loans", "business loans", "medical debt", "store credit accounts"], You extract a single numeric value from an answer string. Return only the number and ensure it matches the requested unit.
**Extraction Prompt:** Extract a single numeric value that answers the question, reported in dollars per year. Return "None" if there is no numerical value.

Answer text: {answer_text}

---

**Question:** What is the average daily calorie intake for {placeholder} in kilocalories per day?
**Values:** ["adult men", "adult women", "teenagers", "infants", "preschool children", "pregnant women", "lactating mothers", "elderly adults", "athletes", "office workers", "manual laborers", "vegetarians"], You extract a single numeric value from an answer string. Return only the number and ensure it matches the requested unit.
**Extraction Prompt:** Extract a single numeric value representing daily calorie intake and report the unit as 'kilocalories per day'. Return "None" if there is no numerical value.

Answer text: {answer_text}

---

**Question:** What is the annual growth rate of cost of {placeholder} in the healthcare sector, measured in percent (%) per year?
**Values:** ["prescription drugs", "medical devices", "hospital services", "physician fees", "nursing care", "diagnostic tests", "surgical procedures", "insurance premiums", "emergency care", "laboratory services", "imaging services", "rehabilitation therapy"], You extract a single numeric value from an answer string. Return only the number and ensure it matches the requested unit.
**Extraction Prompt:** Extract a single numeric value representing an annual growth rate, reported in percent (%) per year. Return "None" if there is no numerical value.

Answer text: {answer_text}

---

**Question:** Estimate the number of {placeholder} operating in New York City on an average weekday, measured in vehicles per day.
**Values:** ["taxis", "buses", "delivery trucks", "rideshare cars", "garbage trucks", "ambulances", "fire

engines", "limousines", "school buses", "police cars", "construction vehicles", "motorcycles"], You extract a single numeric value from an answer string. Return only the number and ensure it matches the requested unit.

**Extraction Prompt:** Extract a single numeric value and report it using the unit: vehicles per day. Return "None" if there is no numerical value.

Answer text: {answer_text}

---

**Question:** What is the median distance traveled annually by a {placeholder} in the United States, measured in kilometers per year?
**Values:** ["passenger car", "pickup truck", "SUV", "motorcycle", "school bus", "delivery van", "taxicab", "minivan", "city bus", "semi-truck", "ambulance", "fire engine"], You extract a single numeric value from an answer string. Return only the number and ensure it matches the requested unit.
**Extraction Prompt:** Extract a single numeric value representing the median annual distance traveled by a {placeholder}. The allowed unit is kilometers per year. Return "None" if there is no numerical value.

Answer text: {answer_text}

---

**Question:** What is the average distance traveled per day by a {placeholder} in urban areas, measured in kilometers per day?
**Values:** ["taxi", "bus", "bicycle", "electric scooter", "motorcycle", "delivery van", "ride-share car", "ambulance", "garbage truck", "fire engine", "postal vehicle", "private car"], You extract a single numeric value from an answer string. Return only the number and ensure it matches the requested unit.
**Extraction Prompt:** Extract a single numeric value for average daily distance and use 'kilometers per day' as the unit. Return "None" if there is no numerical value.

Answer text: {answer_text}

---

**Question:** Estimate the total mass of all {placeholder} currently stored in U.S. hospitals, measured in kilograms.
**Values:** ["MRI machines", "CT scanners", "ultrasound devices", "X-ray tubes", "ventilators", "infusion pumps", "dialysis machines", "defibrillators", "anesthesia workstations", "surgical robots", "incubators", "ECG monitors"], You extract a single numeric value from an answer string. Return only the number and ensure it matches the requested unit.
**Extraction Prompt:** Extract a single numeric value and report it in kilograms. Return "None" if there is no numerical value.

Answer text: {answer_text}

---

**Question:** How many servings of {placeholder} are typically consumed by an adult in the United States per week, measured in servings per week?
**Values:** ["breakfast", "lunch", "dinner", "snack", "dessert", "vegetables", "fruits", "meat", "fish", "dairy products", "grains", "salads"], You extract a single numeric value from an answer string. Return only the number and ensure it matches the requested unit.
**Extraction Prompt:** Extract a single numeric value and specify the unit as 'servings per week'. Return "None" if there is no numerical value.

Answer text: {answer_text}

---

**Question:** On average, how many minutes per week does a person spend on {placeholder}?
**Values:** ["exercise", "reading", "cooking", "commuting", "watching television", "cleaning", "working", "shopping", "studying", "socializing", "gardening", "sleeping"], You extract a single numeric value from an answer string. Return only the number and ensure it matches the requested unit.

**Extraction Prompt:** Extract a single numeric value for time spent per week. Allowed unit: minutes per week. Return "None" if there is no numerical value.

Answer text: {answer_text}

---

LOGISTICS

**Question:** What is the estimated number of {placeholder} employed in warehouse logistics worldwide, reported as individuals?
**Values:** ["forklift operators", "inventory managers", "shipping coordinators", "order pickers", "warehouse supervisors", "logistics analysts", "packers", "receiving clerks", "distribution managers", "material handlers", "customs brokers", "freight forwarders"], You extract a single numeric value from an answer string. Return only the number and ensure it matches the requested unit.
**Extraction Prompt:** Extract a single numeric value representing individuals. The allowed unit is 'individuals'. Return "None" if there is no numerical value.

Answer text: {answer_text}

---

**Question:** Estimate the total annual number of {placeholder} incidents reported in major logistics networks worldwide, measured in cases per year.
**Values:** ["cargo theft", "lost shipment", "delayed delivery", "damaged goods", "customs violation", "fraudulent invoice", "piracy attack", "cyber breach", "hazardous spill", "stolen container", "misrouted package", "supply chain disruption"], You extract a single numeric value from an answer string. Return only the number and ensure it matches the requested unit.
**Extraction Prompt:** Extract a single numeric value representing the estimated total annual number of {placeholder} incidents, reported in cases per year. Return "None" if there is no numerical value.

Answer text: {answer_text}

---

**Question:** What is the average financial loss caused by {placeholder} in global supply chains per year, reported in US dollars per year?
**Values:** ["cyberattacks", "natural disasters", "port congestion", "trade wars", "labor strikes", "piracy", "regulatory changes", "pandemics", "supplier insolvency", "transportation delays", "counterfeit goods", "customs bottlenecks"], You extract a single numeric value from an answer string. Return only the number and ensure it matches the requested unit.
**Extraction Prompt:** Extract a single numeric value and ensure it is reported in US dollars per year. Return "None" if there is no numerical value.

Answer text: {answer_text}

---

**Question:** What is the historical peak number of {placeholder} operating in global logistics, measured in vehicles per year?
**Values:** ["container ships", "cargo planes", "delivery trucks", "freight trains", "oil tankers", "bulk carriers", "vans", "electric trucks", "autonomous vehicles", "motorcycles", "reefer trucks", "intermodal trailers"], You extract a single numeric value from an answer string. Return only the number and ensure it matches the requested unit.
**Extraction Prompt:** Extract a single numeric value for the a quantity in the response. Return "None" if there is no numerical value.

Answer text: {answer_text}

---

**Question:** How much revenue is generated from {placeholder} in the logistics industry worldwide each year, measured in US dollars per year?
**Values:** ["freight forwarding", "warehousing", "customs brokerage", "last-mile delivery", "cold chain logistics", "e-commerce fulfillment", "express shipping", "third-party logistics services", "reverse logistics", "container leasing", "transportation management systems", "supply chain consulting"],

You extract a single numeric value from an answer string. Return only the number and ensure it matches the requested unit.
**Extraction Prompt:** Extract a single numeric value representing annual revenue. The allowed unit is US dollars per year. Return "None" if there is no numerical value.

Answer text: {answer_text}

---

**Question:** What is the probability that a shipping container in transit will experience {placeholder}, measured in percent per shipment?
**Values:** ["mold growth", "water damage", "infestation", "corrosion", "contamination", "spoilage", "condensation", "bacterial infection", "fungal contamination", "rusting", "odorous emission", "rot"], You extract a single numeric value from an answer string. Return only the number and ensure it matches the requested unit.
**Extraction Prompt:** Extract a single numeric value representing the probability, reported in percent per shipment. Return "None" if there is no numerical value.

Answer text: {answer_text}

---

**Question:** What is the consumption rate of {placeholder} in a major logistics hub, measured in units per hour?
**Values:** ["pallets", "shipping containers", "fuel drums", "packaging materials", "barcode labels", "forklift batteries", "loading crates", "delivery vans", "conveyor belts", "storage bins", "handheld scanners", "sorting trays"], You extract a single numeric value from an answer string. Return only the number and ensure it matches the requested unit.
**Extraction Prompt:** Extract a single numeric value representing the consumption rate and specify the unit as 'units per hour'. Return "None" if there is no numerical value.

Answer text: {answer_text}

---

**Question:** Estimate the number of {placeholder} operating in Japan at any given moment, measured in vehicles.
**Values:** ["taxis", "buses", "trains", "ambulances", "fire trucks", "police cars", "delivery vans", "motorcycles", "rental cars", "garbage trucks", "private cars", "ride-sharing vehicles"], You extract a single numeric value from an answer string. Return only the number and ensure it matches the requested unit.
**Extraction Prompt:** Extract a single numeric value and report it in vehicles. Return "None" if there is no numerical value.

Answer text: {answer_text}

---

**Question:** Estimate the total energy consumption attributed to {placeholder} in large logistics centers worldwide each year, measured in megawatt-hours per year.
**Values:** ["lighting", "heating", "cooling", "ventilation", "material handling equipment", "refrigeration", "security systems", "conveyor belts", "automated sorting systems", "charging electric vehicles", "water heating", "data processing centers"], You extract a single numeric value from an answer string. Return only the number and ensure it matches the requested unit.
**Extraction Prompt:** Extract a single numeric value representing annual energy consumption, using megawatt-hours per year as the unit. Return "None" if there is no numerical value.

Answer text: {answer_text}

---

**Question:** What is the probability that {placeholder} will experience a major logistics disruption in a given year, measured in percent per year?
**Values:** ["regional warehouse", "supply chain", "distribution center", "retail outlet", "manufacturing facility", "logistics network", "shipping hub", "inventory system", "transport fleet", "customs terminal", "port authority", "fulfillment center"], You extract a single numeric value from an answer string. Return only the number and ensure it matches the requested unit.

**Extraction Prompt:** Extract a single numeric value representing probability. The allowed unit is percent per year. Return "None" if there is no numerical value.

Answer text: {answer_text}

---

**Question:** Estimate the average interest rate charged for {placeholder} used by logistics companies in percent per year.
**Values:** ["working capital loans", "equipment leases", "revolving credit lines", "invoice factoring", "asset-based loans", "vehicle financing", "commercial mortgages", "bridge loans", "trade credit facilities", "term loans", "letters of credit", "fleet leasing agreements"], You extract a single numeric value from an answer string. Return only the number and ensure it matches the requested unit.
**Extraction Prompt:** Extract a single numeric value representing the average interest rate, and specify the unit as percent per year (%/year). Return "None" if there is no numerical value.

Answer text: {answer_text}

---

**Question:** What is the average annual salary earned by {placeholder} working in the logistics industry, measured in US dollars per year?
**Values:** ["warehouse manager", "forklift operator", "supply chain analyst", "logistics coordinator", "inventory specialist", "transportation manager", "customs broker", "shipping clerk", "freight dispatcher", "delivery driver", "operations supervisor", "procurement officer"], You extract a single numeric value from an answer string. Return only the number and ensure it matches the requested unit.
**Extraction Prompt:** Extract a single numeric value representing average annual salary, reported in US dollars per year. Return "None" if there is no numerical value.

Answer text: {answer_text}

---

**Question:** How much maintenance cost for {placeholder} is incurred per kilometer traveled by a delivery truck, measured in US dollars per kilometer?
**Values:** ["engine repairs", "tire replacement", "oil changes", "brake servicing", "transmission maintenance", "suspension repairs", "coolant system upkeep", "battery replacement", "exhaust system repair", "air conditioning maintenance", "electrical system servicing", "fuel system cleaning"], You extract a single numeric value from an answer string. Return only the number and ensure it matches the requested unit.
**Extraction Prompt:** Extract a single numeric value reported in US dollars per kilometer. Return "None" if there is no numerical value.

Answer text: {answer_text}

---

**Question:** What is the total annual value of {placeholder} issued to logistics companies worldwide, measured in US dollars per year?
**Values:** ["trade finance", "invoice factoring", "letters of credit", "equipment leases", "supply chain loans", "working capital loans", "commercial paper", "asset-backed securities", "revolving credit facilities", "warehouse receipts financing", "export credits", "fleet insurance policies"], You extract a single numeric value from an answer string. Return only the number and ensure it matches the requested unit.
**Extraction Prompt:** Extract a single numeric value and report it in 'US dollars per year'. Return "None" if there is no numerical value.

Answer text: {answer_text}

---

**Question:** What is the annual growth rate of revenue from {placeholder} in the logistics sector, measured in percent per year?
**Values:** ["freight forwarding", "warehousing services", "last-mile delivery", "customs brokerage", "cold chain logistics", "express shipping", "reverse logistics", "e-commerce fulfillment", "intermodal transport", "fleet management", "supply chain consulting", "contract logistics"], You extract a single

numeric value from an answer string. Return only the number and ensure it matches the requested unit.

**Extraction Prompt:** Extract a single numeric value representing an annual growth rate. The allowed unit is percent per year. Return "None" if there is no numerical value.

Answer text: {answer_text}

---

**Question:** Estimate the total number of gallons of {placeholder} consumed by the global logistics industry each year.
**Values:** ["diesel", "gasoline", "jet fuel", "marine fuel", "biodiesel", "hydrogen", "liquefied natural gas", "ethanol blend", "synthetic fuel", "renewable diesel", "compressed natural gas", "aviation fuel"], You extract a single numeric value from an answer string. Return only the number and ensure it matches the requested unit.
**Extraction Prompt:** Extract a single numeric value and ensure the unit is gallons per year. Return "None" if there is no numerical value.

Answer text: {answer_text}

---

**Question:** Approximately how many {placeholder} are loaded onto a cargo ship per voyage? (units: items per voyage)
**Values:** ["containers", "cranes", "forklifts", "pallets", "vehicles", "generators", "refrigerators", "tractors", "bulldozers", "excavators", "computers", "machinery"], You extract a single numeric value from an answer string. Return only the number and ensure it matches the requested unit.
**Extraction Prompt:** Extract a single numeric value representing the number of items loaded per voyage, using 'items per voyage' as the unit. Return "None" if there is no numerical value.

Answer text: {answer_text}

---

**Question:** What is the average number of crates of {placeholder} delivered to supermarkets in New York City per day? (units: crates per day)
**Values:** ["apples", "oranges", "bananas", "lettuce", "tomatoes", "potatoes", "carrots", "onions", "grapes", "spinach", "broccoli", "peppers"], You extract a single numeric value from an answer string. Return only the number and ensure it matches the requested unit.
**Extraction Prompt:** Extract a single numeric value and ensure the unit is 'crates per day'. Return "None" if there is no numerical value.

Answer text: {answer_text}

---

**Question:** Estimate the total weight of {placeholder} transported by trucks across Europe in metric tons per year.
**Values:** ["construction materials", "fresh produce", "electronic goods", "automobiles", "industrial machinery", "textiles", "petroleum products", "furniture", "pharmaceuticals", "beverages", "household appliances", "steel"], You extract a single numeric value from an answer string. Return only the number and ensure it matches the requested unit.
**Extraction Prompt:** Extract a single numeric value representing total weight, and ensure the unit is metric tons per year. Return "None" if there is no numerical value.

Answer text: {answer_text}

---

**Question:** How long does it typically take for {placeholder} to reach completion in days per shipment?
**Values:** ["customs clearance", "order processing", "quality inspection", "inventory restocking", "freight consolidation", "packaging preparation", "route planning", "document verification", "payment confirmation", "load scheduling", "cargo unloading", "delivery coordination"], You extract a single numeric value from an answer string. Return only the number and ensure it matches the requested unit.

**Extraction Prompt:** Extract a single numeric value representing the typical completion time for {placeholder}, reported in days per shipment. Return "None" if there is no numerical value.

Answer text: {answer_text}

---

**Question:** What percentage of warehouse workers experience {placeholder} each year due to repetitive lifting? (units: percent per year)
**Values:** ["lower back pain", "shoulder strain", "tendinitis", "herniated discs", "carpal tunnel syndrome", "muscle fatigue", "sprains", "rotator cuff injuries", "joint inflammation", "elbow pain", "chronic soreness", "ligament tears"], You extract a single numeric value from an answer string. Return only the number and ensure it matches the requested unit.
**Extraction Prompt:** Extract a single numeric value. Unit must be 'percent per year'. Return "None" if there is no numerical value.

Answer text: {answer_text}

---

**Question:** What is the average amount of {placeholder} transported by a single refrigerated truck in kilograms per trip?
**Values:** ["beef", "chicken", "lettuce", "milk", "cheese", "yogurt", "apples", "broccoli", "carrots", "ice cream", "fish fillets", "tomatoes"], You extract a single numeric value from an answer string. Return only the number and ensure it matches the requested unit.
**Extraction Prompt:** Extract one numeric value representing the average quantity, using kilograms per trip as the unit. Return "None" if there is no numerical value.

Answer text: {answer_text}

---

**Question:** Estimate the total number of {placeholder} handled by a medium-sized logistics company in one month (units: shipments per month).
**Values:** ["overnight shipments", "express packages", "international deliveries", "standard parcels", "bulk consignments", "fragile items", "return shipments", "same-day deliveries", "temperature-controlled goods", "e-commerce orders", "seasonal shipments", "high-value packages"], You extract a single numeric value from an answer string. Return only the number and ensure it matches the requested unit.
**Extraction Prompt:** Extract a single numeric value representing the estimated number of shipments handled per month. The allowed unit is 'shipments per month'. Return "None" if there is no numerical value.

Answer text: {answer_text}

---

**Question:** What is the annual turnover rate for {placeholder} working in warehouse logistics, measured in percent per year?
**Values:** ["female employees", "male employees", "temporary staff", "full-time workers", "part-time workers", "seasonal workers", "shift supervisors", "older workers", "younger employees", "new hires", "contractors", "management staff"], You extract a single numeric value from an answer string. Return only the number and ensure it matches the requested unit.
**Extraction Prompt:** Extract a single numeric value representing an annual turnover rate, reported in percent per year. Return "None" if there is no numerical value.

Answer text: {answer_text}

---

**Question:** If all the {placeholder} incidents reported by logistics companies in one year were placed into a single file, how many pages would it contain? (units: pages per year)
**Values:** ["vehicle breakdown", "missed delivery", "cargo theft", "shipment delay", "lost package", "inventory discrepancy", "equipment malfunction", "damaged goods", "routing error", "documentation error", "fuel shortage", "container misplacement"], You extract a single numeric value from an answer string. Return only the number and ensure it matches the requested unit.

**Extraction Prompt:** Extract the estimated total number of pages and report the value in 'pages per year'. Return "None" if there is no numerical value.

Answer text: {answer_text}

---

**Question:** Estimate the total number of cases of {placeholder} reported among long-haul truck drivers in the United States per year (units: cases per year).
**Values:** ["sleep apnea", "hypertension", "diabetes", "depression", "obesity", "back pain", "lung cancer", "hepatitis C", "skin infections", "substance abuse", "cardiovascular disease", "chronic fatigue"], You extract a single numeric value from an answer string. Return only the number and ensure it matches the requested unit.
**Extraction Prompt:** Extract a single numeric value representing the total annual cases, using 'cases per year' as the unit. Return "None" if there is no numerical value.

Answer text: {answer_text}

---

**Question:** Estimate the total market value of all {placeholder} currently stored in U.S. warehouses, measured in US dollars.
**Values:** ["soybeans", "automobiles", "furniture", "pharmaceuticals", "electronics", "apparel", "petroleum", "coffee beans", "lumber", "steel coils", "corn", "copper wire"], You extract a single numeric value from an answer string. Return only the number and ensure it matches the requested unit.
**Extraction Prompt:** Extract a single numeric value representing the estimated total market value. The allowed unit is US dollars. Return "None" if there is no numerical value.

Answer text: {answer_text}

---

**Question:** What is the historical peak number of {placeholder} operating in global logistics, measured in vehicles per year?
**Values:** ["container ships", "cargo planes", "delivery trucks", "freight trains", "oil tankers", "bulk carriers", "vans", "electric trucks", "autonomous vehicles", "motorcycles", "reefer trucks", "intermodal trailers"], You extract a single numeric value from an answer string. Return only the number and ensure it matches the requested unit.
**Extraction Prompt:** Extract a single numeric value followed by 'vehicles per year'. Return "None" if there is no numerical value.

Answer text: {answer_text}

---

**Question:** How much revenue is generated from {placeholder} in the logistics industry worldwide each year, measured in US dollars per year?
**Values:** ["freight forwarding", "warehousing", "customs brokerage", "last-mile delivery", "cold chain logistics", "e-commerce fulfillment", "express shipping", "third-party logistics services", "reverse logistics", "container leasing", "transportation management systems", "supply chain consulting"], You extract a single numeric value from an answer string. Return only the number and ensure it matches the requested unit.
**Extraction Prompt:** Extract a single numeric value representing annual revenue. The allowed unit is US dollars per year. Return "None" if there is no numerical value.

Answer text: {answer_text}

---

**Question:** What is the probability that a shipping container in transit will experience {placeholder}, measured in percent per shipment?
**Values:** ["mold growth", "water damage", "infestation", "corrosion", "contamination", "spoilage", "condensation", "bacterial infection", "fungal contamination", "rusting", "odorous emission", "rot"], You extract a single numeric value from an answer string. Return only the number and ensure it matches the requested unit.

**Extraction Prompt:** Extract a single numeric value representing the probability, reported in percent per shipment. Return "None" if there is no numerical value.

Answer text: {answer_text}

---

**Question:** What is the consumption rate of {placeholder} in a major logistics hub, measured in units per hour?
**Values:** ["pallets", "shipping containers", "fuel drums", "packaging materials", "barcode labels", "forklift batteries", "loading crates", "delivery vans", "conveyor belts", "storage bins", "handheld scanners", "sorting trays"], You extract a single numeric value from an answer string. Return only the number and ensure it matches the requested unit.
**Extraction Prompt:** Extract a single numeric value representing the consumption rate and specify the unit as 'units per hour'. Return "None" if there is no numerical value.

Answer text: {answer_text}

---

**Question:** Estimate the number of {placeholder} operating in Japan at any given moment, measured in vehicles.
**Values:** ["taxis", "buses", "trains", "ambulances", "fire trucks", "police cars", "delivery vans", "motorcycles", "rental cars", "garbage trucks", "private cars", "ride-sharing vehicles"], You extract a single numeric value from an answer string. Return only the number and ensure it matches the requested unit.
**Extraction Prompt:** Extract a single numeric value and report it in vehicles. Return "None" if there is no numerical value.

Answer text: {answer_text}

---

**Question:** Estimate the total energy consumption attributed to {placeholder} in large logistics centers worldwide each year, measured in megawatt-hours per year.
**Values:** ["lighting", "heating", "cooling", "ventilation", "material handling equipment", "refrigeration", "security systems", "conveyor belts", "automated sorting systems", "charging electric vehicles", "water heating", "data processing centers"], You extract a single numeric value from an answer string. Return only the number and ensure it matches the requested unit.
**Extraction Prompt:** Extract a single numeric value representing annual energy consumption, using megawatt-hours per year as the unit. Return "None" if there is no numerical value.

Answer text: {answer_text}

---

**Question:** What is the probability that {placeholder} will experience a major logistics disruption in a given year, measured in percent per year?
**Values:** ["regional warehouse", "supply chain", "distribution center", "retail outlet", "manufacturing facility", "logistics network", "shipping hub", "inventory system", "transport fleet", "customs terminal", "port authority", "fulfillment center"], You extract a single numeric value from an answer string. Return only the number and ensure it matches the requested unit.
**Extraction Prompt:** Extract a single numeric value representing probability. The allowed unit is percent per year. Return "None" if there is no numerical value.

Answer text: {answer_text}

---

**Question:** Estimate the average interest rate charged for {placeholder} used by logistics companies in percent per year.
**Values:** ["working capital loans", "equipment leases", "revolving credit lines", "invoice factoring", "asset-based loans", "vehicle financing", "commercial mortgages", "bridge loans", "trade credit facilities", "term loans", "letters of credit", "fleet leasing agreements"], You extract a single numeric value from an answer string. Return only the number and ensure it matches the requested unit.
**Extraction Prompt:** Extract a single numeric value representing the average interest rate, and specify the unit as percent per year (%/year). Return "None" if there is no numerical value.

Answer text: {answer_text}

---

**Question:** What is the average annual salary earned by {placeholder} working in the logistics industry, measured in US dollars per year?
**Values:** ["warehouse manager", "forklift operator", "supply chain analyst", "logistics coordinator", "inventory specialist", "transportation manager", "customs broker", "shipping clerk", "freight dispatcher", "delivery driver", "operations supervisor", "procurement officer"], You extract a single numeric value from an answer string. Return only the number and ensure it matches the requested unit.
**Extraction Prompt:** Extract a single numeric value representing average annual salary, reported in US dollars per year. Return "None" if there is no numerical value.

Answer text: {answer_text}

---

**Question:** How much maintenance cost for {placeholder} is incurred per kilometer traveled by a delivery truck, measured in US dollars per kilometer?
**Values:** ["engine repairs", "tire replacement", "oil changes", "brake servicing", "transmission maintenance", "suspension repairs", "coolant system upkeep", "battery replacement", "exhaust system repair", "air conditioning maintenance", "electrical system servicing", "fuel system cleaning"], You extract a single numeric value from an answer string. Return only the number and ensure it matches the requested unit.
**Extraction Prompt:** Extract a single numeric value reported in US dollars per kilometer. Return "None" if there is no numerical value.

Answer text: {answer_text}

---

**Question:** What is the total annual value of {placeholder} issued to logistics companies worldwide, measured in US dollars per year?
**Values:** ["trade finance", "invoice factoring", "letters of credit", "equipment leases", "supply chain loans", "working capital loans", "commercial paper", "asset-backed securities", "revolving credit facilities", "warehouse receipts financing", "export credits", "fleet insurance policies"], You extract a single numeric value from an answer string. Return only the number and ensure it matches the requested unit.
**Extraction Prompt:** Extract a single numeric value and report it in 'US dollars per year'. Return "None" if there is no numerical value.

Answer text: {answer_text}

---

**Question:** What is the annual growth rate of revenue from {placeholder} in the logistics sector, measured in percent per year?
**Values:** ["freight forwarding", "warehousing services", "last-mile delivery", "customs brokerage", "cold chain logistics", "express shipping", "reverse logistics", "e-commerce fulfillment", "intermodal transport", "fleet management", "supply chain consulting", "contract logistics"], You extract a single numeric value from an answer string. Return only the number and ensure it matches the requested unit.
**Extraction Prompt:** Extract a single numeric value representing an annual growth rate. The allowed unit is percent per year. Return "None" if there is no numerical value.

Answer text: {answer_text}

---

**Question:** Estimate the total number of gallons of {placeholder} consumed by the global logistics industry each year.
**Values:** ["diesel", "gasoline", "jet fuel", "marine fuel", "biodiesel", "hydrogen", "liquefied natural gas", "ethanol blend", "synthetic fuel", "renewable diesel", "compressed natural gas", "aviation fuel"], You extract a single numeric value from an answer string. Return only the number and ensure it matches the requested unit.

**Extraction Prompt:** Extract a single numeric value and ensure the unit is gallons per year. Return "None" if there is no numerical value.

Answer text: {answer_text}

---

**Question:** Approximately how many {placeholder} are loaded onto a cargo ship per voyage? (units: items per voyage)
**Values:** ["containers", "cranes", "forklifts", "pallets", "vehicles", "generators", "refrigerators", "tractors", "bulldozers", "excavators", "computers", "machinery"], You extract a single numeric value from an answer string. Return only the number and ensure it matches the requested unit.
**Extraction Prompt:** Extract a single numeric value representing the number of items loaded per voyage, using 'items per voyage' as the unit. Return "None" if there is no numerical value.

Answer text: {answer_text}

---

**Question:** What is the average number of crates of {placeholder} delivered to supermarkets in New York City per day? (units: crates per day)
**Values:** ["apples", "oranges", "bananas", "lettuce", "tomatoes", "potatoes", "carrots", "onions", "grapes", "spinach", "broccoli", "peppers"], You extract a single numeric value from an answer string. Return only the number and ensure it matches the requested unit.
**Extraction Prompt:** Extract a single numeric value and ensure the unit is 'crates per day'. Return "None" if there is no numerical value.

Answer text: {answer_text}

---

**Question:** Estimate the total weight of {placeholder} transported by trucks across Europe in metric tons per year.
**Values:** ["construction materials", "fresh produce", "electronic goods", "automobiles", "industrial machinery", "textiles", "petroleum products", "furniture", "pharmaceuticals", "beverages", "household appliances", "steel"], You extract a single numeric value from an answer string. Return only the number and ensure it matches the requested unit.
**Extraction Prompt:** Extract a single numeric value representing total weight, and ensure the unit is metric tons per year. Return "None" if there is no numerical value.

Answer text: {answer_text}

---

**Question:** How long does it typically take for {placeholder} to reach completion in days per shipment?
**Values:** ["customs clearance", "order processing", "quality inspection", "inventory restocking", "freight consolidation", "packaging preparation", "route planning", "document verification", "payment confirmation", "load scheduling", "cargo unloading", "delivery coordination"], You extract a single numeric value from an answer string. Return only the number and ensure it matches the requested unit.
**Extraction Prompt:** Extract a single numeric value representing the typical completion time for {placeholder}, reported in days per shipment. Return "None" if there is no numerical value.

Answer text: {answer_text}

---

**Question:** What percentage of warehouse workers experience {placeholder} each year due to repetitive lifting? (units: percent per year)
**Values:** ["lower back pain", "shoulder strain", "tendinitis", "herniated discs", "carpal tunnel syndrome", "muscle fatigue", "sprains", "rotator cuff injuries", "joint inflammation", "elbow pain", "chronic soreness", "ligament tears"], You extract a single numeric value from an answer string. Return only the number and ensure it matches the requested unit.
**Extraction Prompt:** Extract a single numeric value. Unit must be 'percent per year'. Return "None" if there is no numerical value.

Answer text: {answer_text}

---

**Question:** What is the average amount of {placeholder} transported by a single refrigerated truck in kilograms per trip?
**Values:** ["beef", "chicken", "lettuce", "milk", "cheese", "yogurt", "apples", "broccoli", "carrots", "ice cream", "fish fillets", "tomatoes"], You extract a single numeric value from an answer string. Return only the number and ensure it matches the requested unit.
**Extraction Prompt:** Extract one numeric value representing the average quantity, using kilograms per trip as the unit. Return "None" if there is no numerical value.

Answer text: {answer_text}

---

**Question:** Estimate the total number of {placeholder} handled by a medium-sized logistics company in one month (units: shipments per month).
**Values:** ["overnight shipments", "express packages", "international deliveries", "standard parcels", "bulk consignments", "fragile items", "return shipments", "same-day deliveries", "temperature-controlled goods", "e-commerce orders", "seasonal shipments", "high-value packages"], You extract a single numeric value from an answer string. Return only the number and ensure it matches the requested unit.
**Extraction Prompt:** Extract a single numeric value representing the estimated number of shipments handled per month. The allowed unit is 'shipments per month'. Return "None" if there is no numerical value.

Answer text: {answer_text}

---

**Question:** What is the annual turnover rate for {placeholder} working in warehouse logistics, measured in percent per year?
**Values:** ["female employees", "male employees", "temporary staff", "full-time workers", "part-time workers", "seasonal workers", "shift supervisors", "older workers", "younger employees", "new hires", "contractors", "management staff"], You extract a single numeric value from an answer string. Return only the number and ensure it matches the requested unit.
**Extraction Prompt:** Extract a single numeric value representing an annual turnover rate, reported in percent per year. Return "None" if there is no numerical value.

Answer text: {answer_text}

---

**Question:** If all the {placeholder} incidents reported by logistics companies in one year were placed into a single file, how many pages would it contain? (units: pages per year)
**Values:** ["vehicle breakdown", "missed delivery", "cargo theft", "shipment delay", "lost package", "inventory discrepancy", "equipment malfunction", "damaged goods", "routing error", "documentation error", "fuel shortage", "container misplacement"], You extract a single numeric value from an answer string. Return only the number and ensure it matches the requested unit.
**Extraction Prompt:** Extract the estimated total number of pages and report the value in 'pages per year'. Return "None" if there is no numerical value.

Answer text: {answer_text}

---

**Question:** Estimate the total number of cases of {placeholder} reported among long-haul truck drivers in the United States per year (units: cases per year).
**Values:** ["sleep apnea", "hypertension", "diabetes", "depression", "obesity", "back pain", "lung cancer", "hepatitis C", "skin infections", "substance abuse", "cardiovascular disease", "chronic fatigue"], You extract a single numeric value from an answer string. Return only the number and ensure it matches the requested unit.
**Extraction Prompt:** Extract a single numeric value representing the total annual cases, using 'cases per year' as the unit. Return "None" if there is no numerical value.

Answer text: {answer_text}

**Question:** Estimate the total market value of all {placeholder} currently stored in U.S. warehouses, measured in US dollars.
**Values:** ["soybeans", "automobiles", "furniture", "pharmaceuticals", "electronics", "apparel", "petroleum", "coffee beans", "lumber", "steel coils", "corn", "copper wire"], You extract a single numeric value from an answer string. Return only the number and ensure it matches the requested unit.
**Extraction Prompt:** Extract a single numeric value representing the estimated total market value. The allowed unit is US dollars. Return "None" if there is no numerical value.

Answer text: {answer_text}

PERSONAL FINANCE

**Question:** On average, how many grams of {placeholder} does a U.S. adult consume per day?
**Values:** ["protein", "fiber", "sugar", "fat", "carbohydrate", "sodium", "cholesterol", "calcium", "potassium", "magnesium", "iron", "vitamin C"], You extract a single numeric value from an answer string. Return only the number and ensure it matches the requested unit.
**Extraction Prompt:** Extract a single numeric value representing the amount consumed per day, reported in grams. Return "None" if there is no numerical value.

Answer text: {answer_text}

**Question:** What is the average monthly electricity usage attributed specifically to {placeholder} in a typical U.S. household (kWh per month)?
**Values:** ["refrigerator", "air conditioning", "lighting", "water heater", "clothes dryer", "dishwasher", "television", "microwave oven", "freezer", "space heating", "computer equipment", "washing machine"], You extract a single numeric value from an answer string. Return only the number and ensure it matches the requested unit.
**Extraction Prompt:** Extract a single numeric value and the exact allowed unit: kWh per month. Return "None" if there is no numerical value.

Answer text: {answer_text}

**Question:** On average, how much do households in the United States spend on transportation for {placeholder} per year (dollars per year)?
**Values:** ["retirees", "single adults", "families with children", "urban residents", "rural households", "college students", "low-income families", "high-income households", "immigrants", "senior citizens", "military families", "recent graduates"], You extract a single numeric value from an answer string. Return only the number and ensure it matches the requested unit.
**Extraction Prompt:** Extract a single numeric value representing dollars spent per year; the allowed unit is 'dollars per year'. Return "None" if there is no numerical value.

Answer text: {answer_text}

**Question:** What is the average financial recovery time for a household after experiencing {placeholder} in the United States (months per incident)?
**Values:** ["job loss", "medical emergency", "natural disaster", "house fire", "identity theft", "car accident", "flooding", "burglary", "divorce", "eviction", "cyberattack", "bankruptcy"], You extract a single numeric value from an answer string. Return only the number and ensure it matches the requested unit.
**Extraction Prompt:** Extract a single numeric value representing the average number of months required for financial recovery per incident; unit: months per incident. Return "None" if there is no numerical value.

Answer text: {answer_text}

**Question:** Estimate the total number of active {placeholder} accounts in the United States (accounts).
**Values:** ["monthly", "daily", "weekly", "annual", "student", "business", "retail", "savings", "checking", "joint", "corporate", "online"], You extract a single numeric value from an answer string. Return only the number and ensure it matches the requested unit.
**Extraction Prompt:** Extract a single numeric value representing the estimated total number of active {placeholder} accounts in the United States. Use 'accounts' as the unit. Return "None" if there is no numerical value.

Answer text: {answer_text}

---

**Question:** Estimate the average annual child care expense for households with {placeholder} in the United States (dollars per year).
**Values:** ["single mothers", "two working parents", "military families", "immigrant families", "foster children", "parents under 25", "rural residents", "urban households", "Latino families", "Asian American parents", "low-income households", "same-sex couples"], You extract a single numeric value from an answer string. Return only the number and ensure it matches the requested unit.
**Extraction Prompt:** Extract a single numeric value representing average annual child care expense. The allowed unit is 'dollars per year.' Return "None" if there is no numerical value.

Answer text: {answer_text}

---

**Question:** What is the average monthly income earned from {placeholder} by a typical household in the United States (dollars per month)?
**Values:** ["rental properties", "dividends", "social security", "freelance work", "pension", "stock investments", "side business", "online sales", "royalties", "child support payments", "interest income", "government assistance"], You extract a single numeric value from an answer string. Return only the number and ensure it matches the requested unit.
**Extraction Prompt:** Extract a single numeric value. The unit must be dollars per month. Return "None" if there is no numerical value.

Answer text: {answer_text}

---

**Question:** What is the average monthly expenditure on {placeholder} for a single adult in the United States (dollars per month)?
**Values:** ["groceries", "clothing", "toiletries", "household supplies", "electronics", "furniture", "pet food", "medications", "personal care products", "cleaning products", "books", "stationery"], You extract a single numeric value from an answer string. Return only the number and ensure it matches the requested unit.
**Extraction Prompt:** Extract a single numeric value representing average monthly expenditure, reported in dollars per month. Return "None" if there is no numerical value.

Answer text: {answer_text}

---

**Question:** Estimate the total annual amount spent on replacement of {placeholder} in the United States (dollars per year).
**Values:** ["automobile tires", "roof shingles", "water heaters", "air conditioners", "refrigerators", "light bulbs", "cell phones", "laptop computers", "washing machines", "televisions", "furnaces", "car batteries"], You extract a single numeric value from an answer string. Return only the number and ensure it matches the requested unit.
**Extraction Prompt:** Extract a single numeric value and ensure it is reported in dollars per year. Return "None" if there is no numerical value.

Answer text: {answer_text}

---

**Question:** Estimate the average annual household spending on internet services for {placeholder} in the United States (dollars per year).
**Values:** ["urban families", "rural households", "millennials", "retirees", "college students", "single-parent families", "low-income households", "high-income households", "suburban residents", "tech enthusiasts", "remote workers", "senior citizens"], You extract a single numeric value from an answer string. Return only the number and ensure it matches the requested unit.
**Extraction Prompt:** Extract a single numeric value representing annual household spending. Use 'dollars per year' as the unit. Return "None" if there is no numerical value.

Answer text: {answer_text}

---

**Question:** What is the total number of operational {placeholder} currently in use by households in the United States (units)?
**Values:** ["refrigerators", "microwave ovens", "dishwashers", "washing machines", "televisions", "air conditioners", "water heaters", "clothes dryers", "vacuum cleaners", "computers", "smoke detectors", "space heaters"], You extract a single numeric value from an answer string. Return only the number and ensure it matches the requested unit.
**Extraction Prompt:** Extract a single integer value representing the total count. The allowed unit is 'units'. Return "None" if there is no numerical value.

Answer text: {answer_text}

---

**Question:** Estimate the total annual electricity consumption of all the {placeholder} in the United States (kWh per year).
**Values:** ["refrigerators", "air conditioners", "washing machines", "televisions", "microwave ovens", "dishwashers", "water heaters", "clothes dryers", "computers", "freezers", "electric stoves", "space heaters"], You extract a single numeric value from an answer string. Return only the number and ensure it matches the requested unit.
**Extraction Prompt:** Extract a single numeric value for total annual electricity consumption. The allowed unit is kWh per year. Return "None" if there is no numerical value.

Answer text: {answer_text}

---

**Question:** What is the average monthly payment required to maintain a {placeholder} in the United States (dollars per month)?
**Values:** ["mortgage", "car lease", "health insurance plan", "cell phone plan", "student loan", "gym membership", "internet subscription", "rent", "childcare service", "homeowners insurance policy", "streaming service subscription", "utility bill"], You extract a single numeric value from an answer string. Return only the number and ensure it matches the requested unit.
**Extraction Prompt:** Extract a single numeric value and report it in dollars per month. Return "None" if there is no numerical value.

Answer text: {answer_text}

---

**Question:** Estimate the average financial loss from one incident of {placeholder} for a household in the United States (dollars per incident).
**Values:** ["burglary", "fire", "flooding", "identity theft", "car theft", "vandalism", "cyberattack", "medical emergency", "appliance failure", "water leak", "windstorm damage", "earthquake"], You extract a single numeric value from an answer string. Return only the number and ensure it matches the requested unit.
**Extraction Prompt:** Extract a single numeric value in dollars per incident, e.g., '700 dollars per incident'. Return "None" if there is no numerical value.

Answer text: {answer_text}

---

**Question:** Estimate the total annual spending on repairs for {placeholder} by households in the United States (dollars per year).

**Values:** ["roofing", "plumbing systems", "heating systems", "air conditioning units", "water heaters", "kitchen appliances", "garage doors", "windows", "electrical wiring", "flooring surfaces", "exterior siding", "bathroom fixtures"], You extract a single numeric value from an answer string. Return only the number and ensure it matches the requested unit.
**Extraction Prompt:** Extract a single numeric value representing total annual spending. The allowed unit is 'dollars per year'. Return "None" if there is no numerical value.

Answer text: {answer_text}

---

**Question:** Estimate the total annual spending on {placeholder} for all households in the United States (dollars per year).
**Values:** ["toilet paper", "laundry detergent", "pet food", "paper towels", "coffee beans", "bottled water", "diapers", "cleaning supplies", "milk", "bread", "trash bags", "dish soap"], You extract a single numeric value from an answer string. Return only the number and ensure it matches the requested unit.
**Extraction Prompt:** Extract a single numeric value and report it in dollars per year. Return "None" if there is no numerical value.

Answer text: {answer_text}

---

**Question:** Estimate the total number of miles traveled per year by all {placeholder} in the United States (miles per year).
**Values:** ["cars", "pickup trucks", "motorcycles", "school buses", "delivery vans", "semi trucks", "city buses", "SUVs", "minivans", "ambulances", "fire trucks", "taxis"], You extract a single numeric value from an answer string. Return only the number and ensure it matches the requested unit.
**Extraction Prompt:** Extract a single numeric value representing total miles per year. The allowed unit is 'miles per year'. Return "None" if there is no numerical value.

Answer text: {answer_text}

---

**Question:** Estimate the total annual federal budget allocation for {placeholder} in the United States (dollars per year).
**Values:** ["Medicaid", "defense spending", "education", "infrastructure", "scientific research", "veterans benefits", "environmental protection", "homeland security", "agriculture subsidies", "public transportation", "student loans", "healthcare"], You extract a single numeric value from an answer string. Return only the number and ensure it matches the requested unit.
**Extraction Prompt:** Extract a single numeric value representing an annual amount, reported in 'dollars per year.' Return "None" if there is no numerical value.

Answer text: {answer_text}

---

**Question:** What is the projected lifetime total spent on {placeholder} for an average adult in the United States (dollars over a lifetime)?
**Values:** ["groceries", "healthcare", "education", "transportation", "housing", "vacations", "clothing", "dining out", "insurance premiums", "pet care", "entertainment", "childcare"], You extract a single numeric value from an answer string. Return only the number and ensure it matches the requested unit.
**Extraction Prompt:** Extract a single numeric value representing dollars over a lifetime. Return "None" if there is no numerical value.

Answer text: {answer_text}

---

**Question:** What is the average annual insurance payout for claims related to {placeholder} in the United States (dollars per year)?
**Values:** ["diabetes", "cancer", "stroke", "heart attack", "asthma", "arthritis", "COPD", "hypertension", "kidney failure", "depression", "HIV/AIDS", "multiple sclerosis"], You extract a single numeric value from an answer string. Return only the number and ensure it matches the requested unit.

**Extraction Prompt:** Extract a single numeric value representing average annual insurance payout for claims related to {placeholder}, reported in dollars per year. Return "None" if there is no numerical value.

Answer text: {answer_text}

---

**Question:** What is the average amount of {placeholder} contributed to retirement accounts per working adult in the United States each year (dollars per year)?
**Values:** ["income", "salary", "wages", "bonus", "commission", "overtime pay", "dividends", "interest", "tax refund", "gift money", "inheritance", "profit"], You extract a single numeric value from an answer string. Return only the number and ensure it matches the requested unit.
**Extraction Prompt:** Extract a single numeric value representing the average annual contribution and include the unit 'dollars per year'. Return "None" if there is no numerical value.

Answer text: {answer_text}

---

**Question:** What is the average annual revenue generated from {placeholder} by a small business in the United States (dollars per year)?
**Values:** ["online sales", "consulting services", "product subscriptions", "advertising", "affiliate marketing", "retail operations", "event hosting", "franchise fees", "membership dues", "service contracts", "licensing agreements", "training workshops"], You extract a single numeric value from an answer string. Return only the number and ensure it matches the requested unit.
**Extraction Prompt:** Extract a single numeric value representing annual revenue. The allowed unit is dollars per year. Return "None" if there is no numerical value.

Answer text: {answer_text}

---

**Question:** What is the average annual depreciation rate of a {placeholder} in the United States (% per year)?
**Values:** ["sedan", "pickup truck", "SUV", "motorcycle", "RV", "commercial van", "luxury car", "electric vehicle", "hybrid car", "sports car", "minivan", "cargo trailer"], You extract a single numeric value from an answer string. Return only the number and ensure it matches the requested unit.
**Extraction Prompt:** Extract a single numeric value representing an annual depreciation rate, using percent per year (% per year) as the unit. Return "None" if there is no numerical value.

Answer text: {answer_text}

---

**Question:** What is the average annual household spending on managing {placeholder} in the United States (dollars per year)?
**Values:** ["diabetes", "asthma", "hypertension", "arthritis", "obesity", "depression", "allergies", "cancer", "heart disease", "migraine", "eczema", "osteoporosis"], You extract a single numeric value from an answer string. Return only the number and ensure it matches the requested unit.
**Extraction Prompt:** Extract a single numeric value representing annual household spending. Unit must be 'dollars per year'. Return "None" if there is no numerical value.

Answer text: {answer_text}

---

**Question:** Estimate the total annual household consumption of {placeholder} in the United States (kilograms per year).
**Values:** ["sugar", "rice", "beef", "cheese", "chicken", "potatoes", "wheat flour", "eggs", "milk powder", "pasta", "apples", "fish"], You extract a single numeric value from an answer string. Return only the number and ensure it matches the requested unit.
**Extraction Prompt:** Extract a single numeric value representing annual household consumption. Only use 'kilograms per year' as the unit. Return "None" if there is no numerical value.

Answer text: {answer_text}

---

**Question:** Estimate the total number of {placeholder} applications submitted in the United States in one year (applications per year).
**Values:** ["patent", "trademark", "copyright", "asylum", "immigration", "student visa", "work permit", "green card", "welfare", "unemployment benefit", "Medicaid", "social security"], You extract a single numeric value from an answer string. Return only the number and ensure it matches the requested unit.
**Extraction Prompt:** Extract a single numeric value representing applications per year. Return "None" if there is no numerical value.

Answer text: {answer_text}

---

SOCIAL PLANNING

**Question:** What is the estimated average salary for {placeholder} working in metropolitan areas of Canada? (dollars per year)
**Values:** ["software engineers", "accountants", "registered nurses", "civil engineers", "marketing managers", "financial analysts", "primary school teachers", "construction managers", "graphic designers", "pharmacists", "electricians", "lawyers"], You extract a single numeric value from an answer string. Return only the number and ensure it matches the requested unit.
**Extraction Prompt:** Extract a single numeric value in dollars per year, representing the average annual salary for {placeholder} working in Canadian metropolitan areas. Return "None" if there is no numerical value.

Answer text: {answer_text}

---

**Question:** What is the average annual revenue generated from {placeholder} by a midsize US city (dollars per year)?
**Values:** ["parking fees", "property taxes", "sales taxes", "hotel occupancy taxes", "business licenses", "building permits", "utility services", "transit fares", "recreational facilities", "zoning applications", "waste collection services", "franchise agreements"], You extract a single numeric value from an answer string. Return only the number and ensure it matches the requested unit.
**Extraction Prompt:** Extract a single numeric value representing average annual revenue, and ensure the unit is 'dollars per year'. Return "None" if there is no numerical value.

Answer text: {answer_text}

---

**Question:** What is the expected waiting time until {placeholder} is approved by local government in a typical medium-sized city (days)?
**Values:** ["zoning variance", "building permit", "business license", "environmental impact assessment", "liquor license", "noise ordinance waiver", "parking permit", "signage approval", "health code exception", "short-term rental permit", "street closure request", "public event permit"], You extract a single numeric value from an answer string. Return only the number and ensure it matches the requested unit.
**Extraction Prompt:** Extract a single numeric value representing the expected waiting time, reported in days. Return "None" if there is no numerical value.

Answer text: {answer_text}

---

**Question:** What is the cumulative hours volunteered by {placeholder} in public community programs over one year in a typical large city (hours per year)?
**Values:** ["teenagers", "retirees", "college students", "corporate employees", "high school teachers", "medical professionals", "single parents", "immigrants", "disabled adults", "faith group members", "government workers", "young adults"], You extract a single numeric value from an answer string. Return only the number and ensure it matches the requested unit.
**Extraction Prompt:** Extract a single numeric value representing total hours volunteered per year; report the answer in 'hours per year'. Return "None" if there is no numerical value.

Answer text: {answer_text}

---

**Question:** What fraction of total emergency shelter usage is accounted for by {placeholder} in a typical large metropolitan area (% of total usage)?
**Values:** ["fire", "flood", "earthquake", "hurricane", "tornado", "winter storm", "heatwave", "pandemic outbreak", "power outage", "chemical spill", "civil unrest", "building collapse"], You extract a single numeric value from an answer string. Return only the number and ensure it matches the requested unit.
**Extraction Prompt:** Extract a single numeric value representing a percentage. Only provide the answer as '% of total usage'. Return "None" if there is no numerical value.

Answer text: {answer_text}

---

**Question:** What is the estimated total area covered by {placeholder} in public parks of a typical large city (square meters)?
**Values:** ["grass", "flowerbeds", "trees", "playgrounds", "ponds", "walking paths", "bushes", "picnic areas", "sports fields", "dog parks", "benches zones", "community gardens"], You extract a single numeric value from an answer string. Return only the number and ensure it matches the requested unit.
**Extraction Prompt:** Extract a single numeric value and report it in square meters. Return "None" if there is no numerical value.

Answer text: {answer_text}

---

**Question:** How many individual {placeholder} are installed in public recreation facilities of a typical mid-sized U.S. city (units)?
**Values:** ["basketball hoops", "treadmills", "picnic tables", "water fountains", "playground swings", "benches", "soccer goals", "volleyball nets", "bike racks", "trash cans", "lighting fixtures", "tennis courts"], You extract a single numeric value from an answer string. Return only the number and ensure it matches the requested unit.
**Extraction Prompt:** Extract a single numeric value for the estimated number, reported in units. Return "None" if there is no numerical value.

Answer text: {answer_text}

---

**Question:** What is the average number of hours per week that a typical community center in a mid-sized city allocates to {placeholder} (hours per week)?
**Values:** ["youth programs", "fitness classes", "arts workshops", "senior activities", "sports leagues", "volunteer events", "after-school tutoring", "language courses", "music lessons", "computer training", "parent meetings", "health seminars"], You extract a single numeric value from an answer string. Return only the number and ensure it matches the requested unit.
**Extraction Prompt:** Extract a single numeric value representing the average number of hours per week allocated to {placeholder} in community centers. Only report using 'hours per week'. Return "None" if there is no numerical value.

Answer text: {answer_text}

---

**Question:** Estimate the total annual energy consumption attributable to {placeholder} in public facilities of a typical large city (kWh per year).
**Values:** ["lighting", "heating", "air conditioning", "ventilation systems", "water heating", "elevators", "computers", "security systems", "kitchen appliances", "pumping stations", "outdoor lighting", "refrigeration"], You extract a single numeric value from an answer string. Return only the number and ensure it matches the requested unit.
**Extraction Prompt:** Extract a single numeric value representing energy consumption, using kWh per year as the unit. Return "None" if there is no numerical value.

Answer text: {answer_text}

**Question:** Estimate the total person-hours spent responding to {placeholder} by municipal emergency services per year in a typical large metropolitan area (person-hours per year).
**Values:** ["structure fires", "medical emergencies", "traffic accidents", "hazardous material spills", "water rescues", "wildlife incidents", "natural disasters", "false alarms", "gas leaks", "power outages", "missing persons cases", "active shooter situations"], You extract a single numeric value from an answer string. Return only the number and ensure it matches the requested unit.
**Extraction Prompt:** Extract a single numeric value representing total person-hours, and ensure the unit is 'person-hours per year'. Return "None" if there is no numerical value.

Answer text: {answer_text}

---

**Question:** Estimate the total mass of all {placeholder} currently deployed in public transportation systems of a typical large metropolitan area (tons).
**Values:** ["electric buses", "diesel buses", "tram cars", "subway trains", "hybrid buses", "trolleybuses", "light rail vehicles", "autonomous shuttles", "double-decker buses", "articulated buses", "compressed natural gas buses", "monorail cars"], You extract a single numeric value from an answer string. Return only the number and ensure it matches the requested unit.
**Extraction Prompt:** Extract a single numeric value reported in tons. Return "None" if there is no numerical value.

Answer text: {answer_text}

---

**Question:** What is the average number of {placeholder} provided by local governments per year in a mid-sized European city (services per year)?
**Values:** ["library books", "building permits", "waste collections", "public events", "health inspections", "recycling pickups", "housing grants", "school meals", "parking tickets", "bus routes", "street repairs", "water quality tests"], You extract a single numeric value from an answer string. Return only the number and ensure it matches the requested unit.
**Extraction Prompt:** Extract a single numeric value reported in services per year. Return "None" if there is no numerical value.

Answer text: {answer_text}

---

**Question:** What is the estimated annual growth rate of {placeholder} implemented for urban safety in a typical large city (% per year)?
**Values:** ["surveillance cameras", "facial recognition systems", "emergency alert apps", "crime prediction algorithms", "traffic monitoring sensors", "smart street lighting", "body-worn cameras", "drones for patrols", "gunshot detection systems", "license plate readers", "panic button networks", "public Wi-Fi hotspots"], You extract a single numeric value from an answer string. Return only the number and ensure it matches the requested unit.
**Extraction Prompt:** Extract a single numeric value representing the annual growth rate. The allowed unit is % per year. Return "None" if there is no numerical value.

Answer text: {answer_text}

---

**Question:** What is the total number of {placeholder} currently accessible in all municipal libraries of a typical large city (units)?
**Values:** ["books", "magazines", "newspapers", "audiobooks", "DVDs", "manuscripts", "maps", "journals", "ebooks", "reference guides", "music CDs", "archives"], You extract a single numeric value from an answer string. Return only the number and ensure it matches the requested unit.
**Extraction Prompt:** Extract a single numeric value for quantity and use 'units' as the reporting unit. Return "None" if there is no numerical value.

Answer text: {answer_text}

---

**Question:** What is the elasticity of demand for {placeholder} with respect to price in a typical mid-sized U.S. city (unitless)?
**Values:** ["gasoline", "electricity", "apples", "coffee", "bread", "public transportation", "internet service", "movie tickets", "bottled water", "milk", "restaurant meals", "cigarettes"], You extract a single numeric value from an answer string. Return only the number and ensure it matches the requested unit.
**Extraction Prompt:** Extract a single numeric value representing elasticity, which must be unitless (no units). Return "None" if there is no numerical value.

Answer text: {answer_text}

---

**Question:** What is the estimated annual quantity of {placeholder} required for municipal road maintenance in a typical large city (tons per year)?
**Values:** ["asphalt", "gravel", "salt", "sand", "concrete", "bitumen", "crushed stone", "recycled asphalt pavement", "topsoil", "cement", "road base material", "aggregate"], You extract a single numeric value from an answer string. Return only the number and ensure it matches the requested unit.
**Extraction Prompt:** Extract a single numeric value and report only in 'tons per year'. Return "None" if there is no numerical value.

Answer text: {answer_text}

---

**Question:** What is the average number of hours spent on {placeholder} per month in a typical mid-sized city's social planning department (hours per month)?
**Values:** ["community outreach", "data analysis", "policy drafting", "stakeholder meetings", "public consultations", "report writing", "budget planning", "event coordination", "staff training", "grant applications", "project evaluation", "interdepartmental collaboration"], You extract a single numeric value from an answer string. Return only the number and ensure it matches the requested unit.
**Extraction Prompt:** Extract a single numeric value only. The allowed unit is 'hours per month'. Return "None" if there is no numerical value.

Answer text: {answer_text}

---

**Question:** What is the typical number of hours per week allocated to {placeholder} in municipal youth programs in a large metropolitan area (hours per week)?
**Values:** ["physical education", "arts instruction", "STEM activities", "community service", "leadership training", "sports practice", "mentoring sessions", "health education", "language classes", "career exploration", "environmental projects", "technology workshops"], You extract a single numeric value from an answer string. Return only the number and ensure it matches the requested unit.
**Extraction Prompt:** Extract a single numeric value and ensure the unit reported is 'hours per week'. Return "None" if there is no numerical value.

Answer text: {answer_text}

---

**Question:** What is the average number of public health interventions specifically targeting {placeholder} launched by municipal governments per year in a typical urban area (interventions per year)?
**Values:** ["diabetes", "asthma", "influenza", "obesity", "hypertension", "tuberculosis", "depression", "HIV/AIDS", "malaria", "measles", "dengue fever", "hepatitis"], You extract a single numeric value from an answer string. Return only the number and ensure it matches the requested unit.
**Extraction Prompt:** Extract a single numeric value representing the annual count of interventions, using 'interventions per year' as the unit. Return "None" if there is no numerical value.

Answer text: {answer_text}

---

**Question:** What is the average response time for a {placeholder} reported to social services in a typical metropolitan area (minutes)?
**Values:** ["child abuse case", "domestic violence incident", "elder neglect report", "sexual assault allegation", "runaway youth report", "human trafficking tip", "mental health crisis", "drug overdose

call", "missing person report", "animal cruelty complaint", "suicide threat alert", "youth truancy notification"], You extract a single numeric value from an answer string. Return only the number and ensure it matches the requested unit.

**Extraction Prompt:** Extract a single numeric value representing average response time. The only allowed unit is minutes. Return "None" if there is no numerical value.

Answer text: {answer_text}

---

**Question:** What is the average annual salary paid to {placeholder} working in municipal planning departments in a typical U.S. city (dollars per year)?
**Values:** ["urban planners", "civil engineers", "GIS specialists", "zoning inspectors", "transportation analysts", "environmental planners", "land use planners", "city managers", "planning technicians", "historic preservationists", "community development coordinators", "housing analysts"], You extract a single numeric value from an answer string. Return only the number and ensure it matches the requested unit.
**Extraction Prompt:** Extract a single numeric value representing the average annual salary and ensure the unit is dollars per year. Return "None" if there is no numerical value.

Answer text: {answer_text}

---

**Question:** What is the total amount of {placeholder} distributed by social welfare agencies per year in a typical large city (kilograms per year)?
**Values:** ["food", "rice", "flour", "sugar", "vegetables", "meat", "bread", "milk powder", "canned goods", "lentils", "clothing", "diapers"], You extract a single numeric value from an answer string. Return only the number and ensure it matches the requested unit.
**Extraction Prompt:** Extract a single numeric value reported in kilograms per year. Return "None" if there is no numerical value.

Answer text: {answer_text}

---

**Question:** Estimate the total annual revenue generated from {placeholder} offered by local governments in a typical large metropolitan area (dollars per year).
**Values:** ["municipal bonds", "tax-exempt loans", "public pension funds", "lottery tickets", "parking permits", "business licenses", "building permits", "property tax collections", "transit passes", "utility bills", "court fines", "development fees"], You extract a single numeric value from an answer string. Return only the number and ensure it matches the requested unit.
**Extraction Prompt:** Extract a single numeric value reported in dollars per year. Return "None" if there is no numerical value.

Answer text: {answer_text}

---

**Question:** Estimate the total number of {placeholder} processed by city housing departments per year in a typical medium-sized U.S. city (applications per year).
**Values:** ["rental applications", "permit applications", "eviction filings", "subsidy requests", "complaint forms", "inspection requests", "appeals", "zoning applications", "lease renewals", "housing vouchers", "building code violations", "affordable housing applications"], You extract a single numeric value from an answer string. Return only the number and ensure it matches the requested unit.
**Extraction Prompt:** Extract a single numeric value for the total annual applications, using 'applications per year' as the unit. Return "None" if there is no numerical value.

Answer text: {answer_text}

---

**Question:** What is the average maintenance cost of {placeholder} for municipal infrastructure per year in a typical large city (dollars per year)?
**Values:** ["road resurfacing", "stormwater management", "waste collection", "street lighting", "bridge repairs", "sidewalk upkeep", "traffic signal maintenance", "park landscaping", "public transit facilities", "sewer system repairs", "snow removal operations", "drainage system cleaning"], You extract

a single numeric value from an answer string. Return only the number and ensure it matches the requested unit.
**Extraction Prompt:** Extract a single numeric value representing the average maintenance cost, reported in dollars per year. Return "None" if there is no numerical value.

Answer text: {answer_text}

---

**Question:** On average, how many trips by {placeholder} are made for community events per month in a typical mid-sized city (trips per month)?
**Values:** ["bus", "taxi", "rideshare", "bicycle", "scooter", "carpool", "minivan", "shuttle", "motorcycle", "vanpool", "tram", "light rail"], You extract a single numeric value from an answer string. Return only the number and ensure it matches the requested unit.
**Extraction Prompt:** Extract a single numeric value representing the number of trips by the specified vehicle/mode per month, using 'trips per month' as the unit. Return "None" if there is no numerical value.

Answer text: {answer_text}

---

TRANSPORTATION

**Question:** What is the average number of incidents of {placeholder} reported per 1,000 train journeys worldwide each year?
**Values:** ["signal failure", "track obstruction", "door malfunction", "brake failure", "engine overheating", "derailment", "power outage", "passenger injury", "collision", "vandalism", "fire outbreak", "communication breakdown"], You extract a single numeric value from an answer string. Return only the number and ensure it matches the requested unit.
**Extraction Prompt:** Extract a single numeric value representing the average number of incidents per 1,000 train journeys per year. The allowed unit is 'incidents per 1,000 train journeys per year'. Return "None" if there is no numerical value.

Answer text: {answer_text}

---

**Question:** How much is spent on {placeholder} for public transportation in the United States per year (in dollars per year)?
**Values:** ["fuel", "maintenance", "labor", "insurance", "vehicles", "infrastructure", "security", "cleaning", "technology upgrades", "administration", "marketing", "energy"], You extract a single numeric value from an answer string. Return only the number and ensure it matches the requested unit.
**Extraction Prompt:** Extract a single numeric value and ensure the unit is 'dollars per year'. Return "None" if there is no numerical value.

Answer text: {answer_text}

---

**Question:** Approximately what percentage of annual global freight is transported using {placeholder} each year (% per year)?
**Values:** ["container ships", "railroads", "air cargo", "trucks", "bulk carriers", "pipelines", "coastal shipping", "river barges", "automated guided vehicles", "drones", "roll-on/roll-off vessels", "intermodal transport"], You extract a single numeric value from an answer string. Return only the number and ensure it matches the requested unit.
**Extraction Prompt:** Extract a single numeric value representing a percentage, with unit '% per year'. Return "None" if there is no numerical value.

Answer text: {answer_text}

---

**Question:** Estimate the average fuel consumption for {placeholder} in city traffic, measured in liters per 100 kilometers.
**Values:** ["rush hour", "weekday mornings", "weekend evenings", "holiday season", "summer months",

"winter conditions", "rainy days", "snowy periods", "peak traffic hours", "nighttime driving", "school drop-off hours", "festive weekends"], You extract a single numeric value from an answer string. Return only the number and ensure it matches the requested unit.

**Extraction Prompt:** Extract a single numeric value representing fuel consumption, reported in liters per 100 kilometers. Return "None" if there is no numerical value.

Answer text: {answer_text}

## K  EXAMPLE OF MODEL REWRITTEN PROMPTS

| Paraphrase | Original (with placeholder) |
| --- | --- |
| What is the annual growth rate of revenues from freight forwarding in the logistics sector, measured as a percentage per year? | What is the annual growth rate of revenue from freight forwarding in the logistics sector, measured in percent per year? |
| What is the average monthly income (in US dollars) that an ordinary household in the United States receives from rental properties? | What is the average monthly income earned from rental properties by a typical household in the United States (dollars per month)? |
| What is the probability that regional warehouse will face a significant disruption in logistics services during a given year, measured as an annual percentage? | What is the probability that a regional warehouse will experience a major logistics disruption in a given year, measured in percent per year? |
| What is the average number of public health interventions specifically related to diabetes that are launched annually by municipal governments in a typical urban region (number of interventions per year)? | What is the average number of public health interventions specifically targeting diabetes launched by municipal governments per year in a typical urban area (interventions per year)? |
| What is the annual percentage of warehouse operators who experience lower back pain due to repetitive lifting work? (Unit: annual percentage) | What percentage of warehouse workers experience lower back pain each year due to repetitive lifting? (units: percent per year) |
| Estimate the average annual interest rate (percent) for the working capital loans used by logistics companies. | Estimate the average interest rate charged for working capital loans used by logistics companies in percent per year. |
| What is the approximate average distance (in kilometers) that migratory bird moves within a year? | What is the average distance (in kilometers) that a migratory bird travels in one year? |
| What is the average number of public buses running daily in city? (Number of buses in one day) | What is the average number of public buses operating in a typical city per day? (buses per day) |
| What is the average number of apples boxes delivered to supermarkets in New York City every day? (Unit: boxes/day) | What is the average number of crates of apples delivered to supermarkets in New York City per day? (units: crates per day) |
| Please estimate the annual total energy consumption of all MRI machines in U.S. hospitals in units of kilowatt-hours per year. | Estimate the total annual energy usage of all MRI machines in U.S. hospitals, measured in kilowatt-hours per year. |
| What are the average annual maintenance costs for road resurfacing (USD/year) in the municipal infrastructure of a typical large city? | What is the average maintenance cost of road resurfacing for municipal infrastructure per year in a typical large city (dollars per year)? |
| What is the average number of meals prepared per month by each family using weekdays? (Unit: meals per month) | What is the average number of meals prepared using weekdays per household per month? (unit: meals per month) |

