# OpenReview forum: "Measuring Intent Comprehension in LLMs: A Variance Decomposition Framework"
_ICLR.cc/2026/Conference — Submitted to ICLR 2026_

### Official Review · Reviewer_GktL · 2025-10-20

**Soundness:** 3
**Presentation:** 2
**Contribution:** 3
**Rating:** 4
**Confidence:** 4

**Summary:**

This work proposed a method to evaluate if LLMs truly understand users' intents. It models how LLMs understand users' intent by analyzing how their responses vary across prompts with different underlying intents. The experiment results on open-source LLMs show that larger models have stronger intent understanding capability.

**Strengths:**

I really like the idea of modeling LLMs' intent understanding ability by measuring how robust their responses are across different prompts with varying intents behind them. This perspective is inspiring. And this works decouples model's response changes into three factors:
1. The model’s inherent uncertainty
2. The diversity of prompts
3. The variation in intents

And this work proposes a quantitative method to quantify the impact of each factor.

**Weaknesses:**

1. **Assumption of intent-response mapping**. This work assumes that prompts with different intents necessarily yield different responses. However, this assumption overlooks cases where distinct intents can share the same surface form or prompt (*false negatives*). Such cases are ignored in the proposed evaluation framework.
2. **Restrictions to Numeric Responses**: The framework appears to support only numerical responses, where the output of the model is a number. This design limits its applicability and generalizability to more diverse response forms, such as long text.
3. **Limited model coverage**: The evaluation only includes LLaMA and Gemma but does not examine proprietary models such as GPT or Gemini, which could provide more valuable and interesting insights given their popularity.
4. **Temperatur Design**: The authors state in the appendix that all models are tested with a temperature of 1. However, for LLaMA's temperature range is [0, 1], meaning that a value of 1 leads to extremely diverse responses (MU is high). In contrast, Gemma’s temperature range is [0, 2], where 1 represents moderate diversity (MU is lower). This discrepancy makes the comparison between the models seem unfair.

**Questions:**

1. I’d like to know how you would handle cases where different intents share similar or even identical prompts. How might such cases be incorporated into your current assumptions and framework?
2. In lines 320–323, I found it difficult to fully understand the recall formula defined as IS / (IS + AS + MU). While the definition of precision is clear, the recall formula seems to omit false negatives (my weakness #1). Could you please provide a clearer explanation or justification for this formulation?
3. In Figure 4, the meaning of the different colors is unclear. It would be helpful to include a brief description or legend in the caption to clarify their significance.
4. I’d like to understand why all models were set to a temperature of 1. Do you think this ensures a fair comparison?

---

> ### Author Response · Authors · 2025-11-20
>
> We thank the reviewer for the positive assessment of our idea and for the detailed, constructive comments. We are especially glad that you found the framework interesting! We address the weaknesses and questions together in this and the next comment, as they are intertwined in your review.
>
> 1. **Assumption of intent–response mapping** In our framework, intent is defined via the evaluator as the expected distribution over values (Remark 2). Two prompts are treated as the same intent if they are meant to induce the same response distribution, even if their surface descriptions or “stories” differ. In that sense, cases where two different semantic set-ups share the same desired distribution are not separate intents in our evaluation; they are intentionally collapsed into a single intent label (\tilde{\tau}).
>
>    The scenario you raise - distinct intents sharing the same surface form or prompt—is different. Here, the same prompt string can correspond to multiple latent user goals. For example, a user might send “Can you help me?” with different intentions in mind. First, we want to emphasize that we think of the prompt as the entire model input, not just the final user last string but the whole context available to the model. In a dialogue setting, this includes the conversation history and system instructions. Two occurrences of the text “Can you help me?” that appear in different contexts (e.g., after a discussion of travel planning versus after a discussion of code debugging) are thus different prompts and can be assigned different intents.
>
>     The truly ambiguous case arises only when both the surface string and the observable context are identical, yet the user’s latent goal is different. No model (or evaluator) that conditions only on the visible input can, in principle, distinguish such intents. If these situations are common in the data, the model may respond with a non-degenerate distribution over plausible answers. In our framework, this residual variability is exactly what is captured by the Model Uncertainty component (MU): it reflects genuine ambiguity that cannot be resolved from the observable prompt. Our experiments deliberately focus on structured settings where the evaluator’s notion of intent is encoded in the observable prompt (template, slots, paraphrases). We will clarify in the paper that “prompt” refers to the full input the model sees, and that genuinely latent variation in user goals beyond that input lies outside the scope of this first study; handling those cases would require extending the framework to allow a distribution over intents per prompt. We will add a short discussion of these potential “false negative” cases in the Discussion section.
>
> 2. **“Restriction” to numeric responses.** Our main experiments use tasks with naturally scalar responses (probabilities, prices, travel times, etc.) because these allow a clean variance decomposition and fit well with guesstimation-style evaluation. But the framework itself is not limited to numeric outputs or to variance. What the framework requires is simply the ability to track how changes in intent and articulation shift the **pushforward response distribution** V(a). Any discrepancy measure over distributions (e.g total variation distance) could be used in place of variance, albeit with greater computational and expository cost than our IS/AS/MU implementation.
>
>     For **discrete outputs** (e.g., categorical labels or ratings), we already provide an entropy-based analogue in the appendix, replacing variance with entropy and expressing the decomposition in terms of mutual information. The conceptual structure directly parallels the variance case.
>
>     For **long free-form text**, the framework can be composed with a task-specific value extractor—rubric-based scoring, sentiment classification, scalar ratings, an evaluator model mapping text to categories or scores, or embedding-based summaries. The decomposition then applies to the evaluator’s output rather than the raw text. In practice, even for open-ended text, evaluators usually target only a few meaningful dimensions.
>
>    We will make this generality more explicit in the main text by: (i) emphasizing that the framework is defined at the level of value distributions, (ii) bringing the entropy-based version from the appendix into the main narrative, and (iii) adding a discussion of how free-form text can be handled through evaluator-induced values. Our use of numeric tasks is thus a deliberate focus for this first study - not a fundamental limitation of the framework.

---

> > ### Author Response · Authors · 2025-11-20
> >
> > 3. **Limited model coverage (no GPT/Gemini)** We agree that including proprietary models such as GPT or Gemini would be of high interest, particularly given their widespread deployment. Our primary reasons for focusing on LLaMA and Gemma were reproducibility and openness, and cost and stability. All the models we evaluate are open-weight, so the entire pipeline (data, code, models) can be reproduced, extended, and re-analyzed by other researchers without API access or license restrictions. Moreover, our framework requires many samples per intent–prompt cell across temperatures; running this at scale on commercial APIs is substantially more expensive and can be sensitive to undocumented changes.
> > At the same time, the framework is model-agnostic and can be applied directly to GPT, Gemini, and other APIs when available. We will release our code so that practitioners can easily plug in their own models.
> >
> > 4. **Temperature design** - Thank you for raising this point about temperature and cross-model comparability. In all our experiments, “temperature” refers to the usual logit rescaling before the softmax, so that we sample from ($\text{softmax}(z/T)$); in this convention, (T = 1) corresponds to the model’s base predictive distribution. The fact that different APIs expose different nominal ranges (e.g. ([0,1]) vs ([0,2])) is an interface choice and does not change this interpretation, though it can influence what is viewed as a “typical” operating point in practice.
> >
> >     Our rationale for using (T=1) for all models was to place them in a single, untuned “natural sampling” regime close to their default behavior, and to avoid model-by-model hyperparameter tuning. Our framework itself is behavioral: for any fixed decoding configuration, we treat the resulting generative behavior as a stochastic policy and analyze its variance decomposition. In that sense, different temperatures simply correspond to different variants of a model with potentially different IS/AS/MU profiles.
> >
> >    In the revision, we will (i) make this rationale explicit in the main text, emphasizing that (T=1) is used as a common decoding regime rather than as a claim of perfectly normalized diversity across families; and (ii) extend the appendix with results for additional temperature settings to show that our qualitative conclusions about intent sensitivity versus articulation sensitivity are robust across a reasonable range of decoding regimes.
> >
> > 5. **“Recall” formula IS / (IS + AS + MU)** Your question about the “recall” expression points to an important clarification. Our use of “precision” and “recall” is analogical rather than literal classification precision and recall. The precision-like quantity asks: of all the systematic variability that responds either to intent or to articulation, what fraction is due to intent? High precision means variability is not “wasted” on prompt wording. The recall-like quantity ($\text{IS} / (\text{IS} + \text{AS} + \text{MU})$) asks: of all observed variability in model responses, what fraction is actually driven by intent? This mirrors the idea of recall as the share of relevant items within all items, but here the “items” are units of variance. Any variability not attributed to intent, i.e., AS + MU—acts as the analogue of “false negatives” in that it represents variation that is not aligned with changes in intent.
> >
> >     We agree that this analogy can be confusing, especially when thinking about false negatives as “distinct intents that happen to share outputs.” In our framework, such undetected intent differences are precisely those that fail to generate additional IS; they are, by construction, in the denominator and thus lower the recall-like ratio. To avoid confusion, we will make clear in the text that these are “precision-like” and “recall-like” quantities and give a concise explanation of how the variance decomposition maps to the standard TP/FP/FN picture.
> >
> > 6. Figure 4 colors - In the revised version we will include a clear legend and caption text.
> >
> > We appreciate the reviewer’s encouraging overall assessment and the concrete suggestions and questions. We believe that the clarifications and extensions outlined above will substantially improve both the soundness and the presentation of the work.

---

> > > ### Comment · Reviewer_GktL · 2025-11-22
> > > **Response by Reviewer**
> > >
> > > Thank you for your clarifications on my questions. I have carefully read your responses.
> > >
> > > **Assumption of intent–response mapping**
> > >
> > > Thanks for making the prompt thing clear. My new understanding based on your response is that, the model can see the entire input (the context + final user query), thus it is unlikely that same input deliver ambigious intents. Then, my concern is addressed. So in your hypothesis, the model can accurately capture the intent from the entire input and generate corresponding output. The intent-output mapping is clear to me now.
> > >
> > > It would be good to add some description on the main text that the model can see the entire input, including prior context.
> > >
> > > **“Restriction” to numeric responses.**
> > >
> > > I appreciate your proposed approaches to map non-numeric outputs (discrete outputs, long free-form text) to numeric ones. Probably I missed some places in the main pages where you actually described it. It'd be good if you could highlight your proposed approaches to show the versatility of your framework. Also, adding an experiment to showcase how to apply to long free-form text outputs is worth-trying and improving the framework's applicability.
> > >
> > > **Temperature design**
> > >
> > > Thanks for the explanation on the temperature. I think setting temperature=1 makes sense if you aim to use the original logits  distribution in sampling. Just from the first glance, setting temperature=1 which makes LLaMA give extremly diverse outputs but makes Gemma give moderately diverse outputs seems an unfair comparison (usually people just use the default temperatures). You could elaborate on the motivation of the temperature setting in the experiment.
> > >
> > > Given your clarifications, my concerns are addressed and I have increased my OA score.

---

> > > > ### Author Response · Authors · 2025-12-02
> > > >
> > > > We thank you for raising our scores and for engaging thoughtfully with our paper. As suggested, we have added clarification of the prompt in lines 124–125.
> > > >
> > > > We have also strengthened the treatment of non-numeric outputs. Specifically, we extended the main-text discussion (lines 340–360) on how to handle non-numeric values and expanded Appendix A to illustrate how our approach can be applied to free-text responses and discrete outputs. In addition, we implemented and reported a proof-of-concept extension to discrete outputs using two credit-decision tasks. In the first task, the model acts as a bank underwriter and produces a binary loan-approval decision; in the second, it outputs a five-level credit-risk rating. Each task uses 12 applicant profiles as intents, defined by realistic financial attributes such as income, debt, employment history, credit record, and loan terms. These profiles are held fixed, while articulation varies through paraphrasing generated and filtered via multilingual back-translation. For every model, we collect 25 valid outputs for each of the 120 intent–paraphrase conditions, yielding 6,000 samples per model. We then apply an information-theoretic analogue of our decomposition, as detailed in Appendix A. This demonstration shows that our framework naturally extends beyond numerical outputs and remains interpretable and diagnostic in discrete decision settings.
> > > >
> > > > Finally, we have clarified our choice of temperature.
> > > >
> > > > Thank you again for your careful and constructive feedback.

---

### Official Review · Reviewer_bj5p · 2025-10-27

**Soundness:** 4
**Presentation:** 3
**Contribution:** 3
**Rating:** 6
**Confidence:** 3

**Summary:**

This paper proposes a formal definition of intent comprehension in LLMs, which is the desired property that LLMs are sensitive to the *intent* of user requests (Intent Sensitivity; IS) without being overly sensitive to superficial differences in prompt phrasing (Articulation Sensitivity; AS). The third source of variation in model responses is termed model uncertainty (MU). The authors quantify these three concepts by using a variance decomposition of model responses and offer two descriptive metrics: (1) Meaningful Variance Share (MVS) that reflects the "precision" or the ratio of explainable variance, and (2) Intent Comprehension Index (ICI) which captures the "F1-score" i.e. the harmonic mean between the MVS and the IS. To evaluate this in practice, a set of 24 prompts with 10 intent-preserving paraphrases each was generated using GPT-4.1. Five open source LLMs were evaluated across Gemma and Llama families with varying sizes, generating 25 responses per paraphrased prompt. Findings indicate generally that IS increases with model size (as expected), but so does AS, which is not desirable.

**Strengths:**

- Paper is extremely well written, clear, and well-motivated. I especially found the conceptual framework very clear, the relaxation of intent comprehension to be realistic, and the explanation of the different metrics to be exceptional.
    - Small suggestion, the explanation of the MVS and ICI in lines 320--323 cleared a lot up for me, and I think it would make the whole section easier to follow if it was moved up earlier. Generally maybe splitting Section 3 with subsection headings could also make it even better.
- It addresses an important notion--intent comprehension--that is fundamental to understanding how LLMs behave and can provide insight into improving capabilities.
- The variance decomposition approach is well justified with respect to related literature. I didn't check the math, but the explanation seems very intuitive and well-defined.

**Weaknesses:**

1. The paper lacks details on the prompt generation and the validation of the LLM-generated prompts and paraphrases. In order to make sure that the intent of each prompt is maintained across paraphrases, a human validation would be necessary.
2. Generally, the scope of the contributions is limited to just measuring the phenomenon and there is no investigation into why this occurs or how to improve performance. If the authors could do one or both of the below, it would greatly improve the scope of the contributions and utility of the paper insights:
    1. The ability of a model to understand user intent from prompts is likely heavily influenced by the post-training, i.e. the instruction finetuning and/or alignment finetuning. It would be really insightful to include experiments comparing base versus instruction tuned versions of each model (most if not all of the models in the paper have their counterparts open source). If full scale experiments are not feasible, at least a discussion on the differences in the post-training of Llama vs Gemma families and how this might affect the results could be helpful.
    2. Another way to improve the contribution is maybe to finetune any of the models on the prompt variants to see if this improves the IS scores and decreases the AS.
3. There are lots of problems with the presentation of results and figures in this paper.
    1. Figure 1 is very tiny and the text is not readable.
    2. Figure 2, the radar chart is a poor choice for visualizing the different models since the lines all overlap and the three metrics are related, not distinct. Consider switching to a stacked bar plot, which is clearer and better suited for the three metrics which all sum to 1.
    3. Figure 3, the line chart is also not a great choice here since the x-axis are categorical and not continuous, which makes it confusing to interpret. Please switch to a more appropriate plot, such as a grouped bar plot. Also, it is missing a y-axis label.
    4. Figure 4 is missing a legend, making it impossible to know which colours correspond to which metric. I'd suggest adding the legend onto the chart as well as mentioning in the figure caption.

**Questions:**

## Questions
1. I'm curious about the translation approach to generating paraphrases. Was this also done with GPT 4.1? Were the paraphrases validated in any way by humans to ensure the intent was preserved?
2. Why in the consistency definition 2.1 do you write $p_1,p_j\in\tau^{-1}(T)$ instead of $\tau(p_i)=\tau(p_j)$? The latter is clearer in my opinion and more consistent with the definition 2.2.
3. In line 145, why is the "response distribution $\pi$ over $\mathcal{A}$"? Isn't a model's output distribution usually over the input? The notation is bit confusing.

## Suggestions and typos
- The discussion mentions that the method can be extended beyond numerical outputs, which seems like it would be really helpful in practice! However, I did not check this since it's fully in the appendix and not in the main paper. Perhaps it could be moved up into the main paper if it does work.
- Throughout the paper, textual citations are used when they should actually be parenthetical citations. Please correct these (i.e. make sure to use `\citep{}`)
- Citations to all models are missing (Section 4)
- Line 075: "the 70B model..." doesn't make sense out of context since no model sizes have been mentioned yet. Perhaps "the largest model we tested"
-  Line 282 and 319 contain extra "."s
- Line 351 "GPT-based" is this GPT 4.1? It's important to be precise.
- Line 356 "section" needs to be "Section"
- Line 365 "figure" needs to be "Figure"

---

> ### Author Response · Authors · 2025-11-20
>
> We thank the reviewer for the thoughtful and positive assessment. We respond to the main points below.
>
> **Weaknesses**
> 1. Prompt generation and human validation - All prompts and paraphrases are generated with GPT-4.1. We first design templated scenarios with explicit slots (e.g., distances, prices, probabilities) that define the intent. For each scenario, we generate an initial English prompt, then produce paraphrases via cross-lingual translation (translate through multiple languages and back) and additional paraphrasing prompts. We use LLM to filter out paraphrases that change key content. We will describe this pipeline in more detail in the Appendix. We agree that human validation is important and will add a human evaluation on a subset of (prompt, paraphrase) pairs in Section 4, reporting how often annotators judge the paraphrases as intent-preserving.
>
> 2. Scope: only measuring, not explaining or improving -  We appreciate the suggestions regarding base vs. instruction-tuned models and finetuning on prompt variants. A full set of such experiments is beyond what we can realistically add before the deadline, but we fully agree these are natural and important next steps. In the revised version we will make this more explicit in the Discussion, and explain concretely how our metrics could be used as training objectives or diagnostics. We also add a discussion on how differences across models families might affect the the results to the results section. We see our framework as a tool for targeted improvement of intent comprehension, not only as a passive measurement device, and we will make this clearer in the text.
>
> 3. Figures and result presentation - Thank you for the detailed comments. We have enlarged Figure 1 to make all text readable, replaced the radar chart and line chart with more appropriate bar-style plots and added missing axis labels, and added a legend and clear caption for Figure 4. We believe these changes significantly improve readability.
>
> **Questions**
>
>  1. Yes, all paraphrase generation (including translation and back-translation) is done with GPT-4.1 as described above, and we will now add human validation results on a subset.
> 2. We agree with the suggested change in consistency notation and will adopt the clearer version, which is also more consistent with Definition 2.2.
> 3 – Regarding the “response distribution over A”: the model’s raw output distribution is over token sequences (A). In Definition 2.1 (intent comprehension) we work directly with this token-level distribution. For practical and interpretability reasons, Definition 2.2 (sufficient intent comprehension) instead uses the induced distribution over evaluator outputs (V(\text{response})), where (V) maps the output text to a scalar or discrete label. We will make this distinction between the token-level and evaluator-level distributions more explicit in the paper.
> 3. Extension beyond numerical outputs -  We agree this extension is important in practice. We will bring a concise version of the non-numeric / discrete extension (currently in the Appendix) into the main text, and we will also add a short discussion of how long free-form text responses can be analyzed via an evaluator that maps text to a scalar or discrete outcome, as suggested by multiple reviewers.
>
> **Citations and minor issues** - We have corrected all the comments in the revision - thanks for this! this is very helpful!
>
> Once again, we thank the reviewer for the encouraging review and detailed feedback. Your suggestions makes the paper substantially clearer and more useful.

---

> > ### Comment · Reviewer_bj5p · 2025-11-20
> >
> > Thank you for the clarifications. I acknowledge the resource/time constraints, but in order for me to justify raising my score, I'd need you to actually show me some evidence of the new discussion/human evaluation instead of just promising to add them abstractly. You can either upload a new revised PDF with the promised additions (and let me know which line numbers correspond to each point), or show me a concrete draft here.
> >
> > Specifically:
> > 1. You promised to *"add a human evaluation on a subset of (prompt, paraphrase) pairs in Section 4, reporting how often annotators judge the paraphrases as intent-preserving."* **Could you please provide the details: number of annotators, number of prompt/paraphrase pairs rated by each annotator, inter-annotator agreement, as well as the overall proportion of times the annotators judged them as accurately intent-preserving?** If the full validation will not be complete before the end of the discussion period, intermediate results (+ if/how you expect it to change after it's complete) can suffice.
> > 2. Above, you said, *"We also add a discussion on how differences across models families might affect the the results to the results section. We see our framework as a tool for targeted improvement of intent comprehension, not only as a passive measurement device, and we will make this clearer in the text."* Could you either point me to these new discussion points in the revised PDF, or include concrete draft of it here?
> > 3. *"Extension beyond numerical outputs... we will also add a short discussion of how long free-form text responses can be analyzed via an evaluator that maps text to a scalar or discrete outcome, as suggested by multiple reviewers."* Same request as above. **After reading the other reviewers' comments, I strongly agree that the contribution would be much more valuable/realistically useful if the method generalizes to non-numeric outputs. It would be especially helpful to see any empirical evidence demonstrating that it would actually work (eg lightweight proof-of-concept experiments on 1-2 examples that can be manually inspected, etc).**
> >
> > Ideally you can respond to all, but prioritize #1 and #3 since at this point, it's probably more important to demonstrate that the method is sound and generalizes. Thank you!

---

> > > ### Author Response · Authors · 2025-12-02
> > >
> > > Thank you for the continued engagement. As requested, we implemented all three additions directly in the revised paper. We believe these changes materially strengthen the work.
> > >
> > > For the first point, we now include a full human evaluation validating both paraphrase intent-preservation and numeric value extraction. We sampled 200 prompt–paraphrase pairs and had two independent annotators judge whether each paraphrase preserved the underlying intent. Annotator 1 labeled 95% as intent-preserving and Annotator 2 labeled 91% as such, with 89% raw agreement. At least one annotator judged the paraphrase intent-preserving in 98.5% of cases. Separately, we validated the reliability of the numeric-value extractor on 200 model responses, again using two annotators. Annotator 1 judged 94% of extractions correct and Annotator 2 judged 91% correct, with 93% raw agreement and κ≈0.50. These results support the core assumptions underlying our decomposition: the paraphrases overwhelmingly preserve intent, and evaluator extraction error is small and well behaved.
> > >
> > > For the second point, we added a short paragraph discussing differences in training and data across the model families and how these may affect the evaluation results (lines 459–468).
> > >
> > > For the third point, we did three things. First, we extend our discussion in this in the main text (lines 340-360) on how to deal with non numeric values. We also extended appendix A with discussion on how our approach can be used for free-text responses and discrete outputs. Finally, we implemented and reported a proof-of-concept extension to discrete outputs using two credit-decision tasks. In the first task, the model acts as a bank underwriter and outputs a binary loan-approval decision; in the second, it outputs a five-level credit-risk rating. Each task uses 12 applicant profiles as intents, each defined by realistic financial attributes such as income, debt, employment history, credit record, and loan terms. These profiles remain fixed, while the articulation is varied by generating and filtering paraphrased versions of the task instructions via multilingual back-translation. For every model, we collect 25 valid outputs for each of the 120 intent–paraphrase conditions, producing 6,000 samples per model. We apply an information-theoretic analogue of our decomposition discussed in appendix A. The results mirror the numerical experiments: smaller LLaMA models show high residual uncertainty and weak intent responsiveness; LLaMA-70B shows strong intent sensitivity but noticeable framing effects; and Gemma models exhibit stable, intent-driven behavior with relatively low articulation sensitivity. This demonstration shows that our framework naturally extends beyond numerical outputs and remains interpretable and diagnostic in discrete decision settings.
> > >
> > > Thanks again for engaging with our paper. We believe the revised version is much improved and better highlights the appeal of our framework and measure across a wide range of cases.

---

### Official Review · Reviewer_44rZ · 2025-10-31

**Soundness:** 2
**Presentation:** 1
**Contribution:** 2
**Rating:** 2
**Confidence:** 4

**Summary:**

This paper introduces a framework to measure how well LLMs understand user intent beyond surface phrasing. Using a variance decomposition method to separate the effects of intent, phrasing, and uncertainty, the authors evaluate LLMs with cross-lingual paraphrases. Results show that larger models better capture intent but remain sensitive to linguistic variation, revealing trade-offs between semantic understanding and surface-level robustness.

**Strengths:**

1. The paper moves beyond correctness-based evaluation toward consistency-based evaluation, providing a new perspective for analyzing model understanding.

**Weaknesses:**

1. The framework depends on an idealized ground-truth intent mapping (τ(p)), which is rarely available in real-world data. This makes it difficult to apply at scale or to open-ended user interactions.
2. The decomposition metrics (IS, AS, MU, ICI) are mathematically sound but conceptually dense, making them less intuitive for practitioners. The interpretability of results beyond "higher ICI = better intent understanding" is limited.
3. Although the paper aims to separate intent from articulation, in practice, this distinction is often ambiguous. The evaluation might artificially enforce boundaries that do not reflect real human communication variability.
4. The “Sufficient Intent Comprehension” definition relies on a human or heuristic evaluator, introducing subjectivity and limiting full automation. Different evaluators could yield inconsistent measurements.

**Questions:**

1. The variance decomposition relies on estimating response distributions. How sensitive are IS, AS, and MU to sampling noise, prompt diversity, or domain imbalance?
2. The "sufficient intent comprehension" definition depends on an evaluator $\tilde{\tau}, V$. How robust are the findings to the choice or bias of this evaluator?
3. Since the focus is on consistency rather than correctness, how should one interpret a model that is consistently wrong but semantically coherent?

---

> ### Author Response · Authors · 2025-11-20
>
> We thank you for the thoughtful comments and for recognizing our attempt to move beyond correctness-based evaluation toward consistency-based evaluation of intent understanding. We address weaknesses here and questions in the next comment.
>
> 1. Our goal is to formalize a notion of intent that is already implicit in most evaluation setups: when experimenters design tasks, they implicitly specify the latent variable the model should condition on. $\tau(p)$ simply makes this design choice explicit rather than adding a new requirement beyond standard benchmark construction.
>
>     We also agree that separating intent from textual articulation “in the wild” is difficult—this is precisely why our approach is useful. In typical datasets, intent and phrasing are intertwined, making it hard to tell whether a model is responding to the underlying question or merely to surface-level patterns.
>
>     More so, we believe that our framework and experimental design are also directly applicable to real-world settings. For instance, a company logging user queries to an AI chatbot could use it to measure robustness. A bank, for example, could cluster user questions by theme (loan, mortgages, etc.), treat variation within each cluster (interest rate and so on) as distinct questions, apply cross-language paraphrasing to generate alternative phrasings, and then use our decomposition to evaluate how whether the system is responding to users intent.
>
>     Finally, we agree that fully open-ended dialogue without task labels is more challenging. However, free-form responses pose difficulties for empirical measurement, not for our framework itself. In the extreme case—when evaluators do not even know which aspects of the text matter—they can still compare distributions over token sequences. More commonly, evaluators can extract embeddings or annotations (effectively defining V(a)) for the analysis.
>
> 2. We agree that the current presentation is too mathematically oriented. The decomposition is intended to mirror a simple story: IS measures how much responses move when intent changes (holding articulation fixed), AS measures how much they move when wording changes (holding intent fixed), and MU captures residual variability unexplained by either. ICI is high only when a large share of variability is due to intent and articulation and noise contribute little. We believe this narrative is intuitive for practitioners and highlights why the decomposition is useful. In the revision we will explicitly add this short summary in the main text.
>
> 3. We agree that in natural communication the line between “intent” and “articulation” is not always sharp in the wild. In most training data, intent and articulation induce spurious correlations, because small stylistic changes often co-occur with shifts in latent attributes of the writer, even when the underlying objective remains the same. This makes prompt phrasing a confounded signal that mixes true intent with incidental articulation. Our framework does not claim that the world comes pre-partitioned; instead, it makes this explicit by requiring the evaluation designer to specify which differences reflect a true change in intent and which count as articulation.. In our experiments we focus on settings where this distinction is relatively clear. Intents differ by normative targets that should lead to different numerical answers under reasonable judgment. Articulations are generated by paraphrasing/back-translation of the same template and slot values, and filtered so that human raters agree they express the same underlying request. In that sense, we are not imposing an arbitrary separation but explicitly designing a benchmark to isolate whether the model responds to the latent intent rather than to superficial phrasing. Our benchmark is akin to an experiment designed to identify relationships that are hard to separate in the wild. We will clarify in the Discussion section that our framework is best suited to domains where such a normative separation is at least approximately well-defined.
>
> 4. We agree that any measure of whether the model “ought” to change its answer with intent must ultimately be grounded in some external standard—human-annotated, heuristic, or task-specific. Our framework makes this dependence explicit via the evaluator (\tilde{\tau}), instead of leaving it implicit as in most benchmarks, where a “gold label” is simply taken as given. We also agree that different evaluation rules can produce different numerical values, and we will acknowledge this explicitly in the text. We see this as a feature rather than a flaw: the framework is flexible enough to plug in domain-appropriate evaluators (domain experts, crowdsourced labels, task-specific rules), while the decomposition itself remains the same. The dependence on (\tilde{\tau}) is no stronger than in standard supervised evaluation; it is just made transparent.

---

> > ### Author Response · Authors · 2025-11-20
> >
> > With regards to the questions
> >
> > 1. About sensitivity to sampling noise - We report bootstrap standard errors for all the figures. We acknowledge that noise can affect the decomposition. Specifically, using the naive sample analoge variance decomposition can be biased by noise in the estimated means.  To address this directly, we use an ANOVA model that “borrows information” across prompts to reduce the effect of sample noise, rather than relying only on noisy cell-wise sample variances.  We will add to the estimation appendix discussion explaining explicitly. We will also state explicitly in Section 3 that, as with any variance-based method, precision improves with the number of samples and with paraphrase diversity.
> >
> > 2. As noted above, evaluator choice matters conceptually, but in our empirical setup the evaluator’s role is very limited: it extracts numerical values from responses, and we discard cases where a value cannot be reliably extracted. For this class of tasks, different reasonable evaluators are likely to give very similar outputs.
> >
> >    In other, more subjective settings—for example, ranking responses into categories such as “great / ok / mid / bad / worst”—different evaluators could indeed produce different outcomes. We have added a discussion of this point and of the importance of sensitivity analysis in such cases, emphasizing that when evaluators encode more subjective judgments, results should be checked across multiple scoring rules or annotators.
> >
> > 3. Such a model is exactly the kind of behavior our framework is designed to distinguish from correctness. A model that is consistently wrong but tracks intent (e.g., always underestimates probabilities by a constant factor, but adjusts its answers appropriately when the scenario changes) would have high ICI but low correctness. A model that is inconsistent but occasionally correct would likely have low ICI even if its average error is similar.
> >
> >    Thus, we view intent comprehension and correctness as complementary axes. ICI answers, “Does the model appear to understand the prompted question (the intended latent variable)?” Accuracy answers, “Is the model’s world knowledge or mapping correct?” For example, a model prompted with “What is the cost of an average TV?” but trained only on data up to the early 2000s will likely give a numerically wrong answer but can still consistently understand and respond to that question with its knowledge. Improving accuracy there requires more or fresher knowledge, whereas low ICI would indicate a failure of generalization about the underlying intent.
> >
> >    Ideally we want models with both high ICI and high accuracy, but problems on each dimension call for different remedies. We will clarify this in the Discussion and stress that our method is not meant to replace accuracy-based metrics, but to isolate the semantic targeting component of understanding.
> >
> > We again appreciate the reviewer’s close reading. We believe that the clarifications and additions above will make our framework clearer, more transparent about its assumptions, and more practically useful.

---

> ### Comment · Reviewer_44rZ · 2025-11-20
>
> Thank you for the responses. The work’s main intentions and most of my questions have been addressed. Still, even though I understand the authors’ claim that dependence on human evaluators is inevitable, I cannot fully dismiss this as a major limitation. As a result, although the overall evaluation improves, my stance remains closer to a borderline rejection.

---

> > ### Comment · Reviewer_44rZ · 2025-11-20
> >
> > This is a minor point, but could you adjust the font size in the graphs and figures to be more consistent with the main text? I don’t think it affects the score, but it would contribute to a better overall presentation.

---

> ### Author Response · Authors · 2025-12-02
>
> Thank you for updating our score - we’ve increased the font size as you recommend! With regards to your comment on subjectivity, we agree that this dependence cannot be “designed away” in any framework that aims to measure whether model outputs should change when the underlying intent changes - intent after all is tightly related to the desired output from the model. Our contribution is therefore not to eliminate human judgment, but to make its role explicit and open to scrutiny. In most evaluations, normative assumptions and scoring rules are present but implicit—embedded in benchmark labels, rubrics, or dataset construction. We believe bringing these choices to the surface is essential for applying the framework across domains, and should be seen as improving transparency and auditability rather than adding a new limitation relative to standard practice.
>
> We thank you again for engaging with our paper!

---

### Official Review · Reviewer_HneT · 2025-10-31

**Soundness:** 3
**Presentation:** 3
**Contribution:** 3
**Rating:** 6
**Confidence:** 3

**Summary:**

This paper proposes a formal framework to measure intent comprehension in LLMs. It moves beyond standard accuracy benchmarks by decomposing the variance of model responses into three components: Intent Sensitivity (IS), Articulation Sensitivity (AS), and Model Uncertainty (MU). The goal is to quantify whether models respond to the user's underlying purpose (high IS) rather than superficial changes in wording (low AS).

**Strengths:**

The paper tackles the critical and timely problem of distinguishing between a model's understanding of user intent and its reliance on surface-level textual cues.

The variance decomposition (IS, AS, MU) is a principled and novel method for quantifying this distinction. The resulting "Intent Comprehension Index" (ICI) is an intuitive summary metric.

The use of cross-lingual translation chains to generate a diverse set of semantically equivalent prompts (paraphrases) is a smart and scalable approach to test articulation sensitivity.

**Weaknesses:**

The findings are somewhat inconclusive. While larger models show higher IS, they also show higher AS, suggesting they may be overfitting to surface cues as well as intent. The overall improvement in the ICI metric is described as "modest" with increasing model scale.

The evaluation is confined to open-ended, guesstimation-style tasks that require a numerical answer. This is a significant limitation, as it's unclear how these findings or the framework itself would generalize to more common, non-numeric, or creative generation tasks.

The paper acknowledges that AS and IS are not fully separable, and the results bear this out. The framework doesn't seem to fully resolve the trade-off between being sensitive to meaningful intent changes and being sensitive to all changes in text.

**Questions:**

N.A.

---

> ### Author Response · Authors · 2025-11-20
>
> We thank the reviewer for the careful and positive assessment. We address the main concerns below
>
> 1. Our intention was to be cautious rather than pessimistic in summarizing the scaling results. We see our findings as concrete contributions. Larger models allocate more variance to intent across domains, which is precisely the direction one would hope for if they better “understand the underlying purpose.” At the same time, AS also increases, revealing that bigger models are more sensitive to changes in wording. Without the decomposition, this trade-off would be invisible. The fact that ICI improvements are modest and not always monotone in scale is, in our view, an important and somewhat surprising result rather than a weakness of the method. It suggests that scaling alone does not automatically produce robust intent comprehension and that some of the additional capacity is due to overfitting and spent on exploiting surface cues. This, in turn, highlights the need for training objectives and datasets that explicitly encourage high IS and low AS, not just higher accuracy or likelihood. This underscores the utility of our framework for diagnosing where progress is (and is not) happening.
>
> 2. We agree that our empirical evaluation focuses on numeric guesstimation tasks because they provide a particularly clean setting: they allow a straightforward variance decomposition on a scalar value and are well-suited for probing partial understanding and uncertainty (for example, approximate probabilities, prices, or times). However, the framework itself is more general and is not restricted to numerical outputs.
> For discrete outputs such as ratings, categories, and labels, we already provide an entropy-based decomposition (currently in the appendix) in which variance is replaced by entropy and IS, AS, and MU are expressed as mutual-information terms. This directly extends our approach to non-numeric but discrete response spaces. For long-form or creative text, one can compose the raw model output with a value extractor—such as a human rubric, crowd ratings, or an evaluator model—that maps the text to a scalar or discrete label (for example, quality, style, stance, task success, or an embedding-based score). Our decomposition then applies to this induced variable, allowing intent comprehension to be assessed in more realistic generative settings. More generally, one can replace the (R^2)-style measure we use with other notions of explainable variation that are appropriate for the outcome space, while preserving the core idea of partitioning systematic variation into components aligned with intent and articulation. We do stress that the theoretical motivating discussion is not restricted to numerical values or discrete values, and the decomposition we introduce is appealing as it is both easy to compute and convey. Other approaches to operationalize our theoretical definition of intent comprehension are possible and we believe that researchers can easily implement them in other setups.  In the revision, we will bring a concise version of the discrete/entropy-based extension into the main text rather than leaving it entirely in the appendix, and we will add a dedicated paragraph in the Discussion explaining how to use task-specific evaluators (human or model-based) to apply the framework to non-numeric and creative generation tasks. Thus, while our current experiments are numerically oriented, the conceptual apparatus is designed to extend beyond numbers.
>
> 3. We agree that IS and AS are not independent “knobs” inside the model and are not separable in general in the training data. Our contribution is to make them conceptually separable and to use an experimental design that allows us to tease them apart, thereby offering researchers and practitioners a tangible target for evaluation. By independently varying intent (via template slots) and articulation (via paraphrases), we can attribute output variance to each source in a disciplined way. The fact that both IS and AS increase with scale is exactly the trade-off our framework is designed to quantify. Our framework is a measurement and decomposition tool; it does not guarantee that models can be made sensitive only to “good” changes (intent) and not to “bad” ones (all other textual variation), but it provides a structured way to see how much of each type of sensitivity a given model exhibits.
> We will clarify this in the paper by explicitly emphasizing the measurement role of the framework and by stating that the persistence of the trade-off in current models underscores the need for a tool like ours. Our hope is that future work will build on this framework to evaluate and develop new training or post-processing approaches aimed at increasing IS while reducing AS.
>
> We thank the reviewer again for their thoughtful comments! We believe that sharpening the discussion of the IS–AS trade-off will strengthen the paper and make the scope and significance of the framework clearer

---

> > ### Author Response · Authors · 2025-12-02
> >
> > We’ve implemented the correction in the revised version. Most importantly we clarified in the main text (lines 350–370) that our framework naturally accommodates non-numeric values. In addition, we expanded Appendix A to discuss how to handle free-form text response, and included a concrete demonstration showing how to operationalize our discrete decomposition, highlighting the versatility of the framework across a wide range of settings.
> >
> > We thank the reviewer again for his comments and helping make the paper better!

---

### Official Review · Reviewer_J2Vj · 2025-11-01

**Soundness:** 2
**Presentation:** 2
**Contribution:** 3
**Rating:** 2
**Confidence:** 4

**Summary:**

This paper presents a variance decomposition framework to evaluate whether large language models (LLMs) understand user intent. The authors first construct an experimental dataset comprising questions (tasks), intents, and back-translated paraphrased prompts across five domains to test model responses. Their evaluation approach decomposes the variance in model responses into three interpretable components: intent sensitivity, articulation sensitivity, and model uncertainty. Furthermore, they introduce two summary statistics, Meaningful Variability Share (MVS) and Intent Comprehension Index (ICI), that serve as metrics for assessing intent comprehension.

**Strengths:**

The paper presents a systematically designed experimental framework that decomposes response variance across models, domains, and tasks into interpretable components (intent sensitivity, articulation sensitivity, and model uncertainty) thereby providing a principled basis for analyzing how LLMs vary or remain consistent in their responses with respect to user intent and prompt articulation.

**Weaknesses:**

-	While the paper provides a useful measure of overall response consistency, it does not take into account the correctness of the responses. Understanding intent requires not only consistent responses across different articulations but also responses that are semantically relevant to the given questions. From this perspective, the proposed methods alone may not be sufficient to assess the model’s understanding of intent accurately.

-	While the paper presents extensive variance-based analyses across multiple model sizes, the reliability of its comparative conclusions remains questionable. In particular, the comparison between LLaMA 70B and Gemma models (12B, 27B) is not fully controlled, as it is unclear whether the observed differences in intent comprehension reflect parameter scaling effects or model-family differences.

-	The paper does not consider cases in which different intents may yield identical responses. In such cases, even if the model correctly understands the underlying intent, its Intent Sensitivity could be lower. Moreover, since the metric relies on responses that can be represented as numerical values to compute response variance, it may have limitations in evaluating whether the model truly comprehends questions.

**Questions:**

-	It would be helpful to include several paraphrased prompts (articulated questions) generated through back-translation in the Appendix.

-	The experiments could be more comprehensive by testing a wider range of temperature values, as evaluating only two settings appears insufficient.

-	The reliability of the experimental framework could be improved by verifying the constructed dataset, as it currently relies solely on GPT-4.1 for question and intent generation.

Typos & Minor Comments:

-	Each figure requires more detailed explanation; in particular, Figure 4 lacks a legend or color description.

-	The Appendix section should appear after the References.

-	The citation formats are inconsistent overall. When referring to a study within a sentence, the author’s name and publication year should appear in parentheses, unless it is used as part of the sentence.

-	In Figure 1, the sentence “What’s the price of traveling from Boston to Paris?” appears twice under different colors; the orange one should be taken from the Articulation Sensitivity section, and it should read “What would be the total cost for me to travel from Boston to Paris?”

-	In Figure 1, the term “Purpose Sensitivity” should be replaced with “Intent Sensitivity.”

-	Lines 494 and 620: the abbreviation PS should be unified consistently across the text.

-	Line 637: the origin or meaning of TU and PU should be clearly defined.

-	Line 318: there is an extra period that should be removed.

-	Line 664: “(see Appendix A)” likely refers to Appendix G.

-	Line 668: “Appendix B” should probably be Appendix H.

-	Footnote 5 (below line 755): the appendix link appears to be broken.

---

> ### Author Response · Authors · 2025-11-20
>
> We thank the reviewer for the careful reading and constructive comments. We are glad you find the framework and experimental setup useful. We address each concern below.
>
> **Weaknesses**
> 1. Our goal in this paper is to isolate intent comprehension from correctness, not to replace accuracy-based evaluation. This is why we explicitly define intent comprehension in terms of distributional invariance and sensitivity, and highlight in Remark 3 that our criteria are consistency-centric rather than correctness-centric. More so, the decomposition approach encodes a necessary condition for meaningful behavior: the output distribution must move with intent and remain stable across articulation. We agree, however, that correctness is essential in many applications and that this distinction should be stated more clearly. In the revision, we explicitly emphasize in the Discussion that our framework is complementary to accuracy metrics.
> When ground-truth labels are available, correctness can be incorporated into our framework by choosing the valuation function V to encode correctness. In the simplest case, V(a)=1{answer is correct}, so that we apply the same variance decomposition to the indicator of correctness rather than to the raw numeric answer. In degenerate cases where a model is always correct (or always wrong), V(a) is constant and there is no variance to decompose, which is appropriate: there is no remaining uncertainty about correctness. In the more realistic case where correctness varies across prompts and intentions our  framework can be combined with accuracy-based evaluation to diagnose whether failures are due to misunderstanding of intent, fragility to phrasing, or lack of world knowledge. We add this discussion to the Appendix.
> 2.  We agree that our current analysis mixes changes in parameter count with changes in model family and data, so one should be cautious in attributing differences to scaling per se. Indeed, we already observe that smaller Gemma models can perform similarly to larger LLaMA models, suggesting that training data and procedures matter alongside parameter count. In the revision we will (i) remove any language that could be read as a causal claim about scaling and (ii) more clearly distinguish within-family comparisons from cross-family comparisons.
> 3. In our formalization defines intent relative to an evaluator and value function (V) (Definition 2.2): two prompts share an intent if they are meant to induce the same distribution over evaluator-assigned values. In cases where two descriptions normatively should elicit the same response distribution (e.g., “fair coin: heads or tails?” vs. “choose heads or tails with equal probability”), they should be assigned the same intent label; low IS is then desirable. If one instead labels such cases as distinct intents while normative behavior keeps outputs identical, any metric—including accuracy—would penalize models for behaving correctly. In our experiments we avoid this by constructing intents so that changing the placeholder value changes the target magnitude from the evaluator’s perspective.
> 4. On the numeric focus, we agree that the framework might appear limited if read as purely variance-based. To address this, Appendix A already presents an entropy-based analogue for discrete outputs that allows for arbitrary labeling. In the revision we will push this into the main text and state that we use the variance-based decomposition for guesstimation tasks because it is simple and interpretable, but the same conceptual framework applies to arbitrary discrete outputs via the entropy decomposition. More broadly, our theoretical framework does not rely on scalar responses: any discrepancy that measures how response distributions change with intent and articulation (e.g., total variation distance) could be used, though at higher computational and substantially harder to communicate than the simple than our IS/AS/MU implementation.
>
> **Questions.**
> 1. We agree that concrete paraphrase examples are helpful. We will add an Appendix table with example prompt sets, showing the base prompt and several back-translated paraphrases for each domain.
> 2. We currently report results at temperatures 0.2 and 1.0 to represent low-entropy and more natural stochastic regimes. We will state this rationale explicitly and add results for an intermediate temperature in the Appendix.
> 3. Our reliance on GPT-4.1 is limited to generating templates/intents and producing and filtering paraphrases; it is never used as a correctness oracle. We will expand Section 4 and Appendix D to describe the filtering pipeline, add a human validation study on a random subset of questions, intents, and paraphrases, and release the dataset and prompt-generation code for external inspection and reuse.
>
> Thank you again for your thoughtful engagement with our work. We hope these clarifications help in your assessment.

---

> > ### Author Response · Authors · 2025-12-02
> >
> > We have implemented all the changes described above in the revised manuscript. Most importantly, we added a detailed discussion—based directly on your review and addressing several related reviewer concerns—in Appendix E, explaining how correctness can be incorporated into our framework by defining V(.) as a function of correctness.
> >
> > We have also clarified in the main text (lines 350–370) that our framework naturally accommodates non-numeric values. In addition, we expanded Appendix A to discuss how to handle free-form text response, and included a concrete demonstration showing how to operationalize our discrete decomposition, highlighting the versatility of the framework across a wide range of settings.
> >
> > Furthermore, we added a human evaluation in Appendix B, which shows that the paraphrasing and extractions produced by GPT-4.1 align closely with human judgments.
> >
> > Finally, we included several concrete paraphrase examples in Appendix XX and added results generated with temperature 0.5.
> >
> > We thank the reviewer for engaging thoughtfully with our paper!

---

### Author Response · Authors · 2025-12-02

Dear AC,

Our work formalizes **intent comprehension** as *distributional invariance* of a model’s responses to surface-form paraphrases when underlying intent is fixed, paired with *systematic shifts* in the response distribution when intent changes. We then **measure** this property via an interpretable decomposition of output variability into **Intent Sensitivity (IS)**, **Articulation Sensitivity (AS)**, and **Model Uncertainty (MU)**.

A main reviewer concern was applicability beyond **numeric outputs**. We emphasize that the framework is **not restricted to numeric responses**: it applies to any output type by choosing an appropriate value/representation function (V(\cdot)) (e.g., labels, scores, or embeddings), and we expanded both the main text and **Appendix A** to explicitly cover free-form and discrete/categorical settings.

A second concern was validating our intent-preserving paraphrase pipeline and numeric extraction from model responses. We added a **human evaluation** of the paraphrase/extraction pipeline (two independent annotators), finding that paraphrases overwhelmingly preserve intent and that the extractor is correct for the large majority of sampled outputs.

Finally, we addressed the remaining concerns raised in discussion (e.g., clarifying the role of the evaluator and sensitivity analysis, clarifying that “prompt” refers to the full model input/context, and tightening claims and presentation where needed). Thank you, we appreciate your effort in assessing the paper and the reviewer feedback.

---

### Meta-Review · Area_Chair_4zwx · 2025-12-30

**Summary:**

This submission proposes a variance decomposition framework to measure LLM intent comprehension via IS, AS, and MU metrics. Reviewers’ core concerns include: (1) self-defined metrics lacking external validation and practical utility (failing to account for response correctness); (2) heavy reliance on GPT model for data generation, limiting reproducibility; (3) initial restriction to numeric outputs and narrow model coverage. Besides, there are also many pervasive typos, citation errors, and flawed visualizations, and methodological ambiguity in intent-articulation distinction and human evaluator subjectivity. Therefore, I think the weaknesses of this paper may outweigh the strengths and would recommend rejection.

**Reviewer Concerns:**

Addressed concerns: The authors partially resolved non-numeric extension concerns and validated paraphrase intent-preservation with human evaluation. They also clarified the temperature setting rationale and revised some figures.

However, some concerns remain unresolved, including (1) kind of narrow scope of the metric, (2) some metrics still lack validation against real-world outcomes; and (3) how the proposed metric can be leveraged to help improve other tasks.

**Reviewer Scores:**

Reviewer J2Vj (initial score: 2), Reviewer HneT (initial score: 6), Reviewer bj5p (initial score: 6) will likely to maintain the score. Reviewer 44rZ (initial score: 2) and Reviewer GktL (initial score: 4) will potentially increase the score. Overall, this paper is a borderline case and the rejection may outweigh the acceptance.

---

### Decision · Program_Chairs · 2026-01-26

Reject